
# Matchmakereft: Automated tree-level and one-loop matching

Adrián Carmona[1,2], Achilleas Lazopoulos[2], Pablo Olgoso[1] and José Santiago[1*]

**1** CAFPE and Departamento de Física Teórica y del Cosmos, Universidad de Granada, Campus de Fuentenueva, E–18071 Granada, Spain
**2** Institute for Theoretical Physics, ETZ Zürich, 8093 Zürich, Switzerland

⋆ jsantiago@ugr.es

## Abstract

We introduce `matchmakereft`, a fully automated tool to compute the tree-level and one-loop matching of arbitrary models onto arbitrary effective theories. `Matchmakereft` performs an off-shell matching, using diagrammatic methods and the background field method (BFM) when gauge theories are involved. The large redundancy inherent to the off-shell matching together with explicit gauge invariance offers a significant number of non-trivial checks of the results provided. These results are given in the physical basis but several intermediate results, including the matching in the Green basis before and after canonical normalization, are given for flexibility and the possibility of further cross-checks. As a non-trivial example we provide the complete matching in the Warsaw basis up to one loop of an extension of the Standard Model with a charge −1 vector-like lepton singlet. `Matchmakereft` has been built with generality, flexibility and efficiency in mind. These ingredients allow `matchmakereft` to have many applications beyond the matching between models and effective theories. Some of these applications include the one-loop renormalization of arbitrary theories (including the calculation of the one-loop renormalization group equations for arbitrary theories); the translation between different Green bases for a fixed effective theory or the check of (off-shell) linear independence of the operators in an effective theory. All these applications are performed in a fully automated way by `matchmakereft`.

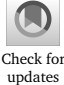

# 1  Introduction

Effective field theories (EFTs) are the most appropriate tool to perform calculations in multi-scale problems when using mass-independent renormalization schemes. The process of matching and running turns an often complicated, and sometimes not even perturbatively convergent, multi-scale problem into a succession of simpler single-scale calculations with the possibility of resummation of large logarithms for a better perturbative convergence [1]. The absence of direct experimental indications of new physics beyond the Standard Model (SM) seems to imply a hierarchy between the scale of new physics and the energies at which experiments are performed. In these circumstances, EFTs are not only applicable, but they become

a powerful tool that allows us to very efficiently solve the problem of comparing experimental data with theoretical predictions. This is a highly non-trivial problem that involves complicated, usually loop-level, calculations of many experimental observables. Calculations that have to be repeated for each new physics model and for each observable. The way EFTs simplify this process is by splitting in two, mostly independent, steps. The first one, the bottom-up approach to EFTs, provides a model-independent parametrization of experimental observables that can be systematically improved but has to be computed only once (for the given precision) for each experimental observable. This efficient parametrization can be provided in the form of global fits to experimental data and a very important effort from the community has been devoted to this task in the last few years (see [2–13] and references therein for some recent global fits). The second step, the top-down approach to EFTs, sacrifices model-independence in favour of model-discrimination. It consists of the process of matching, in which the Wilson coefficients (WCs) of the EFT are computed in terms of the parameters of the ultraviolet (UV) model. This process of matching has to be repeated for each UV model but it can be automated.

When dealing with searches for new physics beyond the SM, the relevant EFT seems to be the SMEFT (see [14] for a recent review). The SMEFT WCs can be run down to the electroweak scale thanks to the renormalization group equations (RGEs) computed in [15–18] at dimension 6 ignoring baryon and lepton number violating operators (see [19,20] for calculations of a subset at dimension 6, [21,22] for the inclusion of lepton-number violating operators, [18] for baryon-number violating ones, [23] for dimension-7 operators and [24,25] for recent efforts towards the calculation of the dimension 8 RGEs). The SMEFT has been matched to the low-energy EFT (LEFT), the relevant EFT below the electroweak scale, both at tree level [26] and at one-loop [27] and then run again using the LEFT RGEs [28] to the relevant experimental energy scale to compute the corresponding experimental observables. This process of running in the SMEFT, matching to the LEFT and running in the LEFT has been implemented in automated tools like `DsixTools` [29,30] or `Wilson` [31]. This leaves the calculation of the matching of arbitrary UV models onto the SMEFT as the only missing step towards a fully automated calculation of the phenomenological implications of new physics models.

We present in this article `matchmakereft`, https://ftae.ugr.es/matchmakereft/, a fully automated tool to perform tree-level and one-loop matching of arbitrary UV models onto arbitrary EFTs. The tree-level matching of the most general extension of the SM onto the SMEFT at dimension 6 has been recently computed in [32], building on previous partial results [33–36]. Going up to the one-loop order in the matching is far more complex and some degree of automation is needed. Functional methods, pioneered in [37] extending ideas from the 80s [38,39], have seen an impressive progress in the recent years [40–51] and they have resulted in computer tools that help with some of the most computational-intensive steps of the calculation [52,53] or that are automated but apply only to specific sets of models [54]. However, to the best of our knowledge, there is currently no code that can provide the complete one-loop matching of arbitrary models onto arbitrary EFTs in a fully automated way. [1] `Matchmakereft` comes to fill this gap.

`Matchmakereft` uses a diagrammatic approach to tree-level and one-loop matching, performed in the BFM when gauge theories are involved. The matching is, as of version `1.0.0`, done off-shell which, together with gauge invariance, provides a significant redundancy that results in a number of non-trivial cross-checks of the calculation. It has been designed with efficiency, generality and flexibility in mind, what allows a number of applications beyond the direct matching of UV models to EFTs. Current applications include the renormalization of arbitrary (effective) theories, the calculation of the RGEs of arbitrary (effective) theories,

---

[1]See [55, 56] for alternative partial efforts using a diagrammatic approach and [57] for the calculation of specific Wilson coefficients at one loop in arbitrary extensions of the SM.

EFT basis translation and checks of (off-shell) linear independence of operators. All these calculations are done in a fully automated way.

The rest of the article is organised as follows. We describe the way `matchmakereft` performs the tree-level and one-loop calculation in Section 2. Model creation in `matchmakereft` is explained in detail in Section 3. The different commands available in `matchmakereft` are defined in Section 4 and common pitfalls when using `matchmakereft` are described in Section 5. Some physical applications are given in Section 6 and we conclude and provide some outlook in Section 7. In the appendices we provide some more technical information, including comprehensive installation instructions in Appendix A, details about the handling of $\gamma_5$ in Appendix B, a minimal but complete example of the capabilities of `matchmakereft` in Appendix C, as well as the complete Green basis for the baryon-number preserving SMEFT, including our definition of evanescent operators, in `matchmakereft` in Appendix D.

## 2 `Matchmakereft` **in a nutshell**

### 2.1 Model classification

Models in `matchmakereft` are classified according to two criteria. Depending on their spectrum they can be **light models**, if only light (but not necessarily massless) particles are present in the spectrum; or **heavy models**, when there is at least one heavy particle in the spectrum. Depending on their role in the process of matching we have **UV models**, which can be light or heavy, that are to be matched onto **EFTs**, which are necessarily light models. Models in `matchmakereft` are created using `FeynRules` [58], as described in detail in Section 3.

`Matchmakereft` performs an off-shell matching, in the BFM when gauge theories are involved, of a UV model onto an EFT. This is done by computing, in dimensional regularization in $D = 4 - 2\epsilon$ space-time dimensions, the hard-region contribution to the one-light-particle-irreducible relevant amplitudes at tree and one-loop level in the UV theory and equating it to the tree level contribution in the EFT for arbitrary kinematic configurations of the external particles. The matching is performed by equating $\overline{\text{MS}}$-renormalized amplitudes. The one-loop contribution from the EFT is scaleless and therefore vanishes. In any case its UV poles are renormalized away and the infrared (IR) ones are cancelled by the soft region contribution of the full theory amplitude. An excellent discussion of the matching process can be found in the two recent reviews [59, 60]. In `matchmakereft` we prefer to keep explicitly the $1/\bar{\epsilon}$ terms to allow for the automatic incorporation of evanescent operators and also to provide further information to the user. Of course, the user should remove the $1/\bar{\epsilon}$ (denoted `invepsilonbar` in `matchmakereft`) terms explicitly from the renormalized Wilson coefficients in the physical basis. The amplitudes, with only physical external light particles, that have to be computed are fixed in an automated way by `matchmakereft` but the user has flexibility on changing this list as described in Section 4. All the relevant diagrams in the UV model and the EFT are then automatically computed by `QGRAF` [61] and the corresponding amplitudes are dressed by `matchmakereft` using the Feynman rules computed during the creation of the model.

### 2.2 Amplitude calculation

`Matchmakereft` runs in two different modes, depending on whether the UV model is light or heavy, called `RGEmaker` and `Matching` modes, respectively. In `RGEmaker` mode, which is used to compute the RGEs of an arbitrary theory, the UV model contains no heavy particles and `matchmakereft` computes the UV-divergent contribution proportional to $1/\epsilon$ of the corresponding one-particle-irreducible amplitudes. In `Matching` mode, there are heavy particles in the spectrum and both the finite and $1/\epsilon$ (both UV and IR) hard-region contributions to

the corresponding one-light-particle-irreducible amplitudes are computed. In this case diagrams including only light particles are not included in the calculation, as they cancel in the matching.

The calculation of the hard region contribution to the amplitudes is performed with FORM [62] and proceeds as follows:

- **Hard region expansion**. This corresponds to the following expansion of the integrand of the amplitude

$$k^2 \sim M^2 \gg p^2 \sim m^2 \,, \tag{1}$$

where $k$ represents the loop momentum, $M$ a heavy mass, $p$ any of the external momenta and $m$ a light mass. This is done by iterating the following identities

$$\frac{1}{(k+p)^2 - M^2} = \frac{1}{k^2 - M^2}\left[1 - \frac{p^2 + 2k \cdot p}{(k+p)^2 - M^2}\right],$$

$$\frac{1}{(k+p)^2 - m^2} = \frac{1}{k^2}\left[1 - \frac{p^2 + 2k \cdot p - m^2}{(k+p)^2 - m^2}\right]. \tag{2}$$

These identities are imposed iteratively until the power of infrared (IR) scales (external momenta or light masses) is the correct one to match the maximum dimension of the operators appearing in the EFT as automatically computed in `matchmakereft`.

- **Tensor reduction**. Tensor reduction is performed by means of the following identities

$$k^{\mu_1} k^{\mu_2} = g^{\mu_1 \mu_2} \frac{k^2}{D} \,, \tag{3}$$

$$k^{\mu_1} k^{\mu_2} k^{\mu_3} k^{\mu_4} = g^{\mu_1 \mu_2 \mu_3 \mu_4} \frac{k^4}{D^2 + 2D} \,, \tag{4}$$

$$k^{\mu_1} \dots k^{\mu_6} = g^{\mu_1 \cdots \mu_6} \frac{k^6}{D^3 + 6D^2 + 8D} \,, \tag{5}$$

$$k^{\mu_1} \dots k^{\mu_8} = g^{\mu_1 \cdots \mu_8} \frac{k^8}{D^4 + 12D^3 + 44D^2 + 48D} \,, \dots \tag{6}$$

where $g^{\mu_1 \cdots \mu_n}$ is the totally symmetric combination of metric tensors.

- **Dirac algebra**. Once the integrals have been reduced to scalar integrals we proceed to perform the corresponding Dirac algebra, in $D$ dimensions in `Matching` mode (in 4 dimensions in `RGEmaker` mode). Version `1.0.0` of `matchmakereft` uses an anticommuting $\gamma^5$ prescription as discussed in Section 2.3. In the case of fermion number violating particles and/or interactions, we follow the rules proposed in Ref. [63, 64].

- **Partial fractioning**. The following identity is used to separate propagators with different masses

$$\frac{1}{(k^2 - m_1^2)(k^2 - m_2^2)} = \frac{1}{m_1^2 - m_2^2}\left[\frac{1}{k^2 - m_1^2} - \frac{1}{k^2 - m_2^2}\right], \tag{7}$$

where masses can be light or heavy and one of them can be vanishing.

- **Integration by parts**. After partial fractioning scaleless integrals are set to zero, except in `RGEmaker` mode, in which case we keep the UV poles using,

$$\int \frac{d^D k}{(2\pi)^D} \frac{1}{k^4} = \frac{i}{(4\pi)^2} \frac{1}{\epsilon} + \dots \,, \tag{8}$$

before eliminating the remaining scaleless integrals. As we explained in Section 2.1 scaleless integrals can be safely set to zero when computing the hard-region contribution to the one-loop amplitude in the full theory. In `RGEmaker` mode the UV poles determine the anomalous dimensions and therefore we have to keep them before setting the scaless integrals to zero. In `Matching` mode the following identity is used to reduce the massive integrals (which appear due to the hard-region expansion) to tadpoles

$$\frac{1}{(k^2 - m^2)^{n+1}} = \frac{D - 2n}{2nm^2} \frac{1}{(k^2 - m^2)^n} \,. \tag{9}$$

At this point we are left with a tadpole integral

$$a_0(m) = \int \frac{d^D k}{(2\pi)^D} \frac{1}{k^2 - m^2} = i \frac{m^2}{16\pi^2} \left[ \frac{1}{\bar{\epsilon}} + 1 - \log\left(\frac{m^2}{\mu^2}\right) \right] + \mathcal{O}(\epsilon), \tag{10}$$

where

$$\frac{1}{\bar{\epsilon}} \equiv \frac{1}{\epsilon} + \gamma_E - \log(4\pi), \tag{11}$$

with $\gamma_E \approx 0.5772$ the Euler-Mascheroni constant. In `matchmakereft` $1/\bar{\epsilon}$ is denoted by `invepsilonbar`.

## 2.3 About $\gamma^5$ and evanescent operators

In order to ensure maximum generality `matchmakereft` assumes no four-dimensional properties when running in `Matching` mode. In particular no Fierz relations or reduction of products of three or more gamma matrices is performed during the matching procedure. This means that all evanescent structures, which are equivalent to operators in the Green basis only in $D = 4$ have to be explicitly defined as part of the Green basis. As an example, we provide in Appendix D the complete Green basis for the SMEFT at dimension 6 as needed for the matching with `matchmakereft` of general theories with renormalizable couplings. [2] This extends the Green basis of [65] with the general set of evanescent structures including fermionic operators.

Regarding $\gamma^5$ it is well known that its implementation in dimensional regularization schemes is problematic (see [66] and references therein for an account of the current status). The current version of `matchmakereft` (1.0.0) implements an anticommuting $\gamma^5$ together with an implementation of the hermiticity properties of the WCs that is enough for the case of the SMEFT (and extensions with multiple scalar doublets) as we discuss more carefully in Appendix B.

## 2.4 Wilson coefficient matching

Once the amplitudes have been computed, the output is written in two files for each amplitude, one including the gauge structure and the other including the kinematic and flavour structures. `Matchmakereft` then uses `Mathematica` to match the amplitudes for all kinematic configurations between the UV model and the EFT both at tree level and one loop. Once the off-shell matching has been performed, that is the Wilson coefficients of the Green basis have been computed in terms of the couplings and masses of the UV model, `matchmakereft` automatically performs a canonical normalization of the results and then reduces the matching to a physical basis as defined by the user (see Section 4 for details). Results at all three levels (Green basis

---

[2]Non-renormalizable theories can be matched also with `matchmakereft` but an extension of the basis with a larger number of gamma matrices in four-fermion operators would be needed. Similarly, if bosonic evanescent operators appear in the process of matching a specific UV model they would have to be included in the EFT basis.

with non-canonical kinetic terms, canonically normalised Green basis and Physical basis) are reported by `matchmakereft` together with the corresponding renormalisation of the gauge couplings as fixed by gauge boson renormalisation in the BFM. If `RGEmaker` mode has been invoked then `matchmakereft` can also automatically compute the beta functions for all the WCs of the EFT used.

The off-shell matching used introduces a large degree of kinematic redundancy. In gauge theories the BFM also introduces a large degree of explicit gauge redundancy. These redundancies provide a very powerful mechanism to cross-check the results obtained in `matchmakereft`, which checks that all kinematic configurations and gauge directions are correctly matched. When any of these redundacies is not fulfilled `matchmakereft` will issue a warning and provide some extra information that can be useful to debug the problem.

## 3 Model creation

Model creation is fully automated in `matchmakereft` but it relies on the explicit input from the user. Thus, it is the step that should be performed with the greatest care, as it is the most likely culprit in case of problems with the matching calculation. Model creation is greatly simplified by the use of `FeynRules` but a few important points should be taken into account.

### 3.1 Required files

`Matchmakereft` expects a number of files with all the relevant information to create the model. The detailed structure of each of these files will be defined below but we list them here first: [3]

- Model files (compulsory): one or more files `modfile1.fr`, ..., `modfilen.fr` that define the model in `FeynRules` format. One of the files, the last one of the list during the creation of the model (see below) is special, as it will define the name of the `matchmakereft` model, which is `modelfilen_MM`, and the name that extra files with additional information need to have. `Matchmakereft` **expects the Lagrangian to be defined as** `Ltot`. If a different name is used the model will not be created.

- Gauge information file (compulsory only if gauge groups are present): a file called `modfilen.gauge` that has the definition of all the gauge functions, including structure constants, group generators in different representations and Clebsch-Gordan coefficients, appearing in the model (see below for more information). The user can choose any gauge basis of interest but they are responsible for the consistency of the chosen basis.

- Symmetry file (optional): a file called `modfilen.symm` that indicates possible symmetries in the parameters of the model. This is particularly important in the case of the EFT model and it is compulsory in this case if symmetries are present. The content of the file should be a `Mathematica` list in which the symmetries are given in the form of replacement rules. As an example we show the case of the symmmetries of the Wilson coefficient of the Weinberg operator (denoted by `alphaWeinberg[i,j]=alphaWeinberg[j,i]`) and the four-lepton operator $\mathcal{O}_{\ell\ell} = \bar{\ell}_i \gamma^\mu \ell_j \bar{\ell}_k \gamma_\mu \ell_l$ (denoted by `alphaOll[i,j,k,l]= alphaOll[k,l,i,j]`):

---

[3] The installation of `matchmakereft` comes with a number of sample models that can be obtained with the command `copy_models` (see below). We encourage the user to check these examples for details on how to implement new models.

```
1  listareplacesymmetry=
2      {
3      alphaWeinberg[i_, j_] -> alphaWeinberg[j, i],
4      alphaWeinbergbar[i_, j_] -> alphaWeinbergbar[j, i],
5      alphaOll[i_, j_, k_, l_] -> alphaOll[k, l, i, j]
6      }
```

where the underscore denoting dummy indices on the left hand side of the rules is compulsory and the `bar` at the end of a name denotes complex conjugation.

- Redundancy file (compulsory): a file called `modfilen.red` that provides the redundancies that express the Wilson coefficient of a physical basis in terms of the ones in the Green basis. It is compulsory but it can be empty if no redundancies are needed (for example if the model is a UV model, if the physical and Green bases coincide or if we just want the results in the Green basis).

- Hermiticity properties file: in `matchmakereft`, complex conjugation is a very important process, providing extra cross-checks of the correctness of the calculation. For that reason it is important to provide the information of which WCs have special (anti)hermiticity properties. A file called `modfilen.herm` can be provided by the user defining a list in `Mathematica` format called `listahermiticity` defining those WCs whose hermitian conjugate can be defined in terms of the original coefficient. As an example, for the case of the hermitian operator $(\mathcal{O}_{Hq}^{(1)})_{ij} = H^\dagger i \overleftrightarrow{D}_\mu H \bar{\ell}_i \gamma^\mu \ell_j$, its Wilson coefficient, denoted `alphaOHq1[i,j]` satisfies `alphaOHq1bar[i,j]=alphaOHq1[j,i]`. This is provided in the form of the hermiticity file as follows

```
1  listahermiticity = {
2                       alphaOHq1bar[i_,j_]->alphaOHq1[j,i]
3                       }
```

## 3.2 Gauge structure

`Matchmakereft` is especially efficient when the matching is performed in the unbroken phase of gauge theories as it keeps gauge indices as dummy indices during the calculation of the amplitudes, replacing them with their explicit values only at the end of the calculation. When creating a model, all gauge functions, including structure constants, generators in different representations and Clebsh-Gordan coefficients need to have a specific name that does not correspond to any function already present in `Mathematica` or `FeynRules`. Structure constants and generators do not have to be defined as `FeynRules` parameters but Clebsch-Gordan coefficients do. The numerical values of these gauge functions are provided in a file with a `modfilen.gauge` extension with the definition of a mathematica list called `replacegaugedata` that consist of a list of substitutions in the form of `Mathematica` sparse arrays. As a simple example, the $SU(2)_L$ weak gauge group can be defined as follows in one of the `.fr` files:

```
1  M$GaugeGroups = {
2    SU2L == {
3      Abelian            -> False,
4      CouplingConstant   -> g2,
5      GaugeBoson         -> Wi,
6      StructureConstant  -> fsu2,
7      Representations    -> {{Ta,SU2D}}
8    }
9  },
```

where we have defined the structure constant symbol and one representation with generator symbol Ta and index definition SU2D. Note that the adjoint representation does not need to be explicitly defined as it is defined by the structure constants and the definition of the corresponding gauge bosons which, in this case reads (again, provided in one of the `.fr` files),

```
M$ClassesDescription = {
  V[2] == {
    ClassName       -> Wi ,
    SelfConjugate   -> True ,
    Indices         -> {Index[SU2W]},
    Mass            -> 0 ,
    FullName        -> "light"
  }
};
```

A few things are worth emphasizing from the above example. First, the mass is set to zero because we are in the unbroken phase of the SM. Second, we define physical fields in entire gauge multiplets, rather than components. [4] Finally, we assign the FullName variable to "light" (FullName->"light"). This is compulsory in matchmakereft. Every particle has to be defined with FullName equal to either "light" or "heavy" to define the corresponding particle as light (and therefore to be kept in the EFT model) or heavy (to be integrated out in the UV model).

The corresponding indices (not only gauge, also flavour indices if present) have to be defined with a finite range within the `.fr` files. As an example the ones corresponding the the adjoint (SU2W) and fundamental (SU2D) representations of $SU(2)_L$, together with flavour indices for fermion generations (Generation), can be defined as follows

```
IndexRange[Index[SU2W]]        = Range[3];
IndexRange[Index[SU2D]]        = Range[2];
IndexRange[Index[Generation]] = Range[3];
IndexStyle[SU2W,n];
IndexStyle[SU2D,l];
IndexStyle[Generation, fl];
```

Only massless particles can have flavour indices in the current version of matchmakereft (1.0.0).

In order to show how new particles with non-trivial quantum numbers and Clebsch-Gordan coefficients are defined we show here the case of a heavy scalar triplet under $SU(2)_L$ and the SM Higgs

```
M$ClassesDescription = {
  S[105] == {
    ClassName       -> tphi ,
    SelfConjugate   -> True ,
    Indices         -> {Index[SU2W]},
    Mass            -> Mtphi ,
    FullName        -> "heavy",
    QuantumNumbers -> {Y -> 0}
  },

  S[11] == {
    ClassName       -> Phi ,
    Indices         -> {Index[SU2D]},
    SelfConjugate  -> False ,
    Mass            -> muH ,
    FullName        -> "light",
```

---

[4]It is actually possible to define also the field components separately as the physical fields. In fact, this can be advantageous when creating complicated models that take very long to generate in FeynRules.

```
17        QuantumNumbers -> {Y -> 1/2}
18    }
19
20 };
```

where the new particle is defined as heavy and has a non-zero mass while the SM Higgs boson is defined as light (but also has a non-vanishing mass). We also see that $U(1)$ quantum numbers have to be defined explicitly as is standard in `FeynRules`. A trilinear coupling between the heavy scalar and two Higgs bosons, which has a non-trivial gauge structure following the corresponding Clebsch-Gordan coefficients, has to be defined explicitly like in the following example

```
1 M$Parameters = {
2
3   C223 == {
4     ParameterType      -> Internal,
5     Indices -> {Index[SU2D],Index[SU2D],Index[SU2W]},
6     ComplexParameter   -> True
7   },
8
9 ...
10 }
```

and in the corresponding Lagrangian

```
1 Ltot := Block[{ii,jj,nn},
2     2 C223[ii,jj,nn] kappatphi tphi[nn] Phibar[ii] Phi[jj] + ...]
```

As mentioned above the explicit values of the gauge functions are given in a file called `modfilen.gauge` (assuming the last `FeynRules` model file is called `modfilen.fr`) which, in the example we are showing would contain the following information

```
1 replacegaugedata = {
2    fsu2 -> SparseArray[Automatic, {3, 3, 3}, 0,
3            {1, {{0, 2, 4, 6}, {{2, 3}, {3, 2},
4                {1, 3}, {3, 1}, {1, 2}, {2, 1}}},
5                      {1, -1, -1, 1, 1, -1}}],
6       Ta -> SparseArray[Automatic, {3, 2, 2}, 0,
7            {1, {{0, 2, 4, 6}, {{1, 2}, {2, 1},
8                {1, 2}, {2, 1}, {1, 1}, {2, 2}}},
9                {1/2, 1/2, -I/2, I/2, 1/2, -1/2}}],
10       C223 -> SparseArray[Automatic, {2, 2, 3}, 0,
11            {1, {{0, 3, 6}, {{1, 3}, {2, 1}, {2, 2},
12                {1, 1}, {1, 2}, {2, 3}}},
13                {1/2, 1/2, -I/2, 1/2, I/2, -1/2}}]}
```

where we have implemented the usual definitions, $\texttt{fsu2[i,j,k]} = \epsilon^{ijk}$, $\texttt{Ta[a,i,j]} = \sigma^a_{ij}/2$ and $\texttt{C223[i,j,a]} = \sigma^a_{ij}/2$. The simplest way to define these matrices is by applying the `Mathematica` function `SparseArray` to the corresponding arrays defined by the user and copying the output to a file.

### 3.2.1 Background field method

`Matchmakereft` assumes that the BFM [67] is used when gauge theories are involved. Following the $SU(2)_L$ example, the associated quantum and ghost fields have to be defined on top of the definition of `Wi` given above

```
1  V[102] == {
2     ClassName          -> WiQuantum,
3     SelfConjugate      -> True,
4     Indices            -> {Index[SU2W]},
5     Mass               -> 0,
```

```
6      FullName         -> "light"
7 },
8   U[1] == {
9     ClassName        -> ghWi,
10    SelfConjugate    -> False,
11    Indices          -> {Index[SU2W]},
12    Ghost            -> Wi,
13    QuantumNumbers   -> {GhostNumber -> 1},
14    Mass             -> 0,
15    FullName         -> "light"
16   },
17 ...
```

and the Lagrangian involving the $SU(2)_L$ part is given by

```
1 gotoBFM={Wi[a__]->Wi[a]+WiQuantum[a]};
2
3 Ltot :=
4 Block[{mu,nu,ii,aa},
5 -1/4 (FS[Wi,mu,nu,ii] FS[Wi,mu,nu,ii])/.gotoBFM
6 -ghWibar[aa].DC[(DC[ghWi[aa],mu]/.gotoBFM),mu]
7 -DC[WiQuantum[mu,aa],mu] DC[WiQuantum[nu,aa],nu]/2
8 ]
```

Note the compulsory use of Ltot for the name of the total Lagrangian of the model.

### 3.3  Defining the EFT model

As we have mentioned above, the EFT model has to be a light model (no heavy particles) and has to include all the relevant operators in a Green basis. The **WCs in the EFT need to have a special format**. They have to be named alphaXXX, where XXX stands for an arbitrary number of alpha-numeric characters (see Section 3.4). The amplitudes that matchmakereft computes are defined by the operators in the EFT. In that sense, one does not need to include all the operators of a Green basis but at least all the operators in a certain class (same fields), including redundant and evanescent operators. [5] Matchmakereft will then generate only the relevant amplitudes to match these operators. Unless the user is absolutely sure that they do not need them, renormalizable operators, including kinetic and mass terms, have to be included in the EFT model.

When performing the matching, matchmakereft will check that all off-shell kinematic configurations and all gauge directions are correctly matched. If these checks are not satisfied, matchmakereft will issue a warning, and store the relevant information. This usually happens because the user made a mistake when defining the models, either because the model is not correctly defined or because there are missing operators in the Green basis. See Section 5 for common problems and possible solutions when running matchmakereft. When doing the external momentum expansion all operators of dimension equal or smaller to the highest dimension of the operators appearing in the EFT will be generated. Thus, one has to include all operators of smaller dimensions within the same class. Failing to do so will make the matching fail but the user can check that all problems appear in sectors that are of no interest for them (for instance in lower-dimensional operators that have not been implemented). Also sometimes some amplitudes are not correctly matched due to the use of an anticommuting $\gamma^5$. Our proposed solution should be enough to ensure the correct results in the SMEFT and similar EFTs (see discussion in Appendix B) but a warning will still be issued and the relevant information stored so that the user can check if the solution is correct or not.

If the user is interested in the matching result in a physical basis, they have to provide the corresponding redundancies to reduce the WCs of the Green basis into the ones of the

---

[5]See section 2.3 for a discussion about evanescent operators.

physical basis. This is done in the file denoted `modfilen.red`. As an example, let's consider the following redundant operators in the SMEFT at dimension 6

$$\mathcal{O}_{HD} = (H^\dagger D_\mu H)^\dagger (H^\dagger D_\mu H),\tag{12}$$

$$\mathcal{R}_{BDH} = (H^\dagger \overleftrightarrow{D}_\mu H)\partial_\nu B^{\mu\nu} \to g_1 \mathcal{O}_{HD} + \dots,\tag{13}$$

$$\mathcal{R}_{2B} = -\frac{1}{2}(\partial_\mu B^{\mu\nu})(\partial^\rho B_{\rho\,\nu}) \to -\frac{g_1^2}{2}\mathcal{O}_{HD} + \dots,\tag{14}$$

where the $\to$ indicates an on-shell equivalence. These redudancies imply the following on-shell condition for the corresponding Wilson coefficients (with obvious notation)

$$\texttt{alphaOHD} \to \texttt{alphaOHD} + 2g_1\texttt{alphaRBDH} - \frac{g_1^2}{2}\texttt{alphaR2B},\tag{15}$$

which we implement in the file `modfilen.red` as follows

```
finalruleordered={
  alphaOHD ->alphaOHD + 2*alphaRBDH*g1 - (alphaR2B*g1^2)/2,
  ...
  }
```

## 3.4 Protected keywords

There is quite a bit of flexibility in the process of model definition in `matchmakereft` but there are a few keywords that are protected and should be used only for their specific purpose. In general, all variables in `matchmakereft` should be made of alphanumeric characters, without including any special characters. The list of protected variables are the following

- `alpha`. All WCs should be defined as `alphaXXX` where `XXX` is an arbitrary string of alphanumeric characters. Similarly no other variable in the model should contain the substring `alpha`. An exception to this rule is that when computing the RGEs of an EFT the Wilson coefficients in the UV model can be kept with their original `alphaXXX` name as they will be changed into `WCXXX` automatically by `matchmakereft`.

- `Ltot`. Ltot should be used to define the complete Lagrangian of the model (and it should not be used for anything else).

- `Quantum`. Gauge bosons are split into a classical background and a quantum excitation. If the classical gauge boson is defined by `ClassName->Vname` then the quantum excitation has to be defined by `ClassName->VnameQuantum`.

- `invepsilonbar` is used for the dimensional regularization variable $1/\bar{\epsilon}$ so it should not be use explicitly in the definition of a model. Similarly `epsilonbar` is used of $\bar{\epsilon}$.

- `Eps[]` denotes the `FeynRules` Levi-civita tensor. When used with four indices it is interpreted by `matchmakereft` as the Minkowskian (with $+---$ metric signature) totally antisymmetric tensor and should therefore not be used for the Euclidean one (with a number of indices different from four it can be used as the Euclidean one). In case one needs to use the totally antisymmetric rank-4 tensor both with Minkowskian and Euclidean signatures, the latter should be explicitly defined as a gauge function and its numerical value defined in the corresponding `modfilen.gauge` file.

- `ee[]` and `dd[]` are used internally to encode the totally antisymmetric tensor and the Euclidean metric, so they should not used in the model definition.

- `onelooporder` is a dummy variable to identify the one-loop order contribution.

- `sSS` is a dummy variable to identify the order in external momenta of a specific contribution.

- `iCPV`$= \epsilon_{0123}$ is used to fix the sign convention for the Levi-Civita symbol.

# 4 Matchmakereft **usage**

An updated version of this manual can be found, once `matchmakereft` is installed, in the directory `matchmakereft-location/matchmaker/docs/` where `matchmakereft-location` is the directory listed under `Location` when the command `pip show matchmakereft` is used or the analogous location in Anaconda (see Appendix A).

Matchmakereft can be run in two different ways. The same commands are available in all different running modes although the syntax is slightly different on each of them.

## 4.1 Matchmakereft **commmand line interface**

The most straight-forward way to run `matchmakereft` is via the command line interface (CLI). This is obtained by just typing on the terminal (after `matchmakereft` has been installed, see Appendix A for details)

```
> matchmakereft
```

The user has then access to the CLI which looks as follows

```
Welcome to matchmakereft
Please refer to arXiv:2112.10787 when using this code

matchmakereft>
```

Inside the CLI tab-completion is available and all file paths can be absolute or relative. The command `help` gives information on all available commands. The core commands in `matchmakereft` CLI are:

```
matchmakereft> test_installation
```

This command runs a number of minimal tests to check that `matchmakereft` has been correctly installed. The process is verbose and provides information on what is being computed. It takes about 6 minutes to complete in a core-i7@3.00 GHz laptop.

```
matchmakereft> copy_models Location
```

This command copies a number of sample models, including the complete baryon-number conserving SMEFT at dimension 6, in the directory `Location` (which can be `.` for the current directory).

```
matchmakereft> create_model modfile1.fr ... modfilen.fr
```

This command creates a `matchmakereft` model called `modfilen_MM` from the `FeynRules` model defined in one or more files with names `modfile1.fr ... modfilen.fr` as described in detail in Section 3. The `matchmakereft` model is created in the same directory `modfilen.fr` is present. Both relative and absolute paths can be given as input. All models provided with the distribution, that can be obtained via the `copy_models` command, that are extensions of the SM require calling also the file `UnbrokenSM_BFM.fr` as the first file in when using this command. See Section 6.2 for a specific example.

```
matchmakereft> match_model_to_eft UVModelName EFTModelName
```

This command performs the complete tree-level and one-loop matching of a `matchmakereft` UV model with name `UVModelName` onto a `matchmakereft` EFT model with name `EFTModelName`. The result of the matching is written in a file called `MatchingResult.dat` under file `UVModelName`. Any possible problems with the matching are reported and stored in a file called `MatchingProblems.dat` under the same directory. `Matchmakereft` automatically checks if the matching is run in `RGEmaker` or `Matching` mode.

The result of the matching stored in `MatchingResult.dat` is a mathematica list called `MatchingResult` with four entries and the following structure

```
1 MatchingResult ={
2 {
3 {{GreenTree , GreenTreeProblems },{ GreenLoop , GreenLoopProblems }} ,
4 {{ NormGreenTree , NormGreenTreeProblems },{ NormGreenLoop ,
      NormGreenLoopProblems }} ,
5 { PhysTreeLoop },
6 { GaugeCouplingMatching }
7 }
```

where `GreenTree` and `GreenLoop` stands for the tree level and one-loop matching in the Green basis, respectively and `GreenTreeProblems,GreenLoopProblems` are filled if problems were found in the process of impossing hermiticity as discussed in Section 2.3. The second level, with the `Norm` prefix stand for the matching in the Green basis (again separately for tree level and one-loop) after canonical normalization. The third level, denoted `PhysTreeLoop` stands for the matching in the physical basis in which the tree level and one-loop contributions have been merged into a single expression (with the dummy variable `onelooporder` denoting the one-loop contribution). If no physical basis is defined by providing an empty `modfilen.red` file then the Green basis is used as a physical basis. Finally, the fourth level, denoted `GaugeCouplingMatching` provides the redefinition of the gauge couplings after matching as fixed by the corresponding gauge boson canonical normalization in the background gauge.

The file `MatchingProblems.dat` provides partial information in problems encountered during the process of matching. It currently consist of a list with the following data. The loop order (plus one) at which the problem was encountered, the external particles involved in the amplitude that was not completely matched, and the result of the amplitudes that were not properly matched. Non-vanishing values of the amplitudes indicate contributions that the EFT is not able to match. In the current format, all gauge information is lost and only kinematic information is retained. This gives only partial information (we plan to provide more detailed information in future versions of `matchmakereft`) but it can still be useful to debug possible problems. As an example, it is common that models that involve couplings proportional to $\gamma^5$ report some problems, due to the ambiguous terms in the anti-commuting $\gamma^5$ scheme that `matchmakereft` uses. In this case all non-vanishing amplitudes are proportional to `ee[]` (see more details in Section 6.2 and Appendix B.

```
matchmakereft > match_model_to_eft_onlytree  UVModelName  EFTModelName
```

Identical to `match_model_to_eft` but only the tree level matching is computed. With this feature, `matchmakereft` can be used as an automated basis translator, as one can simply use the corresponding EFT in a different basis as UV model and the matching will provide the complete translation between the two bases (see Section 6 for an explicit example).

```
matchmakereft > compute_rge_model_to_eft  UVModelName  EFTModelName
```

This command runs `match_model_to_eft UVModelName EFTModelName` in `RGEmaker` mode (the UV model has to be a light model) and then computes the beta functions for the WCs of the EFT given the UV model. They are stored in a file called `RGEResult.dat` under

directory `UVModelName`. We define the beta function of a WC $C$ as

$$\beta(C) = \mu \frac{\mathrm{d}C}{\mathrm{d}\mu}. \tag{16}$$

```
matchmakereft> clean_model ModelName
```

Matchmakereft is designed for the maximal efficiency so that if a specific process has been already computed it is not computed again. If for any reason the user wants to recreate the calculation of all the amplitudes this command should be invoked to clean the previous calculations.

```
matchmakereft> check_linear_dependence EFTModelName
```

Given a set of operators, defined as an EFT model in directory `EFTModelName`, this command checks if they are off-shell linearly independent or not. This command is useful when finding a Green basis as sometimes the off-shell relations between different operators are difficult to obtain analytically. If the set is not linearly independent `matchmakereft` will provide the relations between the different WCs (see Section 6 for an explicit example).

```
matchmakereft> exit
```

This command exits the CLI.

For the sake of flexibility the following commands are also available to perform independently some of the steps of the calculations:

```
matchmakereft> match_model_to_eft_amplitudes UVModelName EFTModelName
```

This command is used to compute all the relevant amplitudes in the UV model and the EFT but no calculation of the WCs is attempted.

```
matchmakereft> match_model_to_eft_amplitudes_onlytree UVModelName
    EFTModelName
```

This command is identical to `match_model_to_eft_amplitudes` but performs only the tree level calculation.

```
matchmakereft> compute_wilson_coefficients UVModelName EFTModelName
```

This command should be run after the call to either `match_model_to_eft_amplitudes` or `match_model_to_eft_amplitudes_onlytree` and it computes the WCs to complete the matching.

## 4.2 Matchmakereft as a Python module

The python CLI described in the previous section provides an interactive experience to the user. However, `matchmakereft` can be also run by importing `matchmakereft` into a python script, a iPython shell or a Jupyter notebook as a module and then running the same commands as in the CLI adding the parameters of the corresponding function as a string. As an example, the commands to create a model stored in model `UVmodel.fr` and to match it to the EFT stored in directory `EFT_MM` (which we assume has been already created) are given e.g. by

```
1 from matchmakereft.libs.mm_offline import *
2 create_model("UVmodel.fr")
3 match_model_to_eft("UVmodel_MM EFT_MM")
```

All other commands in the CLI are also available to use as functions in a script that imports `matchmakereft`.

# 5  Troubleshooting in `matchmakereft`

`Matchmakereft` provides a significant number of cross-checks that usually catch problems with the installation or with the definition of the models. When a problem is encountered, `matchmakereft` tries to provide a useful warning message that can be used to figure out the origin of the problem. If the user encounters a problem that cannot be solved from the information provided by `matchmakereft` we encourage them to check the troubleshooting section in the latest `matchmakereft` manual and the Gitlab `matchmakereft` issue tracker (https://gitlab.com/m4103/matchmaker-eft/-/issues) to see if the problem has been encountered by other users and a solution is available. If no solution can be found, the issue tracker should be use to pose questions to the `matchmakereft` developers or to file possible bugs.

Most of the times an unsuccessful matching is due to a badly defined model. Some common pitfalls are:

- The complete Lagrangian of the model has to be named `Ltot`. Using a different name results in `matchmakereft` not creating the model properly.

- Operators badly defined in `FeynRules` (a common example is indices not properly contracted). This results in wrong Feynman rules that lead to incorrect matching.

- Model generation takes too long. This can happen with complicated models, in particular with effective operators of high mass dimension. As an example, the generation of the SMEFT model can easily take more than 30 minutes in a core-i7@3.00 GHz laptop. In this case it is useful to compute the Feynman rules directly with `FeynRules` to check that the model does not have any obvious problems. Also sometimes expanding in the gauge components can significantly speed up model creation (at the expense of a reduced gauge degeneracy and therefore a smaller set of cross-checks).

- `QGRAF` not running correctly. This could happen when vertices with a larger number of particles than the limit set in `QGRAF` are present. The solution is to modify correspondingly the limit in the `QGRAF` source and compile it again.

- `FORM` not running correctly. This is normally due to variables not being correctly defined (again due to an incorrect implementation of the model). Running directly with `FORM` the offending file can give hints on what is happening.

- `FORM` taking too long to run. Amplitudes with many external legs usually involve a very large number of diagrams that can take a long time to compute. The simplest solution is to not include the corresponding operators in the EFT model if the user is not interested in their matching (in the SMEFT case at dimension 6 the operator $\mathcal{O}_H = (H^\dagger H)^3$ is usually the one that takes the longest to be matched). Other options could imply splitting the diagrams into smaller sets and combining the result after the calculation (this is how the beta function of the $(H^\dagger H)^4$ operator was computed in [24]) but this process is not straight-forward. We expect to automate this procedure in future versions of `matchmakereft`.

- All amplitudes are computed but the matching is unsuccessful. This can be due to a number of reasons, the most common ones being: the WCs of the EFT model or the couplings in the UV model have some symmetry properties that have not been implemented in the corresponding `modfilen.symm` file; the hermiticity properties of the couplings in the EFT or UV models have not been properly defined, either in the definition of the model itself or in the corresponding `modfilen.herm` file; there are some missing operators in the Green basis of the EFT model.

# 6 Physics applications

A preliminary version of `matchmakereft` has been already used in a number of physical applications [24, 68–73] but we list here some of the new applications that `matchmakereft` has.

## 6.1 Cross-checks

As we have emphasized, the large redundancy inherent in the off-shell matching in the BFM gives us confidence on the correctness of the results computed with `matchmakereft`. Nevertheless we have tested `matchmakereft` against some of the few available complete one-loop matching results in the literature. We have found complete agreement except when explicitly described. The list of models we have compared to include:

- `RGEmaker` mode:

  – Complete RGEs for the ALP-SMEFT up to mass dimension-5 as computed in [68]. Exact agreement was found up to a typo in the original reference.

  – RGEs for the purely bosonic and two-fermion operators in the Warsaw basis [74] as computed in [15–17] and implemented in `DSixTools` [29, 30]. Complete agreement was found.

- `Matching` mode:

  – Scalar singlet. The complete matching up to one-loop order of an extension of the SM with a scalar singlet was recently completed in [75], after several partial attempts [37, 76]. We have found complete agreement with the results in [75].

  – Type-I see-saw model, as computed in [77]. Complete agreement was found.

  – Scalar leptoquarks, as computed in [65]. We have found some minor differences that we are discussing with the authors.

  – Charged scalar electroweak singlet, as computed in [78]. We agree with the result except for a sign in Eqs. (4.14), the terms with Pauli matrices in (4.15), (B.4) and (B.5) (the latter is the culprit of the opposite sign in terms with Pauli matrices) and a factor of 2 in Eq. (4.17) and of 4 in (B.7). We have contacted the authors about these differences.

  – Some partial cross-checks have been performed against the results in [79] with full agreement.

## 6.2 Complete one-loop matching of a new charged vector-like lepton singlet

In this section we provide the complete tree-level and one-loop matching of an extension of the SM with a new hypercharge $-1$, electroweak singlet vector-like lepton $E$. The Lagrangian is given by

$$\mathcal{L} = \mathcal{L}_{\text{SM}} + \bar{E}(\mathrm{i}\not{D} - M_E)E - \left[\tilde{\lambda}_i \bar{\ell}_i \phi E_R + \text{h.c.}\right], \tag{17}$$

where $\ell_i$ and $\phi$ stand for the SM lepton doublets and the Higgs boson, respectively and $i$ is a SM flavour index. See [80] for direct experimental limits on such an extension of the SM.

This model is included in the distribution of `matchmakereft` and can be obtained via the `copy_models` command. Once the model is downloaded, and inside the corresponding directory, the following commands will generate the complete one-loop matching, including the complete matching in the Green basis. We use the CLI as an example and replace the output given by `matchmakereft` with ... ,

```
matchmakereft> create_model UnbrokenSM_BFM.fr VLL_Singlet_Y_m1_BFM.fr
...
matchmakereft> match_model_to_eft VLL_Singlet_Y_m1_BFM_MM
    SMEFT_Green_Bpreserving_MM
...
```

Note that, due to the presence of $\gamma^5$ `matchmakereft` will warn the user that some problems are present and they are reported in the `MatchingProblems.dat` file. A close look at this file will show that all the non-vanishing amplitudes are proportional to the `ee[]` symbol and therefore they are indeed related to the $\gamma^5$ issue. As we argue in Appendix B, the procedure followed by `matchmakereft` guarantees the correct result in the SMEFT. Nevertheless we prefer `matchmakereft` to give the warning to alert the user of the $\gamma^5$ issue and the solution adopted.

The non vanishing WCs in the Warsaw basis, including one-loop accuracy are given in the next sections. In order to reduce clutter we only write explicitly flavour indices when necessary. Also we use the following notation

$$\lfloor \tilde{\lambda} \tilde{\lambda}^* \rfloor \equiv \tilde{\lambda}_i \tilde{\lambda}_i^*, \qquad \lfloor \tilde{\lambda} \mathcal{M} \tilde{\lambda}^* \rfloor \equiv \tilde{\lambda}_i \mathcal{M}_{ij} \tilde{\lambda}_j^*, \tag{18}$$

with $\mathcal{M}_{ij}$ an arbitrary matrix with flavour indices. We also define

$$L_E \equiv \log(\mu^2/M_E^2). \tag{19}$$

The tree level result agrees with the calculation in [34] (when taking into account the different notation in the Yukawa coupling).

### 6.2.1 SM couplings

The SM couplings receive the following (one-loop) corrections:

$$\mu_H^2 = \mu_H^{(0)2} + \frac{\lfloor \tilde{\lambda}^* \tilde{\lambda} \rfloor}{16\pi^2} \left[ 2M_E^2 - \frac{1}{2}\mu_H^{(0)2} - \frac{1}{3}\frac{\mu_H^{(0)4}}{M_E^2} - (\mu_H^{(0)2} - 2M_E^2)L_E \right], \tag{20}$$

$$\lambda = \lambda^{(0)} + \frac{1}{16\pi^2} \left[ \frac{\lfloor \tilde{\lambda}^* \tilde{\lambda} \rfloor (5g_2^{(0)2}\mu_H^{(0)2} - 6\lambda^{(0)}(4\mu_H^{(0)2} + 3M_E^2) + 18\mu_H^{(0)2}\lfloor \tilde{\lambda}^* \tilde{\lambda} \rfloor)}{18M_E^2} \right.$$
$$\left. + \frac{(2M_E^2 - \mu_H^{(0)2})\lfloor \tilde{\lambda}^* Y_e^{(0)} Y_e^{(0)\dagger} \tilde{\lambda} \rfloor}{M_E^2} \right] \tag{21}$$

$$- \frac{1}{16\pi^2} \left[ \frac{\lfloor \tilde{\lambda}^* \tilde{\lambda} \rfloor [-g_2^{(0)2}\mu_H^{(0)2} + 6\lambda^{(0)}M_E^2 - 3M_E^2\lfloor \tilde{\lambda}^* \tilde{\lambda} \rfloor]}{3M_E^2} \right.$$
$$\left. + \frac{2(-M_E^2 + \mu_H^{(0)2})\lfloor \tilde{\lambda}^* Y_e^{(0)} Y_e^{(0)\dagger} \tilde{\lambda} \rfloor}{M_E^2} \right] L_E, \tag{22}$$

$$Y_u = Y_u^{(0)} - \frac{1}{16\pi^2} \left[ \left( \frac{1}{4} + \frac{\mu_H^{(0)2}}{3M_E^2} \right) Y_u^{(0)} \lfloor \tilde{\lambda}^* \tilde{\lambda} \rfloor + \frac{1}{2}Y_u^{(0)}\lfloor \tilde{\lambda}^* \tilde{\lambda} \rfloor L_E \right], \tag{23}$$

$$Y_d = Y_d^{(0)} - \frac{1}{16\pi^2} \left[ \left( \frac{1}{4} + \frac{\mu_H^{(0)2}}{3M_E^2} \right) Y_d^{(0)} \lfloor \tilde{\lambda}^* \tilde{\lambda} \rfloor + \frac{1}{2}Y_d^{(0)}\lfloor \tilde{\lambda}^* \tilde{\lambda} \rfloor L_E \right], \tag{24}$$

$$(Y_e)_{ij} = (Y_e^{(0)})_{ij} - \frac{1}{16\pi^2} \left[ \left( \frac{1}{4} + \frac{\mu_H^{(0)2}}{3M_E^2} \right) (Y_e^{(0)})_{ij} \lfloor \tilde{\lambda}^* \tilde{\lambda} \rfloor + \left( \frac{3}{8} + \frac{3\mu_H^{(0)2}}{4M_E^2} \right) \tilde{\lambda}_i \tilde{\lambda}_k^* (Y_e^{(0)})_{kj} \right]$$
$$- \frac{1}{16\pi^2} \left[ \frac{1}{2}(Y_e^{(0)})_{ij} \lfloor \tilde{\lambda}^* \tilde{\lambda} \rfloor + \left( \frac{1}{4} + \frac{\mu_H^{(0)2}}{2M_E^2} \right) \tilde{\lambda}_i \tilde{\lambda}_k^* (Y_e^{(0)})_{kj} \right] L_E, \tag{25}$$

where the (0) superscript denotes the original parameters in the SM Lagrangian. All other SM couplings receive no corrections. In the following we express our results in terms of the physical SM couplings, the ones on the left hand side of Eqs. (20-25).

### 6.2.2  Bosonic operators

Turning now to dimension 6 bosonic operators, we obtain the following non-vanishing WCs.

$$\alpha_{HW} = \frac{1}{16\pi^2} \frac{g_2^2 \lfloor \tilde{\lambda}\tilde{\lambda}^* \rfloor}{24 M_E^2}, \tag{26}$$

$$\alpha_{HB} = \frac{1}{16\pi^2} \frac{g_1^2 \lfloor \tilde{\lambda}\tilde{\lambda}^* \rfloor}{8 M_E^2}, \tag{27}$$

$$\alpha_{HWB} = -\frac{1}{16\pi^2} \frac{g_1 g_2 \lfloor \tilde{\lambda}\tilde{\lambda}^* \rfloor}{6 M_E^2}, \tag{28}$$

$$\alpha_{H\Box} = \frac{1}{16\pi^2} \frac{1}{M_E^2} \left[ -\frac{g_1^4}{30} + \left( \frac{13g_1^2}{72} - \frac{5g_2^2}{24} - \frac{\lfloor \tilde{\lambda}\tilde{\lambda}^* \rfloor}{3} \right) \lfloor \tilde{\lambda}\tilde{\lambda}^* \rfloor + \frac{3}{2} \lfloor \tilde{\lambda}^* Y_e Y_e^\dagger \tilde{\lambda} \rfloor \right]$$
$$+ \frac{1}{16\pi^2} \frac{1}{M_E^2} \left[ \left( \frac{g_1^2}{12} - \frac{g_2^2}{4} \right) \lfloor \tilde{\lambda}\tilde{\lambda}^* \rfloor + \lfloor \tilde{\lambda}^* Y_e Y_e^\dagger \tilde{\lambda} \rfloor \right] L_E, \tag{29}$$

$$\alpha_{HD} = \frac{1}{16\pi^2} \frac{1}{M_E^2} \left[ -\frac{2g_1^4}{15} + \left( \frac{13g_1^2}{18} - \frac{\lfloor \tilde{\lambda}\tilde{\lambda}^* \rfloor}{2} \right) \lfloor \tilde{\lambda}\tilde{\lambda}^* \rfloor + \frac{1}{2} \lfloor \tilde{\lambda}^* Y_e Y_e^\dagger \tilde{\lambda} \rfloor \right]$$
$$+ \frac{1}{16\pi^2} \frac{1}{M_E^2} \left[ \frac{g_1^2}{3} \lfloor \tilde{\lambda}\tilde{\lambda}^* \rfloor + \lfloor \tilde{\lambda}^* Y_e Y_e^\dagger \tilde{\lambda} \rfloor \right] L_E, \tag{30}$$

$$\alpha_H = \frac{1}{16\pi^2} \frac{1}{M_E^2} \left[ \left( \frac{4\lambda^2}{3} - \frac{5\lambda g_2^2}{9} - 2\lambda \lfloor \tilde{\lambda}\tilde{\lambda}^* \rfloor + \frac{\lfloor \tilde{\lambda}\tilde{\lambda}^* \rfloor \lfloor \tilde{\lambda}\tilde{\lambda}^* \rfloor}{3} + 2\lfloor \tilde{\lambda}^* Y_e Y_e^\dagger \tilde{\lambda} \rfloor \right) \lfloor \tilde{\lambda}\tilde{\lambda}^* \rfloor \right.$$
$$\left. + 2\lambda \lfloor \tilde{\lambda}^* Y_e Y_e^\dagger \tilde{\lambda} \rfloor - 2\lfloor \tilde{\lambda}^* Y_e Y_e^\dagger Y_e Y_e^\dagger \tilde{\lambda} \rfloor \right]$$
$$+ \frac{1}{16\pi^2} \frac{1}{M_E^2} \left[ -\frac{2\lambda g_2^2}{3} \lfloor \tilde{\lambda}\tilde{\lambda}^* \rfloor + 4\lambda \lfloor \tilde{\lambda}^* Y_e Y_e^\dagger \tilde{\lambda} \rfloor - 2\lfloor \tilde{\lambda}^* Y_e Y_e^\dagger Y_e Y_e^\dagger \tilde{\lambda} \rfloor \right] L_E. \tag{31}$$

All other bosonic operators do not receive any corrections up to one loop.

### 6.2.3  Bi-fermion operators

Regarding operators in the Warsaw basis with two fermion fields, the non-vanishing contributions in our model are the following.

$$(\alpha_{eW})_{ij} = -\frac{1}{16\pi^2} \frac{g_2}{24 M_E^2} \tilde{\lambda}_i \tilde{\lambda}_k^* (Y_e)_{kj}, \tag{32}$$

$$(\alpha_{eB})_{ij} = -\frac{1}{16\pi^2} \frac{g_1}{12 M_E^2} \tilde{\lambda}_i \tilde{\lambda}_k^* (Y_e)_{kj}, \tag{33}$$

$$(\alpha_{Hq}^{(1)})_{ij} = \frac{g_1^2}{16\pi^2} \frac{1}{M_E^2} \left[ -\frac{g_1^2}{45} + \frac{13}{216} \lfloor \tilde{\lambda}\tilde{\lambda}^* \rfloor + \frac{1}{36} \lfloor \tilde{\lambda}\tilde{\lambda}^* \rfloor L_E \right] \delta_{ij}, \tag{34}$$

$$(\alpha_{Hq}^{(3)})_{ij} = \frac{g_2^2}{16\pi^2} \frac{1}{M_E^2} \left[ -\frac{5}{72} \lfloor \tilde{\lambda}\tilde{\lambda}^* \rfloor - \frac{1}{12} \lfloor \tilde{\lambda}\tilde{\lambda}^* \rfloor L_E \right] \delta_{ij}, \tag{35}$$

$$(\alpha_{Hu})_{ij} = \frac{g_1^2}{16\pi^2} \frac{1}{M_E^2} \left[ -\frac{4g_1^2}{45} + \frac{13}{54} \lfloor \tilde{\lambda}\tilde{\lambda}^* \rfloor + \frac{1}{9} \lfloor \tilde{\lambda}\tilde{\lambda}^* \rfloor L_E \right] \delta_{ij}, \tag{36}$$

$$(\alpha_{Hd})_{ij} = \frac{g_1^2}{16\pi^2}\frac{1}{M_E^2}\left[\frac{2g_1^2}{45} - \frac{13}{108}\lfloor\tilde{\lambda}\tilde{\lambda}^*\rfloor - \frac{1}{18}\lfloor\tilde{\lambda}\tilde{\lambda}^*\rfloor L_E\right]\delta_{ij}, \tag{37}$$

$$(\alpha_{H\ell}^{(1)})_{ij} = -\frac{\tilde{\lambda}_i\tilde{\lambda}_j^*}{4M_E^2} + \frac{1}{16\pi^2}\frac{1}{M_E^2}\left[\left(\frac{g_1^4}{15} - \frac{13g_1^2\lfloor\tilde{\lambda}\tilde{\lambda}^*\rfloor}{72}\right)\delta_{ij} + \left(\frac{31g_1^2}{288} - \frac{33g_2^2}{32} + \frac{13}{16}\lfloor\tilde{\lambda}\tilde{\lambda}^*\rfloor\right)\tilde{\lambda}_i\tilde{\lambda}_j^*\right.$$

$$\left. - \frac{1}{2}\tilde{\lambda}_i\tilde{\lambda}_k^*(Y_e)_{kl}(Y_e^\dagger)_{lj} - \frac{1}{2}(Y_e)_{ik}(Y_e^\dagger)_{kl}\tilde{\lambda}_l\tilde{\lambda}_j^*\right]$$

$$+ \frac{1}{16\pi^2}\frac{1}{M_E^2}\left[-\frac{g_1^2\lfloor\tilde{\lambda}\tilde{\lambda}^*\rfloor}{12}\delta_{ij} + \left(\frac{25g_1^2}{48} - \frac{9g_2^2}{16} + \frac{3\lfloor\tilde{\lambda}\tilde{\lambda}^*\rfloor}{8}\right)\tilde{\lambda}_i\tilde{\lambda}_j^*\right.$$

$$\left. - \frac{1}{2}\tilde{\lambda}_i\tilde{\lambda}_k^*(Y_e)_{kl}(Y_e^\dagger)_{lj} - \frac{1}{2}(Y_e)_{ik}(Y_e^\dagger)_{kl}\tilde{\lambda}_l\tilde{\lambda}_j^*\right]L_E, \tag{38}$$

$$(\alpha_{H\ell}^{(3)})_{ij} = -\frac{\tilde{\lambda}_i\tilde{\lambda}_j^*}{4M_E^2} + \frac{1}{16\pi^2}\frac{1}{M_E^2}\left[-\frac{5g_2^2\lfloor\tilde{\lambda}\tilde{\lambda}^*\rfloor}{72}\delta_{ij} + \left(\frac{9g_1^2}{32} + \frac{77g_2^2}{288} + \frac{5}{16}\lfloor\tilde{\lambda}\tilde{\lambda}^*\rfloor\right)\tilde{\lambda}_i\tilde{\lambda}_j^*\right]$$

$$+ \frac{1}{16\pi^2}\frac{1}{M_E^2}\left[-\frac{g_2^2\lfloor\tilde{\lambda}\tilde{\lambda}^*\rfloor}{12}\delta_{ij} + \left(\frac{9g_1^2}{16} + \frac{7g_2^2}{48} + \frac{3\lfloor\tilde{\lambda}\tilde{\lambda}^*\rfloor}{8}\right)\tilde{\lambda}_i\tilde{\lambda}_j^*\right]L_E, \tag{39}$$

$$(\alpha_{He})_{ij} = \frac{1}{16\pi^2}\frac{1}{M_E^2}\left[g_1^2\left(\frac{2g_1^2}{15} - \frac{13\lfloor\tilde{\lambda}\tilde{\lambda}^*\rfloor}{36}\right)\delta_{ij} + \frac{1}{24}(Y_e^\dagger)_{ik}\tilde{\lambda}_k\tilde{\lambda}_l^*(Y_e)_{lj}\right.$$

$$\left. + \left(-\frac{g_1^2}{6}\lfloor\tilde{\lambda}\tilde{\lambda}^*\rfloor\delta_{ij} + \frac{1}{4}(Y_e^\dagger)_{ik}\tilde{\lambda}_k\tilde{\lambda}_l^*(Y_e)_{lj}\right)L_E\right], \tag{40}$$

$$(\alpha_{uH})_{ij} = \frac{1}{16\pi^2}\frac{1}{M_E^2}\left[\left(\frac{2}{3}\lambda - \frac{5g_2^2}{36} - \frac{1}{2}\lfloor\tilde{\lambda}\tilde{\lambda}^*\rfloor\right)\lfloor\tilde{\lambda}\tilde{\lambda}^*\rfloor + \frac{1}{2}\lfloor\tilde{\lambda}^*Y_eY_e^\dagger\tilde{\lambda}\rfloor\right.$$

$$\left. + \left(-\frac{g_2^2}{6}\lfloor\tilde{\lambda}\tilde{\lambda}^*\rfloor + \lfloor\tilde{\lambda}^*Y_eY_e^\dagger\tilde{\lambda}\rfloor\right)L_E\right](Y_u)_{ij}, \tag{41}$$

$$(\alpha_{dH})_{ij} = \frac{1}{16\pi^2}\frac{1}{M_E^2}\left[\left(\frac{2}{3}\lambda - \frac{5g_2^2}{36} - \frac{1}{2}\lfloor\tilde{\lambda}\tilde{\lambda}^*\rfloor\right)\lfloor\tilde{\lambda}\tilde{\lambda}^*\rfloor + \frac{1}{2}\lfloor\tilde{\lambda}^*Y_eY_e^\dagger\tilde{\lambda}\rfloor\right.$$

$$\left. + \left(-\frac{g_2^2}{6}\lfloor\tilde{\lambda}\tilde{\lambda}^*\rfloor + \lfloor\tilde{\lambda}^*Y_eY_e^\dagger\tilde{\lambda}\rfloor\right)L_E\right](Y_d)_{ij}, \tag{42}$$

$$(\alpha_{eH})_{ij} = \frac{\tilde{\lambda}_i\tilde{\lambda}_k^*(Y_e)_{kj}}{2M_E^2} + \frac{1}{16\pi^2}\frac{1}{M_E^2}\left[\left(\frac{2}{3}\lambda - \frac{5g_2^2}{36} - \frac{1}{2}\lfloor\tilde{\lambda}\tilde{\lambda}^*\rfloor\right)\lfloor\tilde{\lambda}\tilde{\lambda}^*\rfloor(Y_e)_{ij} + \frac{1}{2}\lfloor\tilde{\lambda}^*Y_eY_e^\dagger\tilde{\lambda}\rfloor(Y_e)_{ij}\right.$$

$$\left. + \left(5\lambda - \frac{14}{16}\lfloor\tilde{\lambda}\tilde{\lambda}^*\rfloor\right)\tilde{\lambda}_i\tilde{\lambda}_k^*(Y_e)_{kj} + \frac{37}{24}(Y_e)_{ik}(Y_e^\dagger)_{kl}\tilde{\lambda}_l\tilde{\lambda}_m^*(Y_e)_{mj} - \frac{1}{4}\tilde{\lambda}_i\tilde{\lambda}_k^*(Y_e)_{kl}(Y_e^\dagger)_{lm}(Y_e)_{mj}\right]$$

$$+ \frac{1}{16\pi^2}\frac{1}{M_E^2}\left[-\frac{g_2^2}{6}\lfloor\tilde{\lambda}\tilde{\lambda}^*\rfloor(Y_e)_{ij} + \lfloor\tilde{\lambda}^*Y_eY_e^\dagger\tilde{\lambda}\rfloor(Y_e)_{ij} + \left(4\lambda - \frac{3}{4}\lfloor\tilde{\lambda}\tilde{\lambda}^*\rfloor\right)\tilde{\lambda}_i\tilde{\lambda}_k^*(Y_e)_{kj}\right.$$

$$\left. + \frac{3}{4}(Y_e)_{ik}(Y_e^\dagger)_{kl}\tilde{\lambda}_l\tilde{\lambda}_m^*(Y_e)_{mj}\right]L_E. \tag{43}$$

### 6.2.4 Four-fermion operators

Finally, the following four-fermion operators receive non-vanishing WCs.

$$(\alpha_{qq}^{(1)})_{ijkl} = -\frac{1}{16\pi^2}\frac{g_1^4}{270M_E^2}\delta_{ij}\delta_{kl}\,, \tag{44}$$

$$(\alpha_{uu})_{ijkl} = -\frac{1}{16\pi^2}\frac{8g_1^4}{135M_E^2}\delta_{ij}\delta_{kl}\,, \tag{45}$$

$$(\alpha_{dd})_{ijkl} = -\frac{1}{16\pi^2}\frac{2g_1^4}{135M_E^2}\delta_{ij}\delta_{kl}\,, \tag{46}$$

$$(\alpha_{ud}^{(1)})_{ijkl} = -\frac{1}{16\pi^2}\frac{8g_1^4}{135M_E^2}\delta_{ij}\delta_{kl}\,, \tag{47}$$

$$(\alpha_{qu}^{(1)})_{ijkl} = -\frac{1}{16\pi^2}\frac{1}{M_E^2}\left[\frac{4g_1^4}{135}\delta_{ij}\delta_{kl} + \frac{1}{18}\lfloor\tilde{\lambda}\tilde{\lambda}^*\rfloor(Y_u)_{il}(Y_u^\dagger)_{kj}\right]\,, \tag{48}$$

$$(\alpha_{qu}^{(8)})_{ijkl} = -\frac{1}{16\pi^2}\frac{1}{3M_E^2}\lfloor\tilde{\lambda}\tilde{\lambda}^*\rfloor(Y_u)_{il}(Y_u^\dagger)_{kj}\,, \tag{49}$$

$$(\alpha_{qd}^{(1)})_{ijkl} = \frac{1}{16\pi^2}\frac{1}{M_E^2}\left[\frac{2g_1^4}{135}\delta_{ij}\delta_{kl} - \frac{1}{18}\lfloor\tilde{\lambda}\tilde{\lambda}^*\rfloor(Y_d)_{il}(Y_d^\dagger)_{kj}\right]\,, \tag{50}$$

$$(\alpha_{qd}^{(8)})_{ijkl} = -\frac{1}{16\pi^2}\frac{1}{3M_E^2}\lfloor\tilde{\lambda}\tilde{\lambda}^*\rfloor(Y_d)_{il}(Y_d^\dagger)_{kj}\,, \tag{51}$$

$$(\alpha_{quqd}^{(1)})_{ijkl} = \frac{1}{16\pi^2}\frac{1}{3M_E^2}\lfloor\tilde{\lambda}\tilde{\lambda}^*\rfloor(Y_u)_{ij}(Y_d)_{kl}\,, \tag{52}$$

$$\begin{aligned}
(\alpha_{\ell\ell})_{ijkl} = &\frac{1}{16\pi^2}\frac{1}{M_E^2}\left[-\frac{g_1^4}{30}\delta_{ij}\delta_{kl} + \frac{25g_1^2+11g_2^2}{288}(\delta_{ij}\tilde{\lambda}_k\tilde{\lambda}_l^* + \tilde{\lambda}_i\tilde{\lambda}_j^*\delta_{kl})\right.\\
&\left.-\frac{11g_2^2}{144}(\delta_{il}\tilde{\lambda}_k\tilde{\lambda}_j^* + \delta_{jk}\tilde{\lambda}_i\tilde{\lambda}_l^*) - \frac{1}{8}\tilde{\lambda}_i\tilde{\lambda}_j^*\tilde{\lambda}_k\tilde{\lambda}_l^* + \frac{3}{16}(\tilde{\lambda}_i\tilde{\lambda}_l^*(Y_e)_{km}(Y_e^\dagger)_{mj} + \tilde{\lambda}_k\tilde{\lambda}_j^*(Y_e)_{im}(Y_e^\dagger)_{ml})\right]\\
&+\frac{1}{16\pi^2}\frac{1}{M_E^2}\left[\frac{g_1^2+g_2^2}{48}(\delta_{ij}\tilde{\lambda}_k\tilde{\lambda}_l^* + \tilde{\lambda}_i\tilde{\lambda}_j^*\delta_{kl}) - \frac{g_2^2}{24}(\delta_{il}\tilde{\lambda}_k\tilde{\lambda}_j^* + \delta_{jk}\tilde{\lambda}_i\tilde{\lambda}_l^*)\right.\\
&\left.+\frac{1}{8}(\tilde{\lambda}_i\tilde{\lambda}_l^*(Y_e)_{km}(Y_e^\dagger)_{mj} + \tilde{\lambda}_k\tilde{\lambda}_j^*(Y_e)_{im}(Y_e^\dagger)_{ml})\right]L_E\,, \tag{53}
\end{aligned}$$

$$(\alpha_{ee})_{ijkl} = -\frac{1}{16\pi^2}\frac{2g_1^4}{15M_E^2}\delta_{ij}\delta_{kl}\,, \tag{54}$$

$$\begin{aligned}
(\alpha_{\ell e})_{ijkl} = &\frac{1}{16\pi^2}\frac{1}{M_E^2}\left[-\frac{2g_1^4}{15}\delta_{ij}\delta_{kl} + \frac{25g_1^2}{72}\tilde{\lambda}_i\tilde{\lambda}_j^*\delta_{kl} - \frac{1}{6}(Y_e)_{il}(Y_e^\dagger)_{kj} - \frac{3}{8}\tilde{\lambda}_i\tilde{\lambda}_j^*(Y_e^\dagger)_{km}(Y_e)_{ml}\right]\\
&+\frac{1}{16\pi^2}\frac{1}{M_E^2}\left[+\frac{g_1^2}{12}\tilde{\lambda}_i\tilde{\lambda}_j^*\delta_{kl} - \frac{1}{4}\tilde{\lambda}_i\tilde{\lambda}_j^*(Y_e^\dagger)_{km}(Y_e)_{ml}\right]L_E\,, \tag{55}
\end{aligned}$$

$$\begin{aligned}
(\alpha_{\ell q}^{(1)})_{ijkl} = &\frac{1}{16\pi^2}\frac{1}{M_E^2}\left[\frac{g_1^4}{45}\delta_{ij}\delta_{kl} + \frac{25g_1^2}{432}\tilde{\lambda}_i\tilde{\lambda}_j^*\delta_{kl} + \frac{3}{16}\tilde{\lambda}_i\tilde{\lambda}_j^*\big((Y_d)_{km}(Y_d^\dagger)_{ml} - (Y_u)_{km}(Y_u^\dagger)_{ml}\big)\right]\\
&+\frac{1}{16\pi^2}\frac{1}{M_E^2}\left[-\frac{g_1^2}{72}\tilde{\lambda}_i\tilde{\lambda}_j^*\delta_{kl} + \frac{1}{8}\tilde{\lambda}_i\tilde{\lambda}_j^*\big((Y_d)_{km}(Y_d^\dagger)_{ml} - (Y_u)_{km}(Y_u^\dagger)_{ml}\big)\right]L_E\,, \tag{56}
\end{aligned}$$

$$(\alpha_{\ell q}^{(3)})_{ijkl} = \frac{1}{16\pi^2}\frac{1}{M_E^2}\left[-\frac{11g_2^2}{144}\tilde{\lambda}_i\tilde{\lambda}_j^*\delta_{kl} + \frac{3}{16}\tilde{\lambda}_i\tilde{\lambda}_j^*\big((Y_d)_{km}(Y_d^\dagger)_{ml} + (Y_u)_{km}(Y_u^\dagger)_{ml}\big)\right]$$

$$+\frac{1}{16\pi^2}\frac{1}{M_E^2}\left[-\frac{g_2^2}{24}\tilde{\lambda}_i\tilde{\lambda}_j^*\delta_{kl}+\frac{1}{8}\tilde{\lambda}_i\tilde{\lambda}_j^*\Big((Y_d)_{km}(Y_d^\dagger)_{ml}+(Y_u)_{km}(Y_u^\dagger)_{ml}\Big)\right]L_E\,,\qquad(57)$$

$$(\alpha_{eu})_{ijkl}=\frac{1}{16\pi^2}\frac{8g_1^4}{45M_E^2}\delta_{ij}\delta_{kl}\,,\qquad(58)$$

$$(\alpha_{ed})_{ijkl}=-\frac{1}{16\pi^2}\frac{4g_1^4}{45M_E^2}\delta_{ij}\delta_{kl}\,,\qquad(59)$$

$$(\alpha_{qe})_{ijkl}=\frac{1}{16\pi^2}\frac{2g_1^4}{45M_E^2}\delta_{ij}\delta_{kl}\,,\qquad(60)$$

$$(\alpha_{\ell u})_{ijkl}=\frac{1}{16\pi^2}\frac{1}{M_E^2}\left[\frac{4g_1^4}{45}\delta_{ij}\delta_{kl}-\frac{25g_1^2}{108}\tilde{\lambda}_i\tilde{\lambda}_j^*\delta_{kl}+\frac{3}{8}\tilde{\lambda}_i\tilde{\lambda}_j^*(Y_u^\dagger)_{km}(Y_u)_{ml}\right]$$
$$+\frac{1}{16\pi^2}\frac{1}{M_E^2}\left[-\frac{g_1^2}{18}\tilde{\lambda}_i\tilde{\lambda}_j^*\delta_{kl}+\frac{1}{4}\tilde{\lambda}_i\tilde{\lambda}_j^*(Y_u^\dagger)_{km}(Y_u)_{ml}\right]L_E\,,\qquad(61)$$

$$(\alpha_{\ell d})_{ijkl}=\frac{1}{16\pi^2}\frac{1}{M_E^2}\left[-\frac{2g_1^4}{45}\delta_{ij}\delta_{kl}+\frac{25g_1^2}{216}\tilde{\lambda}_i\tilde{\lambda}_j^*\delta_{kl}-\frac{3}{8}\tilde{\lambda}_i\tilde{\lambda}_j^*(Y_d^\dagger)_{km}(Y_d)_{ml}\right]$$
$$+\frac{1}{16\pi^2}\frac{1}{M_E^2}\left[\frac{g_1^2}{36}\tilde{\lambda}_i\tilde{\lambda}_j^*\delta_{kl}-\frac{1}{4}\tilde{\lambda}_i\tilde{\lambda}_j^*(Y_d^\dagger)_{km}(Y_d)_{ml}\right]L_E\,,\qquad(62)$$

$$(\alpha_{\ell edq})_{ijkl}=\frac{1}{16\pi^2}\frac{1}{3M_E^2}\lfloor\tilde{\lambda}\tilde{\lambda}^*\rfloor(Y_e)_{ij}(Y_d^\dagger)_{kl}\,,\qquad(63)$$

$$(\alpha_{\ell equ}^{(1)})_{ijkl}=-\frac{1}{16\pi^2}\frac{1}{3M_E^2}\lfloor\tilde{\lambda}\tilde{\lambda}^*\rfloor(Y_e)_{ij}(Y_u)_{kl}\,.\qquad(64)$$

## 6.3 Basis translation

Given a specific EFT, defined by its field content and symmetries, there are different options to choose a basis of operators (either Green or physical). Different bases are useful for different purposes and it is very useful to have a systematic way to translate the results from one basis to another. The `Rosetta` code [81] can be used to translate between popular physical bases for the dimension 6 SMEFT. However a more general approach, applicable to different EFTs and any two bases (not necessarily physical) would be very welcome. `Matchmakereft` can do this in a straight-forward way by performing the tree-level matching of the new basis (as a UV model) onto the old one (as an EFT).

As a trivial example we consider the following two operators that appear in the one-loop integration of a scalar singlet as shown in [76]

$$O_R=(H^\dagger H)(D_\mu H^\dagger D^\mu H)$$
$$\rightarrow 2\lambda\mathcal{O}_H+\frac{1}{2}\mathcal{O}_{H\Box}+\frac{1}{2}\Big((Y_u)_{ij}(\mathcal{O}_{uH})_{ij}+(Y_d)_{ij}(\mathcal{O}_{dH})_{ij}+(Y_e)_{ij}(\mathcal{O}_{eH})_{ij}+\text{h.c.}\Big)\,,\qquad(65)$$
$$O_T=\frac{1}{2}(H^\dagger\overleftrightarrow{D}_\mu H)^2\rightarrow-2\mathcal{O}_{HD}-\frac{1}{2}\mathcal{O}_{H\Box}\,,\qquad(66)$$

where in the second equality we have written these operators in terms of the corresponding ones in the Warsaw basis (see Appendix D for the definition of the operators).

When using `matchmakereft` to match at tree level a UV model consisting of the SM plus the two operators in the new basis, $O_R$ and $O_T$, onto the SMEFT in the basis described in Appendix D we obtain the following tree-level matching in the physical basis (we use $\beta$ for the WCs of the operators in the new basis)

$$\alpha_H = 2\lambda\beta_R, \qquad \alpha_{HD} = -2\beta_T, \qquad \alpha_{H\square} = \frac{1}{2}(\beta_R - \beta_T), \qquad (67)$$

$$\alpha_{uH} = \frac{1}{2}\beta_R(Y_u)_{ij}, \qquad \alpha_{dH} = \frac{1}{2}\beta_R(Y_d)_{ij}, \qquad \alpha_{eH} = \frac{1}{2}\beta_R(Y_e)_{ij}, \qquad (68)$$

which exactly reproduce the above equations. This is of course just a minimal example to show the application of `matchmakereft` to basis translation but complete bases (both Green and physical) can be translated in an automated way using this procedure.

### 6.4 Off-shell operator independence

Constructing a Green basis, that can match arbitrary off-shell amplitudes, is in principle straightforward. One has to simply write all possible operators and then eliminate those that are related to others by integration by parts (momentum conservation at the amplitude level) or gauge (Fierz) identities. However, this procedure is sometimes quite cumbersome in practice, in particular when there are many (space-time) indices involved. `Matchmakereft` can be used in this case to check if the operators defined in the EFT are linearly independent for arbitrary off-shell kinematics or not. This is done by checking the rank of the system of equations obtained by matching the EFT to all vanishing amplitudes. If several operators are linearly dependent, the rank will be smaller than the number of operators and `matchmakereft` will solve the system of equations to provide the relationship of the list of the dependent operators in terms of a particular set of independent ones. This is achieved by defining an EFT with all the relevant operators (including possibly linearly dependent ones) and running the command `check_linear_dependece EFTModel` (see Section 4 for details). As a word of caution it should be emphasized that `matchmakereft` makes no assumption about the space-time dimension. Thus, operators that are linearly dependent in $D = 4$ but not in arbitrary $D$ are listed as linearly independent in `matchmakereft`.

## 7 Conclusions and outlook

EFTs have become a standard tool in quantum field theory to address multi-scale problems. Using a mass-independent renormalization scheme, the process of computing observables in models with disparate scales consists of integrating out particles at the highest mass threshold, running the WCs by means of the RGEs down to the next threshold, integrating out the particles at that threshold and so on until we reach the energies at which experimental measurements are performed. We then compute the corresponding experimental observable using the EFT relevant at that scale.

The current situation in particle physics seems to indicate that indeed, physics beyond the SM, if present, is likely to be at a scale much larger than the energies at which we are performing our experiments. In searches beyond the SM, the relevant EFT seems to be the SMEFT at energies above the electroweak scale and the LEFT below that scale. The process of running in the SMEFT, matching to the LEFT and running in the LEFT down to experimental energies has been recently completed, up to mass dimension 6 and one-loop order, and it is now available in computer tools. The matching of arbitrary models to the (dimension 6) SMEFT has been also recently solved at tree level. Thus, the only missing piece to perform a complete one-loop analysis of the implications of experimental data on new physics models is the one-loop matching of arbitrary models onto the SMEFT.

In this article we have introduced `matchmakereft`, a computer tool that performs tree-level and one-loop matching of arbitrary models onto arbitrary EFTs. `Matchmakereft` is

robust, efficient, flexible and fully automated and can be trivially used to perform the step to one-loop matching of new physics models onto the SMEFT to complete the program outlined above. Due to its flexibility `matchmakereft` has many more applications than just the matching of new physics models onto the SMEFT. First it can match any model onto any EFT. Thus, it can be used to match new physics models to operators of dimension higher than 6 in the SMEFT (see [70] for the definition of the bosonic sector of a complete Green basis for the SMEFT at dimension 8), or to other EFTs beyond the SMEFT that can be also phenomenologically interesting, including for instance new light (or heavy particles) like an axion-like particle or a right-handed neutrino [82]. `Matchmakereft` is also able to match divergences in EFTs to compute the RGEs of arbitrary theories. Again this includes other EFTs beyond the dimension-6 SMEFT (or LEFT) but also renormalizable (or not) arbitrary theories. Other applications of `matchmakereft` include an automated basis translation between two (Green or physical) bases of an EFT or the reduction of an off-shell linearly dependent set of operators to a minimal Green basis. All these applications are performed in an automated way, with minimal interaction from the user, that only needs to provide the information of the UV and the EFT models.

Version `1.0.0` of `matchmakereft` performs an off-shell matching in the background-field gauge, thus providing a significant (kinematic and gauge) redundancy that is used to perform numerous cross-checks of the computed WCs. There are nevertheless some limitations in version `1.0.0` that we plan to overcome in future versions of `matchmakereft`. Among these some of the most significant are the following:

- flavour indices are currently not supported for massive particles. There is an experimental version of this in `matchmakereft` and we expect to implement it in the near future.

- Amplitudes with many external particles (as required to match operators with many fields) can have a very large number of Feynman diagrams at one loop. When the number is very large the calculation can become very slow. We plan to introduce new procedures to deal with this issue, including the splitting of groups of Feynman diagrams that contribute to a single amplitude and the parallelization of the calculation to allow an efficient computation in multi-core or multi-node computer systems.

`Matchmakereft` has already been used in several physical applications and we have performed an extensive number of non-trivial cross-checks to ensure the robustness of the calculations produced by `matchmakereft`. Its flexibility and efficiency will allow the particle physics community to analyze in a fully systematic and automated way the one-loop phenomenology or arbitrary new physics models.

## Note added

We would like to emphasize that `matchmakereft` is currently in active development. We encourage the users to download the latest stable version of the program (`matchmakereft` will warn the user that an outdated version is being used) and to consult the manual for new developments with respect to the version presented here. At the time of the acceptance of this article, `matchmakereft v1.0.10` is already available with some minor bugs fixed and some new features like the possibility of computing the relevant amplitudes in a parallel fashion or splitting amplitudes with a large number of diagrams in subsets with a fixed number of them, which are then automatically combined at the end of the calculation.

# Acknowledgments

This has been a long project that has benefited from discussions with many people. We would like to thank F. del Águila, J. de Blas, C. Bobeth, M. Ciuchini, A. Freytas, J. Fuentes-Martín, G. Guedes, M. Gorbahn, U. Haisch, Z. Kunstz, M. Neubert, U. Nierste, M. Pérez-Victoria, A. Pich, E. Salvioni, A. Signer and M. Trott for useful discussions. We are especially grateful to C. Anastasiou for collaboration in the initial stages of the project, to M. Chala, for continuos discussions, suggestions and requests that allowed us to "milk `matchmakereft` for all its worth", to T. Hahn, for patiently explaining many `FORM` tricks, and to G.P. Passarino, for daring us to "finish `matchmakereft` already!". We would like to thank Nicolasa Navarrete for the design of the logo and her creative support. We would also like to thank the referees for their very detailed reports and Alejo Rossia for pointing some typos to us. This work has been partially supported by the Ministry of Science and Innovation and SRA (10.13039/501100011033) under grant PID2019-106087GB-C22, by the Junta de Andalucía grants FQM 101, A-FQM-211-UGR18 and P18-FR-4314 (FEDER). AC acknowledges funding from the European Union's Horizon 2020 research and innovation programme under the Marie Sklodowska-Curie grant agreement No 754446 and UGR Research and Knowledge Transfer Found - Athenea3i. PO is funded by an FPU grant from the Spanish government. Work in Mainz University by AC was supported by the Cluster of Excellence Precision Physics, Fundamental Interactions, and Structure of Matter (PRISMA+ EXC 2118/1) funded by the German Research Foundation(DFG) within the German Excellence Strategy (Project ID 39083149), and by grant 05H18UMCA1 of the German Federal Ministry for Education and Research (BMBF).

# A `Matchmakereft` installation

## A.1 Prerequisites

In order to be able to run `matchmakereft`, some prerequisites need to be met. First of all, the following programs need to be installed:

- `Mathematica`, version 10 or higher.

- `FORM` : Installation can be checked by typing `form -v` in a terminal. Binaries or the source code can be downloaded from http://www.nikhef.nl/~form/

- `QGRAF` : Installation can be checked by typing `qgraf` in a terminal. Binaries or the source code can be downloaded from http://cfif.ist.utl.pt/~paulo/qgraf.html. Please note that versions of `QGRAF` earlier than 3.5 had a maximun multiplicity of vertices equal to 6 particles. If vertices with more than 6 particles are needed (this can happen if dimension-8 operators are considered for instance) the source code has to be modified accordingly and compiled again. The maximum multiplicity of `QGRAF` v3.5 is 8.

The binaries of both `FORM` and `QGRAF` need to be located in some path that is included in the binary path of the system, in such a way that they can be executed from any possible location. It is also necessary to have installed

- `Python` (3.5 or higher): Installation can be checked by typing `python --version` in a terminal. In some systems `python3` might have to be explicitly invoked.

- `FeynRules` the mathematica package, is needed for the creation of `matchmakereft` models. It can be downloaded from https://feynrules.irmp.ucl.ac.be/.

Matchmakereft is available both in the Python Package Index (PyPI) https://pypi.org/project/matchmakereft/ as well as in the Anaconda Python distribution https://anaconda.org/matchmakers/matchmakereft. Moreover,

- pip. In order to access packages from PyPI, pip needs to be installed on the system. It is available in most Linux distributions and can be easily installed on MacOS and Windows. Installation can be checked by typing pip -V in a terminal. In some systems pip3 has to be explicitly invoked. General information about installing packages fcan be found on https://packaging.python.org/en/latest/tutorials/installing-packages/.

- conda. Instructions about Anaconda installation on different OS can be found on https://docs.anaconda.com/anaconda/install/index.html. Explicit information about installing conda packages can be found on https://docs.anaconda.com/anacondaorg/user-guide/howto/.

## A.2 Installing matchmakereft

### A.2.1 PyPI

Once pip is installed in the system, matchmakereft can be installed by just typing in a terminal (we denote the terminal prompt as >)

```
> python3 -m pip install matchmakereft --user
```

or by typing

```
> pip install matchmakereft --user
```

Alternatively, if the installation has been downloaded from the project web page, it can be installed via

```
> python3 -m pip install matchmakereft-x.x.x.tar.gz  --user
```

where the x.x.x correspond to the version being installed.

We can get information about matchmakereft by writing

```
> pip3 show matchmakereft
Name: matchmakereft
Version: 1.0.0
Summary: One loop matching
Home-page: https://ftae.ugr.es/matchmakereft/
Author: Adrian Carmona, Achilleas Lazopoulos, Pablo Olgoso, Jose
   Santiago
Author-email: adrian@ugr.es, lazopoulos@itp.phys.ethz.ch,
   pablolgoso@ugr.es, jsantiago@ugr.es
License: Creative Commons Attribution-Noncommercial-Share Alike
   license
Location: /home/user/.local/lib/python3.9/site-packages
Requires: requests, yolk3k, setuptools
Required-by:
```

If matchmakereft is already installed in the system, it is possible to check for possible updates by writing

```
> pip install --upgrade matchmakereft --user
```

whereas one can remove it by typing

```
> pip uninstall matchmakereft
```

### A.2.2 Anaconda

For users employing Anaconda python distribution, `matchmakereft` can be installed by just typing

```
conda install -c matchmakers matchmakereft
```

## A.3 Updating the system path

Once `matchmakereft` is installed, we have to ensure that the corresponding executable is included in the user path. This can be easily achieved by copying the following two lines in the `~/.bashrc` file (or the equivalent one if a different shell is used)

```
1 export PY_USER_BIN=$(python -c 'import site; print(site.USER_BASE + "/
    bin")')
2 export PATH=$PY_USER_BIN:$PATH
```

If the user prefers to install without the `--user` option they should ensure that the directory where the executables are installed is included in the path.

# B Dealing with $\gamma_5$

When computing the matching equations with `matchmakereft`, we might encounter loop integrals that involve fermionic traces with chiral projectors, $P_{L,R}$. Among such traces, those that contain an odd number of $\gamma_5$ and six or more $\gamma$-matrices are known to be ambiguous, due to the impossibility of finding a regulator that is Lorenz invariant and preserves the chiral structure of the theory. In `matchmakereft` we regulate integrals within Dimensional Regularization, so we have to face the question of how to compute these traces consistently.

Limiting ourselves, at first, to matching renormalizable UV models to the SMEFT at one loop, the cases we have to deal with, in practice, are few. In order to have a trace in the first place, there can be no external fermions in the related diagram. Moreover, the ambiguity is proportional to the fully antisymmetric $\epsilon_{\mu\nu\rho\sigma}$ tensor, and can therefore contribute to the Wilson Coefficient of one of the few CP violating bosonic operators of the SMEFT, with two or three field strength tensors. We therefore need two or three external gauge bosons. In order to have a trace with at least 6 $\gamma$-matrices we, therefore need four or three internal fermionic propagators respectively. In summary, we only need to worry about boxes contributing to $H^\dagger H X_{\mu\nu} \tilde{X}^{\mu\nu}$ type operators or triangles contributing to $X_\nu^\mu X_\rho^\nu \tilde{X}_\rho^\mu$. Triangle diagrams contributing to $X_\nu^\mu X_\rho^\nu \tilde{X}_\rho^\mu$, with fermions in the internal lines, however, are non-ambiguous because there are no $\gamma_5$'s involved in the vertices: all heavy fermions have to be vector-like[6].

Next, let's look at the box diagrams contributing to $H^\dagger H X_{\mu\nu} \tilde{X}^{\mu\nu}$. There are four internal fermionic propagators, of which one or more can correspond to heavy particles in the UV model. As mentioned, the corresponding traces have ambiguous and non-ambiguous parts. The unambiguous part can be computed in any $\gamma_5$-scheme. The ambiguous part is proportional to $D-4$ and is only of interest if it multiplies one of the singularities of the integral. Moreover, since the UV theory is renormalizable, the maximum number of $\gamma$-matrices is 6, which means that terms in the numerator of such integrals with mass insertions have a lower number of $\gamma$'s and are therefore unambiguous. There are two types of singularities: UV singularities corresponding to the UV structure of the full theory, and IR singularities that appear after the hard region expansion is performed. The latter correspond to the UV singularities of the SMEFT.

---

[6]Triangle diagrams with only light particles are ambiguous, but they do not concern us during matching. The corresponding ambiguities are fixed by the anomaly cancellation mechanism that we assume any EFT has built in.

The generic structure of the potentially ambiguous part of such box integrals is, up to couplings and group theory factors:

$$I^{\mu\nu} \sim \int_k T_{\mu_1\dots\mu_4\mu\nu} \prod_{i=1}^4 \frac{(k+q_i)^{\mu_i}}{(k+q_i)^2 - m_i^2}, \tag{69}$$

where $\int_k \equiv \int \frac{d^D k}{(2\pi)^D}$, and $T_{\mu_1\dots\mu_4\mu\nu}$ denotes a trace with an odd number of $\gamma_5$ insertions and six $\gamma$'s, e.g.

$$\text{Tr}[\gamma_{\mu_1}\gamma_5\gamma_{\mu_2}\gamma_\mu\gamma_{\mu_3}\gamma_5\gamma_\nu\gamma_{\mu_4}\gamma_5]. \tag{70}$$

Performing the hard region expansion and the tensor reduction we get

$$
\begin{aligned}
I^{\mu\nu} \sim &\int_k T_{\mu_1\dots\mu_4\mu\nu} \frac{1}{\prod_{i=1}^4(k^2-m_i^2)} \\
&\left( k^4 \frac{g^{\mu_1\mu_2\mu_3\mu_4}}{D(D+2)}(1+\sum_i \frac{q_i^2}{k^2-m_i^2}) + k^6 \frac{g^{\mu_1\mu_2\mu_3\mu_4\rho\sigma}}{D(D+2)(D+4)} \sum_{i\le j} \frac{4q_i^\rho q_j^\sigma}{(k^2-m_i^2)(k^2-m_j^2)} \right. \\
&\left. -2k^4 \sum_{i,m} q_i^{\mu_i} q_m^\rho \frac{g^{\{\hat\mu_i\}\rho}}{D(D+2)} \frac{1}{k^2-m_m^2} + \sum_{i<j} k^2 \frac{g^{\{\hat\mu_i,\hat\mu_j\}}}{D} q^{\mu_i} q^{\mu_j} \right) \\
\sim &\int_k T_{\mu_1\dots\mu_4\mu\nu} \frac{g^{\mu_1\mu_2\mu_3\mu_4}}{D(D+2)} \frac{k^4}{\prod_{i=1}^4(k^2-m_i^2)} + \int_k T_{\mu_1\dots\mu_4\mu\nu} \frac{k^2}{\prod_{i=1}^4(k^2-m_i^2)} \frac{1}{D(D+2)} \\
&\left( g^{\mu_1\mu_2\mu_3\mu_4} \sum_i q_i^2 \left(1+\frac{m_i^2}{k^2-m_i^2}\right) + \frac{g^{\mu_1\mu_2\mu_3\mu_4\rho\sigma}}{(D+4)} \sum_{i\le j} 4q_i^\rho q_j^\sigma \prod_{l=i,j}\left(1+\frac{m_l^2}{(k^2-m_l^2)}\right) \right. \\
&\left. -2\sum_{i,m} q_i^{\mu_i} q_m^\rho g^{\{\hat\mu_i\}\rho}(1+\frac{m_m^2}{k^2-m_m^2}) + \sum_{i<j}(D+2) g^{\{\hat\mu_i,\hat\mu_j\}} q^{\mu_i} q^{\mu_j} \right), \tag{71}
\end{aligned}
$$

where $\{\hat\mu_i\} = \mu_j\mu_k\mu_r$ with $j,k,r \ne i$, and $\{\hat\mu_i\hat\mu_j\} = \mu_k\mu_r$ with $k,r \ne i,j$.

The first term, of $\mathcal{O}(p^0)$, is UV singular. It does not lead to ambiguities, though, since $g^{\mu_1\mu_2\mu_3\mu_4}$ makes the $\gamma_5$-dependent part of the trace vanish. The rest of the terms, of order $\mathcal{O}(p^2)$, are UV finite, but can be IR singular, depending on the number of heavy propagators: for two, three or four heavy propagators, there is no singularity. For one heavy and three light propagators, however, we do have an IR singularity.

We, therefore, conclude that ambiguous contributions from $\gamma_5$-odd traces in box integrals are present only in diagrams with one heavy and three light propagators. They appear as a product of the ambiguous $(D-4)$ coefficient of the trace multiplying a $1/\epsilon$ pole of IR type resulting from the hard region expansion. We would like to remain within the naive anticommuting $\gamma_5$ scheme when computing traces in `matchmakereft`. To this end, we wish to fix the ambiguous contributions of the $\gamma_5$-odd traces in box integrals with one heavy massive fermion, a posteriori. We note that the Wilson coefficient we are trying to match must be real: all operators of the type $H^\dagger H X_{\mu\nu} \tilde{X}^{\mu\nu}$ are hermitian. Therefore any contribution from $\gamma_5$-odd traces, being imaginary, must be multiplied by a purely imaginary product of couplings of the full theory. But, in the case that ambiguities are present, i.e. when the UV theory has Yukawa terms with the Higgs boson, one heavy and one light fermion, leading to box diagrams with a single fermionic propagators, the corresponding Yukawa couplings are complex conjugates of each other, by virtue of the hermiticity of the UV Lagrangian. As a result, the product of all couplings is real and, therefore, the $\gamma_5$-odd contributions are purely imaginary. They can be set to zero by hand, at the end of the computation. Note that this does not imply that the ambiguous contributions are zero: it is the sum of ambiguous and non-ambiguous traces that are set to zero.

Note that this procedure also works if the effective theory is not the SMEFT. In such cases, more than one scalar field might be present, allowing for non-hermitian operators of the type $\phi_1^\dagger \phi_2 X_{\mu\nu} \tilde{X}^{\mu\nu}$. The constraint that the corresponding Wilson coefficient is real does not apply any more. It is, however still true that the Wilson coefficient of this operator should equal the complex conjugate of the Wilson coefficient of the hermitian conjugate operator $\phi_2^\dagger \phi_1 X_{\mu\nu} \tilde{X}^{\mu\nu}$. This is, again, enough to remove the ambiguities.

## C  A minimal complete example

In this section we demonstrate many of the features of `matchmakereft` with a concrete example involving two scalar fields, a light, but not massless, field $\phi$ and a heavy field $\Phi$. Our model is described by the Lagrangian:

$$\mathcal{L} = \frac{1}{2}(\partial_\mu \phi)^2 - \frac{1}{2}m_L^2 \phi^2 + \frac{1}{2}(\partial_\mu \Phi)^2 - \frac{1}{2}M_H^2 \Phi^2 - \frac{\lambda_0}{4!}\phi^4 - \frac{\lambda_2}{4}\phi^2 \Phi^2 - \frac{\kappa}{2}\phi^2 \Phi, \tag{72}$$

which we want to match to the EFT Lagrangian without the heavy scalar,

$$\mathcal{L}_{\text{EFT}} = \frac{\alpha_{4k}}{2}(\partial_\mu \phi)^2 - \frac{\alpha_2}{2}\phi^2 - \frac{\alpha_4}{4!}\phi^4 - \frac{\alpha_6}{6!}\phi^6 - \frac{\tilde{\alpha}_6}{4!}\phi^3 \partial^2 \phi - \frac{\hat{\alpha}_6}{2}\left(\partial^2 \phi\right)^2. \tag{73}$$

We will use this Lagrangian during off-shell matching. Subsequently, the kinetic term can be canonically normalized, and the redundant operators can be eliminated. Two of the three operators of dimension 6 are redundant. We choose $\phi^6$ as the independent operator. Using equations of motion we can readily find that:

$$\phi^3 \partial^2 \phi \rightarrow -\alpha_2 \phi^4 - \frac{1}{3!}\alpha_4 \phi^6, \tag{74}$$

$$\left(\partial^2 \phi\right)^2 \rightarrow \alpha_2^2 \phi^2 + \frac{\alpha_2 \alpha_4}{3}\phi^4 + \frac{\alpha_4^2}{36}\phi^6. \tag{75}$$

Eliminating these operators from the Lagrangian would induce the shifts

$$\alpha_2 \quad \rightarrow \quad \alpha_2 + \alpha_2^2 \hat{\alpha}_6 \tag{76}$$

$$\alpha_4 \quad \rightarrow \quad \alpha_4 - \tilde{\alpha}_6 \alpha_2 + 4\alpha_2 \alpha_4 \hat{\alpha}_6 \tag{77}$$

$$\alpha_6 \quad \rightarrow \quad \alpha_6 - 5\tilde{\alpha}_6 \alpha_4 + 10\hat{\alpha}_6 \alpha_4^2 \tag{78}$$

The coupling $\kappa$ of this model is a dimensionful coupling, and is expected to be parametrically of the order of the heavy mass scale $M_H$. Thus, $\frac{\kappa}{M_H}$ is of $\mathcal{O}(1)$ and is kept throughout the matching procedure consistently.

The `FeynRules` file for the UV model, saved at `two_scalars.fr`, is shown below.

```
1 (* --- Contents of Feynrules file two_scalars.fr --- *)
2 M$ModelName = "two_scalars";
3 (* **** Particle classes **** *)
4 M$ClassesDescription = {
5 S[1] == {ClassName -> phiH, SelfConjugate -> True, Mass -> MH,
6       FullName -> "heavy"},
7 S[2] == {ClassName -> phi, SelfConjugate -> True, Mass -> mL,
8       FullName -> "light"}
9 };
10 (* *****   Parameters   ***** *)
11 M$Parameters = {
12 MH == {ParameterType -> Internal, ComplexParameter -> False},
13 mL == {ParameterType -> Internal, ComplexParameter -> False},
```

```
14 V == {ParameterType -> Internal, ComplexParameter -> False},
15 lambda0 == {ParameterType -> Internal, ComplexParameter -> False},
16 kappa == {ParameterType -> Internal, ComplexParameter -> False},
17 lambda2 == {ParameterType -> Internal, ComplexParameter -> False}
18 };
19 (* *****    Lagrangian   ***** *)
20 Ltot := Block[{mu},
21   + 1/2 * del[phi,mu] *  del[phi,mu]  + 1/2 *del[phiH,mu] * del[phiH,
     mu]
22   - 1/2 * MH^2 * phiH^2 - 1/2 * mL^2 * phi^2
23   - lambda0 / 24 * phi^4 - kappa / 2  * phi^2 * phiH
24   - lambda2 / 4 * phi^2 * phiH^2
25   ];
```

Note that we use the keyword `FullName` to characterize each field as `"heavy"` or `"light"`. This is mandatory: `matchmaker` uses this keyword to distinguish between fields that are integrated out and those that are light and are also present in the EFT. Also note that all the parameters that are used in the Lagrangian, masses as well as couplings, must be declared. In this example all parameters are real.

The `FeynRules` file for the EFT model, saved at `one_scalar.fr`, is:

```
1 (* --- Contents of Feynrules file for the EFT model  one_scalar.fr ---
     *)
2 M$ModelName = "one_scalar";
3 (* **** Particle classes **** *)
4 M$ClassesDescription = {
5 S[2] == {ClassName -> phi, SelfConjugate -> True, Mass -> 0,
6         FullName -> "light"}
7 };
8 (* *****    Parameters    ***** *)
9 M$Parameters = {
10 alpha4kin == {ParameterType -> Internal, ComplexParameter -> False},
11 alpha2mass == {ParameterType -> Internal, ComplexParameter -> False},
12 alpha4 == {ParameterType -> Internal, ComplexParameter -> False},
13 alpha6 == {ParameterType -> Internal, ComplexParameter -> False},
14 alpha6Rtilde == {ParameterType -> Internal, ComplexParameter -> False
     },
15 alpha6Rhat == {ParameterType -> Internal, ComplexParameter -> False}
16 };
17 (* *****    Lagrangian    ***** *)
18 Ltot := Block[{mu,mu2},
19   1/2 * alpha4kin * del[phi,mu] * del[phi,mu]
20   - 1/2 * alpha2mass * phi^2
21   - alpha4/24 * phi^4
22   - alpha6 * phi^6/720
23   - alpha6Rtilde/24 * phi^3 * del[del[phi,mu],mu]
24   - alpha6Rhat/2 * del[del[phi,mu],mu]  *  del[del[phi,mu2],mu2]
25   ];
```

Note that we have included WCs (denoted by `alpha`) also for the kinetic and mass terms (squared), as well as for all operators that are redundant solely due to the equations of motion.

In order for `matchmakereft` to perform the reduction to the physical basis, we need to provide a set of relations that express the redundant WCs in terms of the irreducible ones, see Eq. (78). This is done at `one_scalar.red`:

```
1 (* --- Contents of one_scalar.red --- *)
2 finalruleordered = {
3   alpha6 -> - alpha6Rtilde * alpha4 *5 + alpha6Rhat * alpha4^2 * 10 +
     alpha6 ,
4   alpha4 -> alpha4 - alpha6Rtilde * alpha2mass + 4 * alpha6Rhat *
     alpha2mass * alpha4 ,
5   alpha4kin -> alpha4kin ,
```

```
6   alpha2mass -> alpha2mass +  alpha6Rhat * alpha2mass^2
7   }
```

Note that **only** the WCs corresponding to physical operators, among those appearing in the EFT Lagrangian and defined in the file `one_scalar.fr`, must be present on the left hand side of the replacement rules in this file. The WCs corresponding to redundant operators appear only on the right hand side. When these rules are used, both redundant and non-redundant WCs have been matched and are known as functions of the parameters of the UV theory. The rules are therefore instructions on how to update the non-redundant WCs, to include the effect of the redundant ones.

With these files prepared we are ready to proceed with matching. In the matching directory, where `two_scalars.fr`,`one_scalar.fr`,`one_scalar.red` are present, we can run `matchmakereft`:

```
>matchmakereft
```

upon which we enter the python interface

```
Welcome to matchmakereft
Please refer to arXiv:2112.10787 when using this code

matchmakereft>
```

We first need to create the matchmaker models, i.e. the directories with all the necessary information for the UV and the EFT models. We do this by

```
matchmakereft> create_model two_scalars.fr
```

which has the response

```
Creating model two_scalars_MM. This might take some time depending on
    the complexity of the model
Model two_scalars_MM created
It took 7 seconds to create it.
```

We can now observe that the directory `two_scalars_MM` is created. We proceed with creating the EFT model

```
matchmakereft> create_model one_scalar.fr
Creating model one_scalar_MM. This might take some time depending on
    the complexity of the model
Model one_scalar_MM created
It took 7 seconds to create it.
```

The `one_scalar_MM` directory is now created as well, and we are ready for the matching calculation. This is performed by the `match_model_to_eft` command:

```
matchmakereft> match_model_to_eft two_scalars_MM one_scalar_MM
```

Upon completion, the results of the matching are stored in the UV model directory, in this case `two_scalars_MM`. The file `two_scalars_MM/MatchingProblems.dat` contains troubleshooting information in case the matching procedure failed. In our case it is an empty list, indicating no problems:

```
1  problist = {}
```

The result of the matching procedure is in `two_scalars_MM/MatchingResults.dat`, a Mathematica file with a list of lists of replacement rules. The off-shell matching gives the following results for the WCs of the Green basis. At tree level the non-vanishing contributions

are,

$$\alpha_2^{(0)} = m_L^2, \tag{79}$$

$$\alpha_4^{(0)} = \lambda_0 - \frac{3\kappa^2}{M_H^2}, \tag{80}$$

$$\alpha_6^{(0)} = \frac{45\lambda_2\kappa^2}{M_H^4}, \tag{81}$$

$$\tilde{\alpha}_6^{(0)} = \frac{4\kappa^2}{M_H^4}. \tag{82}$$

At one loop, keeping terms up to $O\left(\frac{\kappa^{2n}}{M_H^{2n}}\frac{m_L^2}{M_H^2}\right)$, we get (we define $L_M \equiv \log(\frac{\mu^2}{M_H^2})$)

$$\alpha_2^{(1)} = -\frac{1}{16\pi^2}(1+L_M)\left(\frac{\kappa^2 m_L^4}{M_H^4} + \frac{\kappa^2 m_L^2}{M_H^2} + \frac{1}{2}\lambda_2 M_H^2 + \kappa^2\right), \tag{83}$$

$$\alpha_{4k}^{(1)} = \frac{1}{16\pi^2}\left(\frac{5\kappa^2 m_L^2}{2M_H^4} + \frac{\kappa^2}{2M_H^2}\right) + \frac{1}{16\pi^2}\frac{\kappa^2 m_L^2}{M_H^4}L_M, \tag{84}$$

$$\alpha_4^{(1)} = \frac{1}{16\pi^2}\left(\frac{48\kappa^4 m_L^2}{M_H^6} + \frac{24\lambda_2\kappa^2 m_L^2}{M_H^4} - \frac{12\lambda_0\kappa^2 m_L^2}{M_H^4} + \frac{18\kappa^4}{M_H^4} + \frac{18\lambda_2\kappa^2}{M_H^2} - \frac{6\lambda_0\kappa^2}{M_H^2}\right) \tag{85}$$

$$+ \frac{1}{16\pi^2}\left(\frac{36\kappa^4 m_L^2}{M_H^6} + \frac{18\lambda_2\kappa^2 m_L^2}{M_H^4} - \frac{12\lambda_0\kappa^2 m_L^2}{M_H^4} + \frac{12\kappa^4}{M_H^4} + \frac{12\lambda_2\kappa^2}{M_H^2} - \frac{6\lambda_0\kappa^2}{M_H^2} - \frac{3\lambda_2^2}{2}\right)L_M, \tag{86}$$

$$\alpha_6^{(1)} = \frac{1}{16\pi^2 M_H^2}\left(-\frac{1290\kappa^6}{M_H^6} + \frac{720\lambda_0\kappa^4}{M_H^4} - \frac{1665\lambda_2\kappa^4}{M_H^4} + \frac{360\lambda_0\lambda_2\kappa^2}{M_H^2} - \frac{90\lambda_0^2\kappa^2}{M_H^2} - \frac{495\lambda_2^2\kappa^2}{M_H^2} + \frac{15\lambda_2^3}{2}\right) \tag{87}$$

$$+ \frac{1}{16\pi^2 M_H^2}\left(-\frac{810\kappa^6}{M_H^6} + \frac{540\lambda_0\kappa^4}{M_H^4} - \frac{945\lambda_2\kappa^4}{M_H^4} + \frac{270\lambda_0\lambda_2\kappa^2}{M_H^2} - \frac{90\lambda_0^2\kappa^2}{M_H^2} - \frac{270\lambda_2^2\kappa^2}{M_H^2}\right)L_M, \tag{88}$$

$$\tilde{\alpha}_6^{(1)} = \frac{1}{16\pi^2 M_H^2}\left(-\frac{107\kappa^4}{3M_H^4} + \frac{9\lambda_0\kappa^2}{M_H^2} - \frac{77\lambda_2\kappa^2}{3M_H^2} + \frac{\lambda_2^2}{3}\right)$$

$$+ \frac{1}{16\pi^2 M_H^2}\left(-\frac{14\kappa^4}{M_H^4} + \frac{2\lambda_0\kappa^2}{M_H^2} - \frac{16\lambda_2\kappa^2}{M_H^2}\right)L_M, \tag{89}$$

$$\hat{\alpha}_6^{(1)} = -\frac{\kappa^2}{96\pi^2 M_H^4}. \tag{90}$$

As can be seen in the equations above, the kinetic operator receives a correction and therefore $\phi$ is no longer canonically normalized. A field redefinition is needed to obtain a canonically normalized theory on which we can apply the corresponding redundancies to go to the physical basis. Matchmakereft does these two processes (canonical normalization and going to the physical basis) automatically. The resulting WCs in the physical basis, up to one loop order and $O\left(\frac{\kappa^{2n}}{M_H^{2n}}\frac{m_L^2}{M_H^2}\right)$, read

$$\alpha_2 = m_L^2 - \frac{1}{16\pi^2}\left(\frac{11\kappa^2 m_L^4}{3M_H^4} + \frac{3\kappa^2 m_L^2}{2M_H^2} + \frac{1}{2}\lambda_2 M_H^2 + \kappa^2\right)$$

$$- \frac{1}{16\pi^2}\left(\frac{2\kappa^2 m_L^4}{M_H^4} + \frac{\kappa^2 m_L^2}{M_H^2} + \frac{1}{2}\lambda_2 M_H^2 + \kappa^2\right)L_M, \tag{91}$$

$$\alpha_4 = \lambda_0 - \frac{4\kappa^2 m_L^2}{M_H^4} - \frac{3\kappa^2}{M_H^2} \tag{92}$$
$$+ \frac{1}{16\pi^2}\left(\frac{332\kappa^4 m_L^2}{3M_H^6} - \frac{80\lambda_0\kappa^2 m_L^2}{3M_H^4} + \frac{149\lambda_2\kappa^2 m_L^2}{3M_H^4} - \frac{\lambda_2^2 m_L^2}{3M_H^2} + \frac{25\kappa^4}{M_H^4} - \frac{7\lambda_0\kappa^2}{M_H^2} + \frac{20\lambda_2\kappa^2}{M_H^2}\right)$$
$$+ \frac{1}{16\pi^2}\left(\frac{60\kappa^4 m_L^2}{M_H^6} - \frac{16\lambda_0\kappa^2 m_L^2}{M_H^4} + \frac{34\lambda_2\kappa^2 m_L^2}{M_H^4} + \frac{16\kappa^4}{M_H^4} - \frac{6\lambda_0\kappa^2}{M_H^2} + \frac{14\lambda_2\kappa^2}{M_H^2} - \frac{3\lambda_2^2}{2}\right)L_M,$$

$$\alpha_6 = \frac{1}{M_H^2}\left(\frac{60\kappa^4}{M_H^4} - \frac{20\lambda_0\kappa^2}{M_H^2} + \frac{45\lambda_2\kappa^2}{M_H^2}\right)$$
$$+ \frac{1}{16\pi^2 M_H^2}\left(-\frac{1560\kappa^6 m_L^2}{M_H^8} + \frac{440\lambda_0\kappa^4 m_L^2}{M_H^6} - \frac{1635\lambda_2\kappa^4 m_L^2}{2M_H^6} - \frac{2320\kappa^6}{M_H^6} + \frac{3610\lambda_0\kappa^4}{3M_H^4}\right.$$
$$\left. - \frac{4955\lambda_2\kappa^4}{2M_H^4} - \frac{410\lambda_0^2\kappa^2}{3M_H^2} - \frac{490\lambda_2^2\kappa^2}{M_H^2} + \frac{1465\lambda_0\lambda_2\kappa^2}{3M_H^2} + \frac{15\lambda_2^3}{2} - \frac{5\lambda_0\lambda_2^2}{3}\right)$$
$$+ \frac{1}{16\pi^2 M_H^2}\left(-\frac{960\kappa^6 m_L^2}{M_H^8} + \frac{320\lambda_0\kappa^4 m_L^2}{M_H^6} - \frac{495\lambda_2\kappa^4 m_L^2}{M_H^6} - \frac{1260\kappa^6}{M_H^6} + \frac{760\lambda_0\kappa^4}{M_H^4}\right.$$
$$\left. - \frac{1425\lambda_2\kappa^4}{M_H^4} - \frac{100\lambda_0^2\kappa^2}{M_H^2} - \frac{240\lambda_2^2\kappa^2}{M_H^2} + \frac{350\lambda_0\lambda_2\kappa^2}{M_H^2}\right)L_M. \tag{93}$$

Next, we would like to compute the RGE equations for both models. For each model we need a new pair of FeynRules files as a starting point. We start for the RGEs for the UV model. The corresponding UV model, located in the file rge_two_scalars_uv.fr is the same as two_scalars.fr with two important differences. First, the heavy field H is also declared as "light" here.

```
S[1]  == {ClassName -> H, SelfConjugate ->True, Mass ->MH,
    FullName ->"light"}
```

Second, we introduce a tadpole term in the Lagrangian

```
(* ***** Lagrangian ***** *)
Ltot  := Block[{mu},
+ 1/2 * del[phi,mu] * del[phi,mu]
+ 1/2 *del[phiH,mu] * del[phiH,mu]
- 1/2 * MH^2 * phiH^2
- 1/2 * mL^2 * phi^2
-lambda0/24*phi^4
-kappa/2 *phi^2*phiH
- lambda2 / 4 * phi^2 * phiH^2
+ V*phiH (* Tadpole Term *)
];
```

The target model is now at rge_two_scalars_eft.fr. It consists of an EFT Lagrangian with the field *H* considered as a light field:

```
(* ---  Contents of rge_two_scalars_eft.fr  --- *)
M$ModelName = "rge_two_scalars_eft";
(* **** Particle classes **** *)
M$ClassesDescription = {
S[1]  == {ClassName ->  phiH, SelfConjugate -> True, Mass -> 0,
        FullName -> "light"},
S[2]  == {ClassName -> phi,SelfConjugate -> True,Mass -> 0,
        FullName -> "light"}
};
(* *****   Parameters   ***** *)
M$Parameters = {
```

```
12 alpha4kinphi == {ParameterType -> Internal, ComplexParameter -> False
      },
13 alpha4kinH == {ParameterType -> Internal, ComplexParameter -> False},
14 alpha2MH == {ParameterType -> Internal, ComplexParameter -> False},
15 alpha2ML == {ParameterType -> Internal, ComplexParameter -> False},
16 alpha4 == {ParameterType -> Internal, ComplexParameter -> False},
17 alphaV == {ParameterType -> Internal, ComplexParameter -> False},
18 alpha1 == {ParameterType -> Internal, ComplexParameter -> False},
19 alpha4H == {ParameterType -> Internal, ComplexParameter -> False},
20 alpha2 == {ParameterType -> Internal, ComplexParameter -> False},
21 alpha3 == {ParameterType -> Internal, ComplexParameter -> False}
22 };
23 (* *****   Lagrangian   ***** *)
24 Ltot := Block[{mu},
25   alphaV *  phiH
26   +1/2 * alpha4kinphi* del[phi,mu] *  del[phi,mu]
27   + 1/2 * alpha4kinH * del[phiH,mu] * del[phiH,mu]
28   - 1/2 * alpha2MH * phiH^2
29   -1/2 * alpha2ML * phi^2
30   - alpha4 / 24 * phi^4
31   -  alpha1 / 2  * phi^2 * phiH
32   - alpha2 / 4 * phi^2 * phiH^2
33   -alpha3 * phiH^3
34   -alpha4H * phiH^4
35 ];
```

Note the presence of the new interaction terms $\Phi^3, \Phi^4$: all possible operators of dimension up to 4, compatible with the symmetries should appear here. Also note that we now have a WC also for the kinetic term of the $\Phi$ field.

We don't need a `rge_two_scalars_eft.red` file at all, since all operators of dimension four present are physical. We can now create the two models with

```
matchmakereft> create_model rge_two_scalars_uv.fr
```

and

```
matchmakereft> create_model rge_two_scalars_eft.fr
```

and then we can proceed with the RGE computation via

```
matchmakereft> compute_rge_model_to_eft  rge_two_scalars_uv_MM
    rge_two_scalars_eft_MM
```

We then get, as before, inside the `rge_two_scalars_uv_MM` directory an empty problem file `MatchingProblems.dat`, as well as a `MatchingResults.dat` file. Moreover, there is another file produced, `RGEResult.dat` which contains the explicit form of the beta functions for our UV model.

```
1 RGEResult = {
2 \[Beta][alphaV] -> -1/16*(kappa*mL^2)/Pi^2
3 \[Beta][alpha4kinphi] -> 0,
4 \[Beta][alpha4kinH] -> 0,
5 \[Beta][alpha2MH] -> kappa^2/(16*Pi^2)+(lambda2*mL^2)/(16*Pi^2),
6 \[Beta][alpha2ML] -> kappa^2/(8*Pi^2)+(lambda2*MH^2)/(16*Pi^2) +
7       (lambda0*mL^2)/(16*Pi^2),
8 \[Beta][alpha4] -> (3*lambda0^2)/(16*Pi^2) +
9       (3*lambda2^2)/(16*Pi^2),
10 \[Beta][alpha1] ->(lambda0*kappa)/(16*Pi^2) + (kappa*lambda2)/(4*Pi^2)
      ,
11 \[Beta][alpha2] -> (lambda0*lambda2)/(16*Pi^2) + lambda2^2/(4*Pi^2),
12 \[Beta][alpha3] -> (kappa*lambda2)/(32*Pi^2),
13 \[Beta][alpha4H] -> lambda2^2/(128*Pi^2)}
```

Let us now touch upon the topic of tadpole contributions, which will also explain the reason for the $\alpha_V \Phi$ operator defined in line 25 of `rge_two_scalars_eft.fr`. In our model there are tadpole contributions to many 1LPI diagrams, due to the operator $\kappa \phi^2 \Phi$. They would vanish, up to one loop, if the mass of the light field was set to zero, but are non-vanishing otherwise. The resulting finite contributions are taken into account by our matching procedure. However, the corresponding poles, which contribute to the beta functions of $m_L$ and $\kappa$ are disregarded, during the RGE computation. However, there is an easy way to account for them. By adding the $\alpha_V \Phi$ term in the Lagrangian of `rge_two_scalars_eft.fr`, we induce the computation of the one-point function beta-function: as we see from line 2 of `RGEResult.dat`, we have

$$\beta(\alpha_V) = -\frac{1}{16\pi^2}\kappa m_L^2 . \tag{94}$$

If, instead of working with tadpole contributions, we would have shifted the field $\Phi \to \Phi + V$ we would have induced an explicit linear term $V M_H^2 \Phi$ and, due to the operators $\phi^2 \Phi$ and $\phi^2 \Phi^2$, we would modify the mass term for the light scalar, $m_L$ and the $\kappa$ coupling:

$$\tilde{m}_L^2 = m_L^2 + \kappa V + \frac{1}{2}\lambda_2 V^2 , \quad \tilde{\kappa} = \kappa + \lambda_2 V . \tag{95}$$

By setting the tree-level contribution of $V$ such that it cancels loop corrections order by order, we could eliminate tadpoles from the theory completely. Instead, here, we include their finite part in the calculation, and we absorb the pole in the renormalization of $m_L^2$ and $\kappa$. We should therefore modify the beta functions we read from `RGEResult.dat`, to account for the tadpole pole, by

$$\delta\beta(m_L^2) = \beta(\kappa V) = \frac{\kappa}{M_H^2}\beta(\alpha_V) = -\frac{\kappa^2}{16\pi^2}\frac{m_L^2}{M_H^2} , \tag{96}$$

$$\delta\beta(\kappa) = \beta(\lambda_2 V) = \frac{\lambda_2}{M_H^2}\beta(\alpha_V) = -\frac{\kappa\lambda_2}{16\pi^2}\frac{m_L^2}{M_H^2} . \tag{97}$$

Reading the results of `RGEResult.dat` and adding these contributions we get

$$\beta(m_L^2) = \frac{\lambda_2 M_H^2}{16\pi^2} + \frac{\kappa^2}{8\pi^2} + \frac{\lambda_0 m_L^2}{16\pi^2} - \frac{\kappa^2}{16\pi^2}\frac{m_L^2}{M_H^2} , \tag{98}$$

$$\beta(M_H^2) = \frac{\kappa^2}{16\pi^2} + \frac{\lambda_2 m_L^2}{16\pi^2} , \tag{99}$$

$$\beta(\lambda_0) = \frac{3\lambda_0^2}{16\pi^2} + \frac{3\lambda_2^2}{16\pi^2} , \tag{100}$$

$$\beta(\kappa) = \frac{\lambda_0 \kappa}{16\pi^2} + \frac{\lambda_2 \kappa}{4\pi^2} - \frac{\kappa\lambda_2}{16\pi^2}\frac{m_L^2}{M_H^2} , \tag{101}$$

$$\beta(\lambda_2) = \frac{\lambda_2^2}{4\pi^2} + \frac{\lambda_0 \lambda_2}{16\pi^2} . \tag{102}$$

We can perform the same procedure, but now with the EFT model. We create the model via the following `rge_one_scalar_uv.fr` file

```
1 M$ModelName = "rge_one_scalar_uv";
2 (* **** Particle classes **** *)
3 M$ClassesDescription = {
4 S[2] == {ClassName -> phi,SelfConjugate -> True,Mass -> mL,
5         FullName -> "light"}
6 };
```

```
7  (* *****    Parameters    ***** *)
8  M$Parameters = {
9  mL == {ParameterType -> Internal, ComplexParameter -> False},
10 a4 == {ParameterType -> Internal, ComplexParameter -> False},
11 a6 == {ParameterType -> Internal, ComplexParameter -> False}
12 };
13 (* *****    Lagrangian    ***** *)
14 Ltot := Block[{mu,mu2},
15   1/2  * del[phi,mu] * del[phi,mu] - 1/2 * mL^2 * phi^2
16   -a4/24 phi^4 - a6 * phi^6/720
17   ];
18
```

which contains the physical operators of the EFT model. Note that we have changed the names of the couplings to something other than alphaXXX[7]. We also create rge_one_scalar_eft.fr and rge_one_scalar_eft.red that are identical with one_scalar.fr and one_scalar.red defined above. Once again, we can create the two models and run compute_rge_model_to_eft. The result is

$$\beta(\alpha_2) = \frac{\alpha_2 \alpha_4}{16\pi^2}, \tag{103}$$

$$\beta(\alpha_4) = \frac{3\alpha_4^2}{16\pi^2} + \frac{\alpha_2 \alpha_6}{16\pi^2}, \tag{104}$$

$$\beta(\alpha_6) = \frac{15\alpha_4 \alpha_6}{16\pi^2}. \tag{105}$$

We can now check whether the matching conditions and the RGE equations are consistent with each other: if we match at $\mu = M_H$ and we evolve all couplings using the RGEs of the UV and the EFT model, to a lower scale $Q$, we should find the same expressions as when we match directly at $\mu = Q$. Let's see how this works in the case of the mass coefficient, $\alpha_2$. The matching condition for $\alpha_2$, at scale $\mu = M_H$, gives, see Eq. (91),

$$\alpha_2(M_H) = m_L^2(M_H) - \frac{1}{16\pi^2} \left( \frac{11\kappa^2 m_L^4}{3M_H^4} + \frac{3\kappa^2 m_L^2}{2M_H^2} + \frac{1}{2}\lambda_2 M_H^2 + \kappa^2 \right). \tag{106}$$

We can use Eq. (103) to evolve $\alpha_2$ from $M_H$ to a lower scale $Q$. In the leading-log approximation it reads,

$$\alpha_2(Q) = \alpha_2(M_H) + \frac{1}{2} L_Q \beta(\alpha_2) = \alpha_2(M_H) + \frac{1}{2} L_Q \frac{\alpha_2 \alpha_4}{16\pi^2}, \tag{107}$$

where $L_Q \equiv \log(\frac{Q^2}{M_H^2})$. Replacing the tree-level matched values of $\alpha_2$ and $\alpha_4$ we get

$$\alpha_2(Q) = m_L^2(M_H) - \frac{1}{16\pi^2} \left( \frac{11\kappa^2 m_L^4}{3M_H^4} + \frac{3\kappa^2 m_L^2}{2M_H^2} + \frac{1}{2}\lambda_2 M_H^2 + \kappa^2 \right) \tag{108}$$

$$- \frac{L_Q}{16\pi^2} \left( \frac{2\kappa^2 m_L^4}{M_H^4} + \frac{3\kappa^2 m_L^2}{2M_H^2} - \frac{\lambda_0 m_L^2}{2} \right). \tag{109}$$

If the matching condition for $\alpha_2$ had no tree-level contribution, this would be the full expression for $\alpha_2(Q)$. In our case, however, there is a tree-level contribution, $m_L^2(M_H)$. We need to use the RGE equation of the UV model, for $m_L$, Eq. (98) to evolve it to the scale $Q$. We then

---

[7]If alphaXXX names are used then matchmakereft automatically changes them to WCXXX.

get

$$\alpha_2(Q) = m_L^2(Q) - \frac{1}{16\pi^2}\left(\frac{11\kappa^2 m_L^4}{3M_H^4} + \frac{3\kappa^2 m_L^2}{2M_H^2} + \frac{1}{2}\lambda_2 M_H^2 + \kappa^2\right) \tag{110}$$

$$- \frac{L_Q}{16\pi^2}\left(\frac{2\kappa^2 m_L^4}{M_H^4} + \frac{\kappa^2 m_L^2}{M_H^2} + \frac{\lambda_2 M_H^2}{2} + \kappa^2\right). \tag{111}$$

Had we match directly at the scale $\mu = Q$ using Eq. (91), we would have found exactly the same result.

Similarly for $\alpha_4$, matching at $\mu = M_H$ and evolving to $\mu = Q$, with the help of the RGEs for both the UV and the EFT couplings, gives

$$\begin{aligned}\alpha_4(Q) = {}& \lambda_0 - \frac{4\kappa^2 m_L^2}{M_H^4} - \frac{3\kappa^2}{M_H^2} + \frac{1}{16\pi^2}\left(\frac{332\kappa^4 m_L^2}{3M_H^6} + \frac{149\lambda_2\kappa^2 m_L^2}{3M_H^4} - \frac{80\lambda_0\kappa^2 m_L^2}{3M_H^4}\right.\\ & \left. - \frac{\lambda_2^2 m_L^2}{3M_H^2} + \frac{25\kappa^4}{M_H^4} + \frac{20\lambda_2\kappa^2}{M_H^2} - \frac{7\lambda_0\kappa^2}{M_H^2}\right) + \frac{1}{16\pi^2}L_Q\left(\frac{60\kappa^4 m_L^2}{M_H^6} + \frac{34\lambda_2\kappa^2 m_L^2}{M_H^4}\right.\\ & \left. - \frac{16\lambda_0\kappa^2 m_L^2}{M_H^4} + \frac{16\kappa^4}{M_H^4} + \frac{14\lambda_2\kappa^2}{M_H^2} - \frac{6\lambda_0\kappa^2}{M_H^2} - \frac{3\lambda_2^2}{2}\right), \end{aligned} \tag{112}$$

in agreement (up to terms of $O\left(\frac{\kappa^{2n}}{M_H^{2n}}\frac{m_L^2}{M_H^2}\right)$) with what we would get by matching directly at $\mu = Q$, see Eq. (92).

## D    SMEFT Green basis

Here we present a Green basis of the dimension 6 SMEFT including two and four-fermion evanescent operators. As far as possible, we will follow the notation and the conventions used in the model file included with `matchmakereft`. We present first the renormalizable SM Lagrangian, which reads

$$\begin{aligned}\mathcal{L}_{\text{SM}} = {}& -\frac{1}{4}G_{\mu\nu}^A G^{A\mu\nu} - \frac{1}{4}W_{\mu\nu}^I W^{I\mu\nu} - \frac{1}{4}B_{\mu\nu}B^{\mu\nu} + (D_\mu H)^\dagger D^\mu H - m^2 H^\dagger H - \lambda(H^\dagger H)^2 \\ & + i[\bar{\ell}\slashed{D}\ell + \bar{e}\slashed{D}e + \bar{q}\slashed{D}q + \bar{u}\slashed{D}u + \bar{d}\slashed{D}d] - [\bar{\ell}Y_e eH + \bar{q}Y_u u\tilde{H} + \bar{q}Y_d dH + \text{h.c.}]. \end{aligned} \tag{113}$$

Hereinafter, we omit flavour and gauge indices whenever possible. Otherwise, we use $i, j, k, l, \ldots$ as flavour indices and $A, B, C, \ldots$ and $I, J, K, \ldots$ for the adjoint representation of $SU(3)$ and $SU(2)$, respectively. We will use on the other hand $a, b, c, \ldots$ and $r, s, t, \ldots$ for the fundamental representation of the color and the electroweak group, respectively. In the model file coming along with `matchmakereft` we use a slightly different notation for the couplings in the renormalizable Lagrangian. For instance,

$$Y_u \rightarrow \texttt{alpha0lambdau}, \ \lambda \rightarrow \texttt{alpha0lambda}, \ m^2 \rightarrow \texttt{alpha0muH2}, \ \ldots . \tag{114}$$

We refer the reader to the model file `SMEFT_Green_Bpreserving.fr` for more details.

The doublet $\tilde{H}$ is defined by $\tilde{H} = i\sigma^2 H^*$ as usual and we assume the following definition for the covariant derivative

$$D_\mu q = (\partial_\mu - ig_3 T^A G_\mu^A - ig_2 \frac{\sigma^I}{2}W_\mu^I - ig_1 YB_\mu)q, \tag{115}$$

where $T^A = \lambda^A/2$ with $\lambda^A$ and $\sigma^I$ the Gell-Mann and Pauli matrices, respectively. Correspondingly, the field strength tensors are

$$
\begin{aligned}
G^A_{\mu\nu} &= \partial_\mu G^A_\nu - \partial_\nu G^A_\mu + g_3 f^{ABC} G^B_\mu G^C_\nu \,, &(116)\\
W^I_{\mu\nu} &= \partial_\mu W^I_\nu - \partial_\nu W^I_\mu + g_2 \epsilon^{IJK} W^J_\mu W^K_\nu \,, &(117)\\
B_{\mu\nu} &= \partial_\mu B_\nu - \partial_\nu B_\mu, &(118)
\end{aligned}
$$

and their covariant derivatives

$$
\begin{aligned}
(D_\rho G_{\mu\nu})^A &= \partial_\rho G^A_{\mu\nu} + g_3 f^{ABC} G^B_\rho G^C_{\mu\nu}, &(119)\\
(D_\rho W_{\mu\nu})^I &= \partial_\rho W^I_{\mu\nu} + g_2 \epsilon^{IJK} W^J_\rho W^K_{\mu\nu}, &(120)\\
(D_\rho B_{\mu\nu}) &= \partial_\rho B_{\mu\nu}, &(121)
\end{aligned}
$$

with $f^{ABC}$ and $\epsilon^{IJK}$ the $SU(3)$ and $SU(2)$ structure constants, respectively.

The $B$-number preserving dimension 6 Green basis is presented in Tables 1-9. We follow the convention of denoting operators in the Warsaw basis [74] by $\mathcal{O}$, whereas redundant and evanescent operators are denoted by $\mathcal{R}$ and $\mathcal{E}$, respectively. We should stress that we define a redundant operator, not as one that vanishes when using the equations of motion, but rather as one that is equivalent to a combination of operators in the physical basis when using the equations of motion. Similarly, evanescent operators do not vanish when going to $D = 4$ dimensions but rather they are equivalent (including order $\epsilon$ terms) to operators in the physical basis. We follow the notation of [65] regarding redundant operators and include evanescent operators with two and four fermions as well as possibly featuring charge conjugation. We use the shorthand notation $\gamma^{\mu_1\cdots\mu_n} \equiv \gamma^{\mu_1}\ldots\gamma^{\mu_n}$ with no (anti)symmetrization. In the case of four fermion operators with charge conjugation we do not include for the moment structures with three gammas, which we plan to include in the near future.

We denote by $\texttt{iCPV} = \epsilon_{0123} \in \{1, -1\}$ with $\epsilon^{\alpha\beta\mu\nu}$ the Levi-Civita tensor, in such a way that for $D = 4$ e.g.

$$
\sigma^{\mu\nu}\epsilon_{\mu\nu\rho\sigma} = 2\texttt{iCPV}\,i\sigma_{\rho\sigma}\gamma_5 \,, \tag{122}
$$

with $\sigma^{\mu\nu} = \frac{i}{2}[\gamma^\mu, \gamma^\nu]$ and

$$
\gamma_5 = i\gamma^0\gamma^1\gamma^2\gamma^3 = \texttt{iCPV}\frac{i}{4!}\epsilon_{\mu\nu\alpha\beta}\gamma^\mu\gamma^\nu\gamma^\alpha\gamma^\beta \,. \tag{123}
$$

At any rate, dual tensors are defined by

$$
\tilde{X}_{\mu\nu} = \frac{1}{2}\epsilon_{\mu\nu\alpha\beta}X^{\mu\nu}, \quad \text{with} \quad X = G, W, B \,. \tag{124}
$$

We follow [27] for our definitions of evanescent operators, with the difference that we do not define them to be zero. Considering at most three gamma matrices, they read

$$
\begin{aligned}
E^{(2)}_{LR} &= P_L\gamma^{\mu\nu}P_L \otimes P_R\gamma_{\mu\nu}P_R = 4(1 + a_{\text{ev}}\epsilon)P_L \otimes P_R, , &(125)\\
E^{(2)}_{RL} &= P_R\gamma^{\mu\nu}P_R \otimes P_L\gamma_{\mu\nu}P_L = 4(1 + a'_{\text{ev}}\epsilon)P_R \otimes P_L, &(126)\\
E^{(3)}_{LL} &= P_R\gamma^{\mu\nu\lambda}P_L \otimes P_R\gamma_{\mu\nu\lambda}P_L = 4(4 - b_{\text{ev}}\epsilon)P_R\gamma^\mu P_L \otimes P_R\gamma_\mu P_L, &(127)\\
E^{(3)}_{RR} &= P_L\gamma^{\mu\nu\lambda}P_R \otimes P_L\gamma_{\mu\nu\lambda}P_R = 4(4 - b'_{\text{ev}}\epsilon)P_L\gamma^\mu P_R \otimes P_L\gamma_\mu P_R, &(128)\\
E^{(3)}_{LR} &= P_R\gamma^{\mu\nu\lambda}P_L \otimes P_L\gamma_{\mu\nu\lambda}P_R = 4(1 + c_{\text{ev}}\epsilon)P_R\gamma^\mu P_L \otimes P_L\gamma_\mu P_R, &(129)\\
E^{(3)}_{RL} &= P_L\gamma^{\mu\nu\lambda}P_R \otimes P_R\gamma_{\mu\nu\lambda}P_L = 4(1 + c'_{\text{ev}}\epsilon)P_L\gamma^\mu P_R \otimes P_R\gamma_\mu P_L, &(130)
\end{aligned}
$$

while

$$P_L\gamma^{\mu\nu}P_L \otimes P_L\gamma_{\mu\nu}P_L = (4-2\epsilon)P_L \otimes P_L - P_L\sigma^{\mu\nu}P_L \otimes P_L\sigma_{\mu\nu}P_L \,, \tag{131}$$

$$P_R\gamma^{\mu\nu}P_R \otimes P_R\gamma_{\mu\nu}P_R = (4-2\epsilon)P_R \otimes P_R - P_R\sigma^{\mu\nu}P_R \otimes P_R\sigma_{\mu\nu}P_R, \tag{132}$$

assuming the following basis of the space of Lorentz singlets and pseudo-singlets in $D = 4$

$$\begin{aligned}
\left\{\Gamma_1^i \otimes \Gamma_2^i \,|\, i=1,\ldots,10\right\} = &\left\{P_L \otimes P_L, P_R \otimes P_R, P_L \otimes P_R, P_R \otimes P_L, P_R\gamma^\mu P_L \otimes P_R\gamma_\mu P_L,\right.\\
&P_L\gamma^\mu P_R \otimes P_L\gamma_\mu P_R, P_R\gamma^\mu P_L \otimes P_L\gamma_\mu P_R, P_L\gamma^\mu P_R \otimes P_R\gamma_\mu P_L,\\
&\left. P_L\sigma^{\mu\nu}P_L \otimes P_L\sigma_{\mu\nu}P_L, P_R\sigma^{\mu\nu}P_R \otimes P_R\sigma_{\mu\nu}P_R\right\}. \tag{133}
\end{aligned}$$

In order to be consistent with flavour, we further assume that $a_{\text{ev}} = a'_{\text{ev}}, c_{\text{ev}} = c'_{\text{ev}}$. Therefore, at the end of the day the one-loop matching will be dependant on five free parameters: `iCPV, aEV, bEV, bpEV` and `cEV`. However, physical observables computed with the obtained dimension-6 EFT cannot depend on such parameters.

One final important remark should be made. In the model file `.fr` of the SMEFT used by `matchmakereft` to do the matching, the operator of the Warsaw basis $\mathcal{O}_{\ell equ}^{(3)}$ is replaced by $\mathcal{E}_{\ell equ}^{[2]}$ since `matchmakereft` and FORM use always $\gamma^{\mu\nu}$ instead of its antisymmetric version $\sigma^{\mu\nu}$. After doing the matching, such operator is reduced onto the one present in the Warsaw basis by the relation (we use $\alpha$, $\beta$ and $\gamma$ for the WCs of physical, redundant and evanescent operators, respectively)

$$\begin{aligned}
\left(\alpha_{\ell equ}^{(3)}\right)_{ijkl} = &-\left(\gamma_{\ell equ}^{[2]}\right)_{ijkl} - \frac{1}{8}\left(\gamma_{\ell uqe}\right)_{ilkj} - \left(1-\frac{\epsilon}{4}\right)\left(\gamma_{\ell uqe}^{[2]}\right)_{ilkj}\\
&+ \frac{1}{8}\left(\gamma_{ue\ell q}^c\right)_{ljik} + \left(1-\frac{\epsilon}{4}\right)\left(\gamma_{ue\ell q}^{c[2]}\right)_{ljik}. \tag{134}
\end{aligned}$$

Table 1: Physical and redundant bosonic operators.

| $X^3$ | | $X^2H^2$ | | $H^2D^4$ | |
|---|---|---|---|---|---|
| $\mathcal{O}_{3G}$ | $f^{ABC}G_\mu^{A\nu}G_\nu^{B\rho}G_\rho^{C\mu}$ | $\mathcal{O}_{HG}$ | $G_{\mu\nu}^A G^{A\mu\nu}(H^\dagger H)$ | $\mathcal{R}_{DH}$ | $(D_\mu D^\mu H)^\dagger(D_\nu D^\nu H)$ |
| $\mathcal{O}_{\widetilde{3G}}$ | $f^{ABC}\widetilde{G}_\mu^{A\nu}G_\nu^{B\rho}G_\rho^{C\mu}$ | $\mathcal{O}_{H\widetilde{G}}$ | $\widetilde{G}_{\mu\nu}^A G^{A\mu\nu}(H^\dagger H)$ | | $H^4D^2$ |
| $\mathcal{O}_{3W}$ | $\epsilon^{IJK}W_\mu^{I\nu}W_\nu^{J\rho}W_\rho^{K\mu}$ | $\mathcal{O}_{HW}$ | $W_{\mu\nu}^I W^{I\mu\nu}(H^\dagger H)$ | $\mathcal{O}_{H\Box}$ | $(H^\dagger H)\Box(H^\dagger H)$ |
| $\mathcal{O}_{\widetilde{3W}}$ | $\epsilon^{IJK}\widetilde{W}_\mu^{I\nu}W_\nu^{J\rho}W_\rho^{K\mu}$ | $\mathcal{O}_{H\widetilde{W}}$ | $\widetilde{W}_{\mu\nu}^I W^{I\mu\nu}(H^\dagger H)$ | $\mathcal{O}_{HD}$ | $(H^\dagger D^\mu H)^\dagger(H^\dagger D_\mu H)$ |
| | $X^2D^2$ | $\mathcal{O}_{HB}$ | $B_{\mu\nu}B^{\mu\nu}(H^\dagger H)$ | $\mathcal{R}'_{HD}$ | $(H^\dagger H)(D_\mu H)^\dagger(D^\mu H)$ |
| $\mathcal{R}_{2G}$ | $-\frac{1}{2}(D_\mu G^{A\mu\nu})(D^\rho G_{\rho\nu}^A)$ | $\mathcal{O}_{H\widetilde{B}}$ | $\widetilde{B}_{\mu\nu}B^{\mu\nu}(H^\dagger H)$ | $\mathcal{R}''_{HD}$ | $(H^\dagger H)D_\mu(H^\dagger i\overleftrightarrow{D}^\mu H)$ |
| $\mathcal{R}_{2W}$ | $-\frac{1}{2}(D_\mu W^{I\mu\nu})(D^\rho W_{\rho\nu}^I)$ | $\mathcal{O}_{HWB}$ | $W_{\mu\nu}^I B^{\mu\nu}(H^\dagger\sigma^I H)$ | | $H^6$ |
| $\mathcal{R}_{2B}$ | $-\frac{1}{2}(\partial_\mu B^{\mu\nu})(\partial^\rho B_{\rho\nu})$ | $\mathcal{O}_{H\widetilde{W}B}$ | $\widetilde{W}_{\mu\nu}^I B^{\mu\nu}(H^\dagger\sigma^I H)$ | $\mathcal{O}_H$ | $(H^\dagger H)^3$ |
| | | | $H^2XD^2$ | | |
| | | $\mathcal{R}_{WDH}$ | $D_\nu W^{I\mu\nu}(H^\dagger i\overleftrightarrow{D}_\mu^I H)$ | | |
| | | $\mathcal{R}_{BDH}$ | $\partial_\nu B^{\mu\nu}(H^\dagger i\overleftrightarrow{D}_\mu H)$ | | |

Table 2: Physical and redundant operators with two fermions.

| $\psi^2 D^3$ | | $\psi^2 XD$ | | $\psi^2 DH^2$ | |
|---|---|---|---|---|---|
| $\mathcal{R}_{qD}$ | $\frac{i}{2}\bar{q}\{D_\mu D^\mu,\slashed{D}\}q$ | $\mathcal{R}_{Gq}$ | $(\bar{q}T^A\gamma^\mu q)D^\nu G^A_{\mu\nu}$ | $\mathcal{O}^{(1)}_{Hq}$ | $(\bar{q}\gamma^\mu q)(H^\dagger i\overleftrightarrow{D}_\mu H)$ |
| $\mathcal{R}_{uD}$ | $\frac{i}{2}\bar{u}\{D_\mu D^\mu,\slashed{D}\}u$ | $\mathcal{R}'_{Gq}$ | $\frac{1}{2}(\bar{q}T^A\gamma^\mu i\overleftrightarrow{D}^\nu q)G^A_{\mu\nu}$ | $\mathcal{R}'^{(1)}_{Hq}$ | $(\bar{q}\,i\overleftrightarrow{\slashed{D}}\,q)(H^\dagger H)$ |
| $\mathcal{R}_{dD}$ | $\frac{i}{2}\bar{d}\{D_\mu D^\mu,\slashed{D}\}d$ | $\mathcal{R}'_{\tilde{G}q}$ | $\frac{1}{2}(\bar{q}T^A\gamma^\mu i\overleftrightarrow{D}^\nu q)\tilde{G}^A_{\mu\nu}$ | $\mathcal{R}''^{(1)}_{Hq}$ | $(\bar{q}\gamma^\mu q)\partial_\mu(H^\dagger H)$ |
| $\mathcal{R}_{\ell D}$ | $\frac{i}{2}\bar{\ell}\{D_\mu D^\mu,\slashed{D}\}\ell$ | $\mathcal{R}_{Wq}$ | $(\bar{q}\sigma^I\gamma^\mu q)D^\nu W^I_{\mu\nu}$ | $\mathcal{O}^{(3)}_{Hq}$ | $(\bar{q}\sigma^I\gamma^\mu q)(H^\dagger i\overleftrightarrow{D}^I_\mu H)$ |
| $\mathcal{R}_{eD}$ | $\frac{i}{2}\bar{e}\{D_\mu D^\mu,\slashed{D}\}e$ | $\mathcal{R}'_{Wq}$ | $\frac{1}{2}(\bar{q}\sigma^I\gamma^\mu i\overleftrightarrow{D}^\nu q)W^I_{\mu\nu}$ | $\mathcal{R}'^{(3)}_{Hq}$ | $(\bar{q}\,i\overleftrightarrow{\slashed{D}}^I q)(H^\dagger \sigma^I H)$ |
| **$\psi^2 HD^2$ + h.c.** | | $\mathcal{R}'_{\tilde{W}q}$ | $\frac{1}{2}(\bar{q}\sigma^I\gamma^\mu i\overleftrightarrow{D}^\nu q)\tilde{W}^I_{\mu\nu}$ | $\mathcal{R}''^{(3)}_{Hq}$ | $(\bar{q}\sigma^I\gamma^\mu q)D_\mu(H^\dagger \sigma^I H)$ |
| $\mathcal{R}_{uHD1}$ | $(\bar{q}u)D_\mu D^\mu \tilde{H}$ | $\mathcal{R}_{Bq}$ | $(\bar{q}\gamma^\mu q)\partial^\nu B_{\mu\nu}$ | $\mathcal{O}_{Hu}$ | $(\bar{u}\gamma^\mu u)(H^\dagger i\overleftrightarrow{D}_\mu H)$ |
| $\mathcal{R}_{uHD2}$ | $(\bar{q}\,i\sigma_{\mu\nu}D^\mu u)D^\nu \tilde{H}$ | $\mathcal{R}'_{Bq}$ | $\frac{1}{2}(\bar{q}\gamma^\mu i\overleftrightarrow{D}^\nu q)B_{\mu\nu}$ | $\mathcal{R}'_{Hu}$ | $(\bar{u}\,i\overleftrightarrow{\slashed{D}}\,u)(H^\dagger H)$ |
| $\mathcal{R}_{uHD3}$ | $(\bar{q}D_\mu D^\mu u)\tilde{H}$ | $\mathcal{R}'_{\tilde{B}q}$ | $\frac{1}{2}(\bar{q}\gamma^\mu i\overleftrightarrow{D}^\nu q)\tilde{B}_{\mu\nu}$ | $\mathcal{R}''_{Hu}$ | $(\bar{u}\gamma^\mu u)\partial_\mu(H^\dagger H)$ |
| $\mathcal{R}_{uHD4}$ | $(\bar{q}D_\mu u)D^\mu \tilde{H}$ | $\mathcal{R}_{Gu}$ | $(\bar{u}T^A\gamma^\mu u)D^\nu G^A_{\mu\nu}$ | $\mathcal{O}_{Hd}$ | $(\bar{d}\gamma^\mu d)(H^\dagger i\overleftrightarrow{D}_\mu H)$ |
| $\mathcal{R}_{dHD1}$ | $(\bar{q}d)D_\mu D^\mu H$ | $\mathcal{R}'_{Gu}$ | $\frac{1}{2}(\bar{u}T^A\gamma^\mu i\overleftrightarrow{D}^\nu u)G^A_{\mu\nu}$ | $\mathcal{R}'_{Hd}$ | $(\bar{d}\,i\overleftrightarrow{\slashed{D}}\,d)(H^\dagger H)$ |
| $\mathcal{R}_{dHD2}$ | $(\bar{q}\,i\sigma_{\mu\nu}D^\mu d)D^\nu H$ | $\mathcal{R}'_{\tilde{G}u}$ | $\frac{1}{2}(\bar{u}T^A\gamma^\mu i\overleftrightarrow{D}^\nu u)\tilde{G}^A_{\mu\nu}$ | $\mathcal{R}''_{Hd}$ | $(\bar{d}\gamma^\mu d)\partial_\mu(H^\dagger H)$ |
| $\mathcal{R}_{dHD3}$ | $(\bar{q}D_\mu D^\mu d)H$ | $\mathcal{R}_{Bu}$ | $(\bar{u}\gamma^\mu u)\partial^\nu B_{\mu\nu}$ | $\mathcal{O}_{Hud}$ | $(\bar{u}\gamma^\mu d)(\tilde{H}^\dagger iD_\mu H)$ |
| $\mathcal{R}_{dHD4}$ | $(\bar{q}D_\mu d)D^\mu H$ | $\mathcal{R}'_{Bu}$ | $\frac{1}{2}(\bar{u}\gamma^\mu i\overleftrightarrow{D}^\nu u)B_{\mu\nu}$ | $\mathcal{O}^{(1)}_{H\ell}$ | $(\bar{\ell}\gamma^\mu \ell)(H^\dagger i\overleftrightarrow{D}_\mu H)$ |
| $\mathcal{R}_{eHD1}$ | $(\bar{\ell}e)D_\mu D^\mu H$ | $\mathcal{R}'_{\tilde{B}u}$ | $\frac{1}{2}(\bar{u}\gamma^\mu i\overleftrightarrow{D}^\nu u)\tilde{B}_{\mu\nu}$ | $\mathcal{R}'^{(1)}_{H\ell}$ | $(\bar{\ell}\,i\overleftrightarrow{\slashed{D}}\,\ell)(H^\dagger H)$ |
| $\mathcal{R}_{eHD2}$ | $(\bar{\ell}\,i\sigma_{\mu\nu}D^\mu e)D^\nu H$ | $\mathcal{R}_{Gd}$ | $(\bar{d}T^A\gamma^\mu d)D^\nu G^A_{\mu\nu}$ | $\mathcal{R}''^{(1)}_{H\ell}$ | $(\bar{\ell}\gamma^\mu \ell)\partial_\mu(H^\dagger H)$ |
| $\mathcal{R}_{eHD3}$ | $(\bar{\ell}D_\mu D^\mu e)H$ | $\mathcal{R}'_{Gd}$ | $\frac{1}{2}(\bar{d}T^A\gamma^\mu i\overleftrightarrow{D}^\nu d)G^A_{\mu\nu}$ | $\mathcal{O}^{(3)}_{H\ell}$ | $(\bar{\ell}\sigma^I\gamma^\mu \ell)(H^\dagger i\overleftrightarrow{D}^I_\mu H)$ |
| $\mathcal{R}_{eHD4}$ | $(\bar{\ell}D_\mu e)D^\mu H$ | $\mathcal{R}'_{\tilde{G}d}$ | $\frac{1}{2}(\bar{d}T^A\gamma^\mu i\overleftrightarrow{D}^\nu d)\tilde{G}^A_{\mu\nu}$ | $\mathcal{R}'^{(3)}_{H\ell}$ | $(\bar{\ell}\,i\overleftrightarrow{\slashed{D}}^I \ell)(H^\dagger \sigma^I H)$ |
| **$\psi^2 XH$ + h.c.** | | $\mathcal{R}_{Bd}$ | $(\bar{d}\gamma^\mu d)\partial^\nu B_{\mu\nu}$ | $\mathcal{R}''^{(3)}_{H\ell}$ | $(\bar{\ell}\sigma^I\gamma^\mu \ell)D_\mu(H^\dagger \sigma^I H)$ |
| $\mathcal{O}_{uG}$ | $(\bar{q}T^A\sigma^{\mu\nu}u)\tilde{H}G^A_{\mu\nu}$ | $\mathcal{R}'_{Bd}$ | $\frac{1}{2}(\bar{d}\gamma^\mu i\overleftrightarrow{D}^\nu d)B_{\mu\nu}$ | $\mathcal{O}_{He}$ | $(\bar{e}\gamma^\mu e)(H^\dagger i\overleftrightarrow{D}_\mu H)$ |
| $\mathcal{O}_{uW}$ | $(\bar{q}\sigma^{\mu\nu}u)\sigma^I\tilde{H}W^I_{\mu\nu}$ | $\mathcal{R}'_{\tilde{B}d}$ | $\frac{1}{2}(\bar{d}\gamma^\mu i\overleftrightarrow{D}^\nu d)\tilde{B}_{\mu\nu}$ | $\mathcal{R}'_{He}$ | $(\bar{e}\,i\overleftrightarrow{\slashed{D}}\,e)(H^\dagger H)$ |
| $\mathcal{O}_{uB}$ | $(\bar{q}\sigma^{\mu\nu}u)\tilde{H}B_{\mu\nu}$ | $\mathcal{R}_{W\ell}$ | $(\bar{\ell}\sigma^I\gamma^\mu \ell)D^\nu W^I_{\mu\nu}$ | $\mathcal{R}''_{He}$ | $(\bar{e}\gamma^\mu e)\partial_\mu(H^\dagger H)$ |
| $\mathcal{O}_{dG}$ | $(\bar{q}T^A\sigma^{\mu\nu}d)HG^A_{\mu\nu}$ | $\mathcal{R}'_{W\ell}$ | $\frac{1}{2}(\bar{\ell}\sigma^I\gamma^\mu i\overleftrightarrow{D}^\nu \ell)W^I_{\mu\nu}$ | **$\psi^2 H^3$ + h.c.** | |
| $\mathcal{O}_{dW}$ | $(\bar{q}\sigma^{\mu\nu}d)\sigma^I HW^I_{\mu\nu}$ | $\mathcal{R}'_{\tilde{W}\ell}$ | $\frac{1}{2}(\bar{\ell}\sigma^I\gamma^\mu i\overleftrightarrow{D}^\nu \ell)\tilde{W}^I_{\mu\nu}$ | $\mathcal{O}_{uH}$ | $(H^\dagger H)\bar{q}\tilde{H}u$ |
| $\mathcal{O}_{dB}$ | $(\bar{q}\sigma^{\mu\nu}d)HB_{\mu\nu}$ | $\mathcal{R}_{B\ell}$ | $(\bar{\ell}\gamma^\mu \ell)\partial^\nu B_{\mu\nu}$ | $\mathcal{O}_{dH}$ | $(H^\dagger H)\bar{q}Hd$ |
| $\mathcal{O}_{eW}$ | $(\bar{\ell}\sigma^{\mu\nu}e)\sigma^I HW^I_{\mu\nu}$ | $\mathcal{R}'_{B\ell}$ | $\frac{1}{2}(\bar{\ell}\gamma^\mu i\overleftrightarrow{D}^\nu \ell)B_{\mu\nu}$ | $\mathcal{O}_{eH}$ | $(H^\dagger H)\bar{\ell}He$ |
| $\mathcal{O}_{eB}$ | $(\bar{\ell}\sigma^{\mu\nu}e)HB_{\mu\nu}$ | $\mathcal{R}'_{\tilde{B}\ell}$ | $\frac{1}{2}(\bar{\ell}\gamma^\mu i\overleftrightarrow{D}^\nu \ell)\tilde{B}_{\mu\nu}$ | | |
| | | $\mathcal{R}_{Be}$ | $(\bar{e}\gamma^\mu e)\partial^\nu B_{\mu\nu}$ | | |
| | | $\mathcal{R}'_{Be}$ | $\frac{1}{2}(\bar{e}\gamma^\mu i\overleftrightarrow{D}^\nu e)B_{\mu\nu}$ | | |
| | | $\mathcal{R}'_{\tilde{B}e}$ | $\frac{1}{2}(\bar{e}\gamma^\mu i\overleftrightarrow{D}^\nu e)\tilde{B}_{\mu\nu}$ | | |

Table 3: Baryon and lepton number conserving operators with four fermions.

| Four-quark | | Four-lepton | | Semileptonic | |
| --- | --- | --- | --- | --- | --- |
| $\mathcal{O}_{qq}^{(1)}$ | $(\bar{q}\gamma^\mu q)(\bar{q}\gamma_\mu q)$ | $\mathcal{O}_{\ell\ell}$ | $(\bar{\ell}\gamma^\mu \ell)(\bar{\ell}\gamma_\mu \ell)$ | $\mathcal{O}_{\ell q}^{(1)}$ | $(\bar{\ell}\gamma^\mu \ell)(\bar{q}\gamma_\mu q)$ |
| $\mathcal{O}_{qq}^{(3)}$ | $(\bar{q}\gamma^\mu \sigma^I q)(\bar{q}\gamma_\mu \sigma^I q)$ | $\mathcal{O}_{ee}$ | $(\bar{e}\gamma^\mu e)(\bar{e}\gamma_\mu e)$ | $\mathcal{O}_{\ell q}^{(3)}$ | $(\bar{\ell}\gamma^\mu \sigma^I \ell)(\bar{q}\gamma_\mu \sigma^I q)$ |
| $\mathcal{O}_{uu}$ | $(\bar{u}\gamma^\mu u)(\bar{u}\gamma_\mu u)$ | $\mathcal{O}_{\ell e}$ | $(\bar{\ell}\gamma^\mu \ell)(\bar{e}\gamma_\mu e)$ | $\mathcal{O}_{eu}$ | $(\bar{e}\gamma^\mu e)(\bar{u}\gamma_\mu u)$ |
| $\mathcal{O}_{dd}$ | $(\bar{d}\gamma^\mu d)(\bar{d}\gamma_\mu d)$ | | | $\mathcal{O}_{ed}$ | $(\bar{e}\gamma^\mu e)(\bar{d}\gamma_\mu d)$ |
| $\mathcal{O}_{ud}^{(1)}$ | $(\bar{u}\gamma^\mu u)(\bar{d}\gamma_\mu d)$ | | | $\mathcal{O}_{qe}$ | $(\bar{q}\gamma^\mu q)(\bar{e}\gamma_\mu e)$ |
| $\mathcal{O}_{ud}^{(8)}$ | $(\bar{u}\gamma^\mu T^A u)(\bar{d}\gamma_\mu T^A d)$ | | | $\mathcal{O}_{\ell u}$ | $(\bar{\ell}\gamma^\mu \ell)(\bar{u}\gamma_\mu u)$ |
| $\mathcal{O}_{qu}^{(1)}$ | $(\bar{q}\gamma^\mu q)(\bar{u}\gamma_\mu u)$ | | | $\mathcal{O}_{\ell d}$ | $(\bar{\ell}\gamma^\mu \ell)(\bar{d}\gamma_\mu d)$ |
| $\mathcal{O}_{qu}^{(8)}$ | $(\bar{q}\gamma^\mu T^A q)(\bar{u}\gamma_\mu T^A u)$ | | | $\mathcal{O}_{\ell edq}$ | $(\bar{\ell}e)(\bar{d}q)$ |
| $\mathcal{O}_{qd}^{(1)}$ | $(\bar{q}\gamma^\mu q)(\bar{d}\gamma_\mu d)$ | | | $\mathcal{O}_{\ell equ}^{(1)}$ | $(\bar{\ell}_r e)\epsilon_{rs}(\bar{q}_s u)$ |
| $\mathcal{O}_{qd}^{(8)}$ | $(\bar{q}\gamma^\mu T^A q)(\bar{d}\gamma_\mu T^A d)$ | | | $\mathcal{O}_{\ell equ}^{(3)}$ | $(\bar{\ell}_r \sigma^{\mu\nu}e)\epsilon_{rs}(\bar{q}_s \sigma_{\mu\nu}u)$ |
| $\mathcal{O}_{quqd}^{(1)}$ | $(\bar{q}_r u)\epsilon_{rs}(\bar{q}_s d)$ | | | | |
| $\mathcal{O}_{quqd}^{(8)}$ | $(\bar{q}_r T^A u)\epsilon_{rs}(\bar{q}_s T^A d)$ | | | | |

Table 4: Evanescent operators with two fermions.

| $\boldsymbol{\Psi^2 XH}$ + h.c. | | $\boldsymbol{\Psi^2 XD}$ | | | |
| --- | --- | --- | --- | --- | --- |
| $\mathcal{E}_{uG}$ | $\bar{q}T^A\sigma^{\mu\nu}u\widetilde{H}\widetilde{G}^A_{\mu\nu}$ | $\mathcal{E}_{Gq}$ | $\bar{q}T^A(\sigma^{\mu\nu}\gamma^\rho + \gamma^\rho\sigma^{\mu\nu})qD_\rho\widetilde{G}^A_{\mu\nu}$ | $\mathcal{E}_{Gd}$ | $\bar{d}T^A(\sigma^{\mu\nu}\gamma^\rho + \gamma^\rho\sigma^{\mu\nu})dD_\rho\widetilde{G}^A_{\mu\nu}$ |
| $\mathcal{E}_{uW}$ | $\bar{q}\sigma^I\sigma^{\mu\nu}u\widetilde{H}\widetilde{W}^I_{\mu\nu}$ | $\mathcal{E}'_{Gq}$ | $i\bar{q}(T^A\sigma^{\mu\nu}\slashed{D} - \overleftarrow{\slashed{D}}\sigma^{\mu\nu}T^A)qG^A_{\mu\nu}$ | $\mathcal{E}'_{Gd}$ | $i\bar{d}(T^A\sigma^{\mu\nu}\slashed{D} - \overleftarrow{\slashed{D}}\sigma^{\mu\nu}T^A)dG^A_{\mu\nu}$ |
| $\mathcal{E}_{uB}$ | $\bar{q}\sigma^{\mu\nu}u\widetilde{H}\widetilde{B}_{\mu\nu}$ | $\mathcal{E}'_{\widetilde{G}q}$ | $i\bar{q}(T^A\sigma^{\mu\nu}\slashed{D} - \overleftarrow{\slashed{D}}\sigma^{\mu\nu}T^A)q\widetilde{G}^A_{\mu\nu}$ | $\mathcal{E}'_{\widetilde{G}d}$ | $i\bar{d}(T^A\sigma^{\mu\nu}\slashed{D} - \overleftarrow{\slashed{D}}\sigma^{\mu\nu}T^A)d\widetilde{G}^A_{\mu\nu}$ |
| $\mathcal{E}_{dG}$ | $\bar{q}T^A\sigma^{\mu\nu}dH\widetilde{G}^A_{\mu\nu}$ | $\mathcal{E}_{Wq}$ | $\bar{q}\sigma^I(\sigma^{\mu\nu}\gamma^\rho + \gamma^\rho\sigma^{\mu\nu})qD_\rho\widetilde{W}^I_{\mu\nu}$ | $\mathcal{E}_{Bd}$ | $\bar{d}(\sigma^{\mu\nu}\gamma^\rho + \gamma^\rho\sigma^{\mu\nu})d\partial_\rho\widetilde{B}_{\mu\nu}$ |
| $\mathcal{E}_{dW}$ | $\bar{q}\sigma^I\sigma^{\mu\nu}dH\widetilde{W}^I_{\mu\nu}$ | $\mathcal{E}'_{Wq}$ | $i\bar{q}(\sigma^I\sigma^{\mu\nu}\slashed{D} - \overleftarrow{\slashed{D}}\sigma^{\mu\nu}\sigma^I)qW^I_{\mu\nu}$ | $\mathcal{E}'_{Bd}$ | $i\bar{d}(\sigma^{\mu\nu}\slashed{D} - \overleftarrow{\slashed{D}}\sigma^{\mu\nu})dB_{\mu\nu}$ |
| $\mathcal{E}_{dB}$ | $\bar{q}\sigma^{\mu\nu}dH\widetilde{B}_{\mu\nu}$ | $\mathcal{E}'_{\widetilde{W}q}$ | $i\bar{q}(\sigma^I\sigma^{\mu\nu}\slashed{D} - \overleftarrow{\slashed{D}}\sigma^{\mu\nu}\sigma^I)q\widetilde{W}^I_{\mu\nu}$ | $\mathcal{E}'_{\widetilde{B}d}$ | $i\bar{d}(\sigma^{\mu\nu}\slashed{D} - \overleftarrow{\slashed{D}}\sigma^{\mu\nu})d\widetilde{B}_{\mu\nu}$ |
| $\mathcal{E}_{eW}$ | $\bar{\ell}\sigma^I\sigma^{\mu\nu}eH\widetilde{W}^I_{\mu\nu}$ | $\mathcal{E}_{Bq}$ | $\bar{q}(\sigma^{\mu\nu}\gamma^\rho + \gamma^\rho\sigma^{\mu\nu})q\partial_\rho\widetilde{B}_{\mu\nu}$ | $\mathcal{E}_{W\ell}$ | $\bar{\ell}\sigma^I(\sigma^{\mu\nu}\gamma^\rho + \gamma^\rho\sigma^{\mu\nu})\ell D_\rho\widetilde{W}^I_{\mu\nu}$ |
| $\mathcal{E}_{eB}$ | $\bar{\ell}\sigma^{\mu\nu}eH\widetilde{B}_{\mu\nu}$ | $\mathcal{E}'_{Bq}$ | $i\bar{q}(\sigma^{\mu\nu}\slashed{D} - \overleftarrow{\slashed{D}}\sigma^{\mu\nu})qB_{\mu\nu}$ | $\mathcal{E}'_{W\ell}$ | $i\bar{\ell}(\sigma^I\sigma^{\mu\nu}\slashed{D} - \overleftarrow{\slashed{D}}\sigma^{\mu\nu}\sigma^I)\ell W^I_{\mu\nu}$ |
| $\boldsymbol{\psi^2 HD^2}$ + h.c. | | $\mathcal{E}'_{\widetilde{B}q}$ | $i\bar{q}(\sigma^{\mu\nu}\slashed{D} - \overleftarrow{\slashed{D}}\sigma^{\mu\nu})q\widetilde{B}_{\mu\nu}$ | $\mathcal{E}'_{\widetilde{W}\ell}$ | $i\bar{\ell}(\sigma^I\sigma^{\mu\nu}\slashed{D} - \overleftarrow{\slashed{D}}\sigma^{\mu\nu}\sigma^I)\ell\widetilde{W}^I_{\mu\nu}$ |
| $\mathcal{E}_{uH}$ | $\bar{q}\sigma^{\mu\nu}D^\rho u D^\sigma \widetilde{H}\epsilon_{\mu\nu\rho\sigma}$ | $\mathcal{E}_{Gu}$ | $\bar{u}T^A(\sigma^{\mu\nu}\gamma^\rho + \gamma^\rho\sigma^{\mu\nu})uD_\rho\widetilde{G}^A_{\mu\nu}$ | $\mathcal{E}_{B\ell}$ | $\bar{\ell}(\sigma^{\mu\nu}\gamma^\rho + \gamma^\rho\sigma^{\mu\nu})\ell\partial_\rho\widetilde{B}_{\mu\nu}$ |
| $\mathcal{E}_{dH}$ | $\bar{q}\sigma^{\mu\nu}D^\rho d D^\sigma H\epsilon_{\mu\nu\rho\sigma}$ | $\mathcal{E}'_{Gu}$ | $i\bar{u}(T^A\sigma^{\mu\nu}\slashed{D} - \overleftarrow{\slashed{D}}\sigma^{\mu\nu}T^A)uG^A_{\mu\nu}$ | $\mathcal{E}'_{B\ell}$ | $i\bar{\ell}(\sigma^{\mu\nu}\slashed{D} - \overleftarrow{\slashed{D}}\sigma^{\mu\nu})\ell B_{\mu\nu}$ |
| $\mathcal{E}_{eH}$ | $\bar{\ell}\sigma^{\mu\nu}D^\rho e D^\sigma H\epsilon_{\mu\nu\rho\sigma}$ | $\mathcal{E}'_{\widetilde{G}u}$ | $i\bar{u}(T^A\sigma^{\mu\nu}\slashed{D} - \overleftarrow{\slashed{D}}\sigma^{\mu\nu}T^A)u\widetilde{G}^A_{\mu\nu}$ | $\mathcal{E}'_{\widetilde{B}\ell}$ | $i\bar{\ell}(\sigma^{\mu\nu}\slashed{D} - \overleftarrow{\slashed{D}}\sigma^{\mu\nu})\ell\widetilde{B}_{\mu\nu}$ |
| | | $\mathcal{E}_{Bu}$ | $\bar{u}(\sigma^{\mu\nu}\gamma^\rho + \gamma^\rho\sigma^{\mu\nu})u\partial_\rho\widetilde{B}_{\mu\nu}$ | $\mathcal{E}_{Be}$ | $\bar{e}(\sigma^{\mu\nu}\gamma^\rho + \gamma^\rho\sigma^{\mu\nu})e\partial_\rho\widetilde{B}_{\mu\nu}$ |
| | | $\mathcal{E}'_{Bu}$ | $i\bar{u}(\sigma^{\mu\nu}\slashed{D} - \overleftarrow{\slashed{D}}\sigma^{\mu\nu})uB_{\mu\nu}$ | $\mathcal{E}'_{Be}$ | $i\bar{e}(\sigma^{\mu\nu}\slashed{D} - \overleftarrow{\slashed{D}}\sigma^{\mu\nu})eB_{\mu\nu}$ |
| | | $\mathcal{E}'_{\widetilde{B}u}$ | $i\bar{u}(\sigma^{\mu\nu}\slashed{D} - \overleftarrow{\slashed{D}}\sigma^{\mu\nu})u\widetilde{B}_{\mu\nu}$ | $\mathcal{E}'_{\widetilde{B}e}$ | $i\bar{e}(\sigma^{\mu\nu}\slashed{D} - \overleftarrow{\slashed{D}}\sigma^{\mu\nu})e\widetilde{B}_{\mu\nu}$ |

Table 5: Evanescent operators with four fermions involving only quarks.

| $\bar{L}R\bar{R}L$ | | $\bar{R}R\bar{R}R$ | | $\bar{L}L\bar{R}R$ | |
|---|---|---|---|---|---|
| $\mathcal{E}_{qu}$ | $(\bar{q}u)(\bar{u}q)$ | $\mathcal{E}_{uu}^{(8)}$ | $(\bar{u}\gamma^\mu T^A u)(\bar{u}\gamma_\mu T^A u)$ | $\mathcal{E}_{qu}^{[3]}$ | $(\bar{q}\gamma^{\mu\nu\rho}q)(\bar{u}\gamma_{\mu\nu\rho}u)$ |
| $\mathcal{E}_{qu}^{(8)}$ | $(\bar{q}T^A u)(\bar{u}T^A q)$ | $\mathcal{E}_{uu}^{[3]}$ | $(\bar{u}\gamma^{\mu\nu\rho}u)(\bar{u}\gamma_{\mu\nu\rho}u)$ | $\mathcal{E}_{qu}^{[3](8)}$ | $(\bar{q}\gamma^{\mu\nu\rho}T^A q)(\bar{u}\gamma_{\mu\nu\rho}T^A u)$ |
| $\mathcal{E}_{qd}$ | $(\bar{q}d)(\bar{d}q)$ | $\mathcal{E}_{uu}^{[3](8)}$ | $(\bar{u}\gamma^{\mu\nu\rho}T^A u)(\bar{u}\gamma_{\mu\nu\rho}T^A u)$ | $\mathcal{E}_{qd}^{[3]}$ | $(\bar{q}\gamma^{\mu\nu\rho}q)(\bar{d}\gamma_{\mu\nu\rho}d)$ |
| $\mathcal{E}_{qd}^{(8)}$ | $(\bar{q}T^A d)(\bar{d}T^A q)$ | $\mathcal{E}_{dd}^{(8)}$ | $(\bar{d}\gamma^\mu T^A d)(\bar{d}\gamma_\mu T^A d)$ | $\mathcal{E}_{qd}^{[3](8)}$ | $(\bar{q}\gamma^{\mu\nu\rho}T^A q)(\bar{d}\gamma_{\mu\nu\rho}T^A d)$ |
| $\mathcal{E}_{qu}^{[2]}$ | $(\bar{q}\gamma^{\mu\nu}u)(\bar{u}\gamma_{\mu\nu}q)$ | $\mathcal{E}_{dd}^{[3]}$ | $(\bar{d}\gamma^{\mu\nu\rho}d)(\bar{d}\gamma_{\mu\nu\rho}d)$ | | $\bar{L}L\bar{L}L$ |
| $\mathcal{E}_{qu}^{[2](8)}$ | $(\bar{q}\gamma^{\mu\nu}T^A u)(\bar{u}\gamma_{\mu\nu}T^A q)$ | $\mathcal{E}_{dd}^{[3](8)}$ | $(\bar{d}\gamma^{\mu\nu\rho}T^A d)(\bar{d}\gamma_{\mu\nu\rho}T^A d)$ | $\mathcal{E}_{qq}^{(8)}$ | $(\bar{q}\gamma^\mu T^A q)(\bar{q}\gamma_\mu T^A q)$ |
| $\mathcal{E}_{qd}^{[2]}$ | $(\bar{q}\gamma^{\mu\nu}d)(\bar{d}\gamma_{\mu\nu}q)$ | $\mathcal{E}_{ud}$ | $(\bar{u}\gamma^\mu d)(\bar{d}\gamma_\mu u)$ | $\mathcal{E}_{qq}^{(3,8)}$ | $(\bar{q}\gamma^\mu \sigma^I T^A q)(\bar{q}\gamma_\mu \sigma^I T^A q)$ |
| $\mathcal{E}_{qd}^{[2](8)}$ | $(\bar{q}\gamma^{\mu\nu}T^A d)(\bar{d}\gamma_{\mu\nu}T^A q)$ | $\mathcal{E}_{ud}^{(8)}$ | $(\bar{u}\gamma^\mu T^A d)(\bar{d}\gamma_\mu T^A u)$ | $\mathcal{E}_{qq}^{[3](1)}$ | $(\bar{q}\gamma^{\mu\nu\rho}q)(\bar{q}\gamma_{\mu\nu\rho}q)$ |
| | $\bar{L}R\bar{L}R$ | $\mathcal{E}_{ud}^{[3]}$ | $(\bar{u}\gamma^{\mu\nu\rho}d)(\bar{d}\gamma_{\mu\nu\rho}u)$ | $\mathcal{E}_{qq}^{[3](3)}$ | $(\bar{q}\gamma^{\mu\nu\rho}\sigma^I q)(\bar{q}\gamma_{\mu\nu\rho}\sigma^I q)$ |
| $\mathcal{E}_{quqd}^{[2]}$ | $(\bar{q}_r\gamma^{\mu\nu}u)\epsilon_{rs}(\bar{q}_s\gamma_{\mu\nu}d)$ | $\mathcal{E}_{ud}^{[3](8)}$ | $(\bar{u}\gamma^{\mu\nu\rho}T^A d)(\bar{d}\gamma_{\mu\nu\rho}T^A u)$ | $\mathcal{E}_{qq}^{[3](8)}$ | $(\bar{q}\gamma^{\mu\nu\rho}T^A q)(\bar{q}\gamma_{\mu\nu\rho}T^A q)$ |
| $\mathcal{E}_{quqd}^{[2](8)}$ | $(\bar{q}_r\gamma^{\mu\nu}T^A u)\epsilon_{rs}(\bar{q}_s\gamma_{\mu\nu}T^A d)$ | $\mathcal{E}_{ud}'^{[3]}$ | $(\bar{u}\gamma^{\mu\nu\rho}u)(\bar{d}\gamma_{\mu\nu\rho}d)$ | $\mathcal{E}_{qq}^{[3](3,8)}$ | $(\bar{q}\gamma^{\mu\nu\rho}\sigma^I T^A q)(\bar{q}\gamma_{\mu\nu\rho}\sigma^I T^A q)$ |
| | | $\mathcal{E}_{ud}'^{[3](8)}$ | $(\bar{u}\gamma^{\mu\nu\rho}T^A u)(\bar{d}\gamma_{\mu\nu\rho}T^A d)$ | | |

Table 6: Semileptonic four-fermion evanescent operators. We use the shorthand notation $\gamma^{\mu_1\cdots\mu_n} \equiv \gamma^{\mu_1}\ldots\gamma^{\mu_n}$ with no (anti)symmetrization.

| $\bar{L}R\bar{R}L$ | | $\bar{R}R\bar{R}R$ | | $\bar{L}L\bar{R}R$ | |
|---|---|---|---|---|---|
| $\mathcal{E}_{\ell u}$ | $(\bar{\ell}u)(\bar{u}\ell)$ | $\mathcal{E}_{eu}$ | $(\bar{e}\gamma^\mu u)(\bar{u}\gamma_\mu e)$ | $\mathcal{E}_{\ell qde}$ | $(\bar{\ell}\gamma^\mu q)(\bar{d}\gamma_\mu e)$ |
| $\mathcal{E}_{\ell d}$ | $(\bar{\ell}d)(\bar{d}\ell)$ | $\mathcal{E}_{ed}$ | $(\bar{e}\gamma^\mu d)(\bar{d}\gamma_\mu e)$ | $\mathcal{E}_{\ell u}^{[3]}$ | $(\bar{\ell}\gamma^{\mu\nu\rho}\ell)(\bar{u}\gamma_{\mu\nu\rho}u)$ |
| $\mathcal{E}_{qe}$ | $(\bar{q}e)(\bar{e}q)$ | $\mathcal{E}_{eu}^{[3]}$ | $(\bar{e}\gamma^{\mu\nu\rho}u)(\bar{u}\gamma_{\mu\nu\rho}e)$ | $\mathcal{E}_{\ell d}^{[3]}$ | $(\bar{\ell}\gamma^{\mu\nu\rho}\ell)(\bar{d}\gamma_{\mu\nu\rho}d)$ |
| $\mathcal{E}_{\ell edq}^{[2]}$ | $(\bar{\ell}\gamma^{\mu\nu}e)(\bar{d}\gamma_{\mu\nu}q)$ | $\mathcal{E}_{ed}^{[3]}$ | $(\bar{e}\gamma^{\mu\nu\rho}d)(\bar{d}\gamma_{\mu\nu\rho}e)$ | $\mathcal{E}_{qe}^{[3]}$ | $(\bar{q}\gamma^{\mu\nu\rho}q)(\bar{e}\gamma_{\mu\nu\rho}e)$ |
| $\mathcal{E}_{\ell u}^{[2]}$ | $(\bar{\ell}\gamma^{\mu\nu}u)(\bar{u}\gamma_{\mu\nu}\ell)$ | $\mathcal{E}_{eu}'^{[3]}$ | $(\bar{e}\gamma^{\mu\nu\rho}e)(\bar{u}\gamma_{\mu\nu\rho}u)$ | $\mathcal{E}_{\ell qde}^{[3]}$ | $(\bar{\ell}\gamma^{\mu\nu\rho}q)(\bar{d}\gamma_{\mu\nu\rho}e)$ |
| $\mathcal{E}_{\ell d}^{[2]}$ | $(\bar{\ell}\gamma^{\mu\nu}d)(\bar{d}\gamma_{\mu\nu}\ell)$ | $\mathcal{E}_{ed}'^{[3]}$ | $(\bar{e}\gamma^{\mu\nu\rho}e)(\bar{d}\gamma_{\mu\nu\rho}d)$ | | $\bar{L}L\bar{L}L$ |
| $\mathcal{E}_{qe}^{[2]}$ | $(\bar{q}\gamma^{\mu\nu}e)(\bar{e}\gamma_{\mu\nu}q)$ | | | $\mathcal{E}_{\ell q}$ | $(\bar{\ell}\gamma^\mu q)(\bar{q}\gamma_\mu \ell)$ |
| | $\bar{L}R\bar{L}R$ | | | $\mathcal{E}_{\ell q}^{(3)}$ | $(\bar{\ell}\gamma^\mu \sigma^I q)(\bar{q}\gamma_\mu \sigma^I \ell)$ |
| $\mathcal{E}_{\ell equ}^{[2]}$ | $(\bar{\ell}_r\gamma^{\mu\nu}e)\epsilon_{rs}(\bar{q}_s\gamma_{\mu\nu}u)$ | | | $\mathcal{E}_{\ell q}^{[3]}$ | $(\bar{\ell}\gamma^{\mu\nu\rho}q)(\bar{q}\gamma_{\mu\nu\rho}\ell)$ |
| $\mathcal{E}_{\ell uqe}$ | $(\bar{\ell}_r u)\epsilon_{rs}(\bar{q}_s e)$ | | | $\mathcal{E}_{\ell q}^{[3](3)}$ | $(\bar{\ell}\gamma^{\mu\nu\rho}\sigma^I q)(\bar{q}\gamma_{\mu\nu\rho}\sigma^I \ell)$ |
| $\mathcal{E}_{\ell uqe}^{[2]}$ | $(\bar{\ell}_r\gamma^{\mu\nu}u)\epsilon_{rs}(\bar{q}_s\gamma_{\mu\nu}e)$ | | | $\mathcal{E}_{\ell q}'^{[3]}$ | $(\bar{\ell}\gamma^{\mu\nu\rho}\ell)(\bar{q}\gamma_{\mu\nu\rho}q)$ |
| | | | | $\mathcal{E}_{\ell q}'^{[3](3)}$ | $(\bar{\ell}\gamma^{\mu\nu\rho}\sigma^I \ell)(\bar{q}\gamma_{\mu\nu\rho}\sigma^I q)$ |

Table 7: Leptonic four-fermion evanescent operators. We use the shorthand notation $\gamma^{\mu_1\cdots\mu_n} \equiv \gamma^{\mu_1}\ldots\gamma^{\mu_n}$ with no (anti)symmetrization.

| $\bar{R}R\bar{R}R$ | | $\bar{L}L\bar{L}L$ | | $\bar{L}L\bar{R}R$ | |
|---|---|---|---|---|---|
| $\mathcal{E}_{ee}^{[3]}$ | $(\bar{e}\gamma^{\mu\nu\rho}e)(\bar{e}\gamma_{\mu\nu\rho}e)$ | $\mathcal{E}_{\ell\ell}^{(3)}$ | $(\bar{\ell}\gamma^\mu \sigma^I \ell)(\bar{\ell}\gamma_\mu \sigma^I \ell)$ | $\mathcal{E}_{\ell e}^{[3]}$ | $(\bar{\ell}\gamma^{\mu\nu\rho}\ell)(\bar{e}\gamma_{\mu\nu\rho}e)$ |
| | $\bar{L}R\bar{R}L$ | $\mathcal{E}_{\ell\ell}^{[3]}$ | $(\bar{\ell}\gamma^{\mu\nu\rho}\ell)(\bar{\ell}\gamma_{\mu\nu\rho}\ell)$ | | |
| $\mathcal{E}_{\ell e}$ | $(\bar{\ell}e)(\bar{e}\ell)$ | $\mathcal{E}_{\ell\ell}^{[3](3)}$ | $(\bar{\ell}\gamma^{\mu\nu\rho}\sigma^I \ell)(\bar{\ell}\gamma_{\mu\nu\rho}\sigma^I \ell)$ | | |
| $\mathcal{E}_{\ell e}^{[2]}$ | $(\bar{\ell}\gamma^{\mu\nu}e)(\bar{e}\gamma_{\mu\nu}\ell)$ | | | | |

Table 8: Evanescent operators with four fermions involving only quarks and featuring charge conjugation. We use the shorthand notation $\gamma^{\mu_1\cdots\mu_n} \equiv \gamma^{\mu_1}\dots\gamma^{\mu_n}$ with no (anti)symmetrization.

| $\bar{L}^c L \bar{L} L^c$ | | $\bar{R}^c R \bar{R} R^c$ | | $\bar{L}^c R \bar{R} L^c$ | |
|---|---|---|---|---|---|
| $\mathcal{E}^c_{qq}$ | $(\overline{q^c}_{ar} q_{bs})(\bar{q}_{bs} q^c_{ar})$ | $\mathcal{E}^c_{uu}$ | $(\overline{u^c}_a u_b)(\bar{u}_b u^c_a)$ | $\mathcal{E}^c_{qu}$ | $(\overline{q^c}_a \gamma^\mu u_b)(\bar{u}_b \gamma_\mu q^c_a)$ |
| $\mathcal{E}^{c\,\prime}_{qq}$ | $(\overline{q^c}_{ar} q_{bs})(\bar{q}_{as} q^c_{br})$ | $\mathcal{E}^c_{dd}$ | $(\overline{d^c}_a d_b)(\bar{d}_b d^c_a)$ | $\mathcal{E}^c_{qd}$ | $(\overline{q^c}_a \gamma^\mu d_b)(\bar{d}_b \gamma_\mu q^c_a)$ |
| $\mathcal{E}^{c\,[2]}_{qq}$ | $(\overline{q^c}_{ar} \gamma^{\mu\nu} q_{bs})(\bar{q}_{bs} \gamma_{\mu\nu} q^c_{ar})$ | $\mathcal{E}^c_{ud}$ | $(\overline{u^c}_a d_b)(\bar{d}_b u^c_a)$ | $\mathcal{E}^{c\,\prime}_{qu}$ | $(\overline{q^c}_a \gamma^\mu u_b)(\bar{u}_a \gamma_\mu q^c_b)$ |
| $\mathcal{E}^{c\,\prime[2]}_{qq}$ | $(\overline{q^c}_{ar} \gamma^{\mu\nu} q_{bs})(\bar{q}_{as} \gamma_{\mu\nu} q^c_{br})$ | $\mathcal{E}^{c\,\prime}_{ud}$ | $(\overline{u^c}_a d_b)(\bar{d}_a u^c_b)$ | $\mathcal{E}^{c\,\prime}_{qd}$ | $(\overline{q^c}_a \gamma^\mu d_b)(\bar{d}_a \gamma_\mu q^c_b)$ |
| $\bar{R}^c R \bar{L} L^c$ | | $\mathcal{E}^{c\,[2]}_{uu}$ | $(\overline{u^c}_a \gamma^{\mu\nu} u_b)(\bar{u}_b \gamma_{\mu\nu} u^c_a)$ | $\mathcal{E}^{c\,[3]}_{qu}$ | $(\overline{q^c}_a \gamma^{\mu\nu\rho} u_b)(\bar{u}_b \gamma_{\mu\nu\rho} q^c_a)$ |
| $\mathcal{E}^c_{udqq}$ | $(\overline{u^c}_a d_b)(\bar{q}_{br} \epsilon_{rs} q^c_{as})$ | $\mathcal{E}^{c\,[2]}_{dd}$ | $(\overline{d^c}_a \gamma^{\mu\nu} d_b)(\bar{d}_b \gamma_{\mu\nu} d^c_a)$ | $\mathcal{E}^{c\,[3]}_{qd}$ | $(\overline{q^c}_a \gamma^{\mu\nu\rho} d_b)(\bar{d}_b \gamma_{\mu\nu\rho} q^c_a)$ |
| $\mathcal{E}^{c\,[2]}_{udqq}$ | $(\overline{u^c}_a \gamma^{\mu\nu} d_b)(\bar{q}_{br} \epsilon_{rs} \gamma_{\mu\nu} q^c_{as})$ | $\mathcal{E}^{c\,[2]}_{ud}$ | $(\overline{u^c}_a \gamma^{\mu\nu} d_b)(\bar{d}_b \gamma_{\mu\nu} u^c_a)$ | $\mathcal{E}^{c\,\prime[3]}_{qu}$ | $(\overline{q^c}_a \gamma^{\mu\nu\rho} u_b)(\bar{u}_a \gamma_{\mu\nu\rho} q^c_b)$ |
| | | $\mathcal{E}^{c\,\prime[2]}_{ud}$ | $(\overline{u^c}_a \gamma^{\mu\nu} d_b)(\bar{d}_a \gamma_{\mu\nu} u^c_b)$ | $\mathcal{E}^{c\,\prime[3]}_{qd}$ | $(\overline{q^c}_a \gamma^{\mu\nu\rho} d_b)(\bar{d}_a \gamma_{\mu\nu\rho} q^c_b)$ |

Table 9: Semileptonic and leptonic evanescent operators with four fermions featuring charge conjugation. We use the shorthand notation $\gamma^{\mu_1\cdots\mu_n} \equiv \gamma^{\mu_1}\dots\gamma^{\mu_n}$ with no (anti)symmetrization.

| $\bar{L}^c L \bar{L} L^c$ | | $\bar{R}^c R \bar{R} R^c$ | | $\bar{L}^c R \bar{R} L^c$ | |
|---|---|---|---|---|---|
| $\mathcal{E}^c_{\ell\ell}$ | $(\overline{\ell^c}_r \ell_s)(\bar{\ell}_s \ell^c_r)$ | $\mathcal{E}^c_{ee}$ | $(\overline{e^c} e)(\bar{e} e^c)$ | $\mathcal{E}^c_{\ell e}$ | $(\overline{\ell^c} \gamma^\mu e)(\bar{e} \gamma_\mu \ell^c)$ |
| $\mathcal{E}^c_{q\ell}$ | $(\overline{q^c}_r \ell_s)(\bar{\ell}_s q^c_r)$ | $\mathcal{E}^c_{eu}$ | $(\overline{e^c} u)(\bar{u} e^c)$ | $\mathcal{E}^c_{qe}$ | $(\overline{q^c} \gamma^\mu e)(\bar{e} \gamma_\mu q^c)$ |
| $\mathcal{E}^{c\,\prime}_{q\ell}$ | $(\overline{q^c}_r \ell_s)(\bar{q}_r \ell^c_s)$ | $\mathcal{E}^c_{ed}$ | $(\overline{e^c} d)(\bar{d} e^c)$ | $\mathcal{E}^c_{\ell u}$ | $(\overline{\ell^c} \gamma^\mu u)(\bar{u} \gamma_\mu \ell^c)$ |
| $\mathcal{E}^{c\,[2]}_{\ell\ell}$ | $(\overline{\ell^c}_r \gamma^{\mu\nu} \ell_s)(\bar{\ell}_s \gamma_{\mu\nu} q^c_r)$ | $\mathcal{E}^{c\,[2]}_{ee}$ | $(\overline{e^c} \gamma^{\mu\nu} e)(\bar{e} \gamma_{\mu\nu} e^c)$ | $\mathcal{E}^c_{\ell d}$ | $(\overline{\ell^c} \gamma^\mu d)(\bar{d} \gamma_\mu \ell^c)$ |
| $\mathcal{E}^{c\,[2]}_{q\ell}$ | $(\overline{q^c}_r \gamma^{\mu\nu} \ell_s)(\bar{q}_s \gamma_{\mu\nu} q^c_r)$ | $\mathcal{E}^{c\,[2]}_{eu}$ | $(\overline{e^c} \gamma^{\mu\nu} u)(\bar{u} \gamma_{\mu\nu} e^c)$ | $\mathcal{E}^c_{qed\ell}$ | $(\overline{q^c} \gamma^\mu e)(\bar{d} \gamma_\mu \ell^c)$ |
| $\mathcal{E}^{c\,\prime[2]}_{q\ell}$ | $(\overline{q^c}_r \gamma^{\mu\nu} \ell_s)(\bar{q}_r \gamma_{\mu\nu} \ell^c_s)$ | $\mathcal{E}^{c\,[2]}_{ed}$ | $(\overline{e^c} \gamma^{\mu\nu} d)(\bar{d} \gamma_{\mu\nu} e^c)$ | $\mathcal{E}^{c\,[3]}_{\ell e}$ | $(\overline{\ell^c} \gamma^{\mu\nu\rho} e)(\bar{e} \gamma_{\mu\nu\rho} \ell^c)$ |
| $\bar{R}^c R \bar{L} L^c$ | | | | $\mathcal{E}^{c\,[3]}_{qe}$ | $(\overline{q^c} \gamma^{\mu\nu\rho} e)(\bar{e} \gamma_{\mu\nu\rho} q^c)$ |
| $\mathcal{E}^c_{ue\ell q}$ | $(\overline{u^c} e)(\bar{\ell}_r \epsilon_{rs} q^c_s)$ | | | $\mathcal{E}^{c\,[3]}_{\ell u}$ | $(\overline{\ell^c} \gamma^{\mu\nu\rho} u)(\bar{u} \gamma_{\mu\nu\rho} \ell^c)$ |
| $\mathcal{E}^{c\,[2]}_{ue\ell q}$ | $(\overline{u^c} \gamma^{\mu\nu} e)(\bar{\ell}_r \gamma_{\mu\nu} \epsilon_{rs} q^c_s)$ | | | $\mathcal{E}^{c\,[3]}_{\ell d}$ | $(\overline{\ell^c} \gamma^{\mu\nu\rho} d)(\bar{d} \gamma_{\mu\nu\rho} \ell^c)$ |
| | | | | $\mathcal{E}^{c\,[3]}_{qed\ell}$ | $(\overline{q^c} \gamma^{\mu\nu\rho} e)(\bar{d} \gamma_{\mu\nu\rho} \ell^c)$ |

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
