# Peer review of "Matchmakereft: automated tree-level and one-loop matching"

_SciPost Physics, doi:SciPost Phys. 12, 198 (2022)_

## Round 1 · Referee Report · Anonymous (Referee 1) · 2022-3-8

Strengths

1 - The authors present a very powerful tool to perform matching up to the one-loop level.

2 - Several examples are provided and thoroughly explained.

3 - Results obtained with Matchmaker have been compared with previous literature and the authors are very honest about (minor) disagreements found.

Weaknesses

1 - Information on how to use the outputs provided by Matchmakereft for debugging could be extended.

Report

The authors introduce an automated tool to compute matching onto effective field theories up to the one-loop level. The manuscript is well written and provides future users of Matchmakereft with a comprehensive manual. The python package Matchmakereft can easily be downloaded using conda or pip and the installation process as well as dependencies are well explained. Examples in the manual and those provided with the package make it easy to understand the programme's syntax.

I highly recommend the publication of the manuscript in SciPost physics after the following (mostly editorial) changes have been made:

Requested changes

1 - Downloading the programme via conda currently only seems to work on linux systems. This is most probably because only the path https://conda.anaconda.org/matchmakers/linux-64 exists and there is no */matchmakers/osx-64 equivalent. This should be fixed or explained in the manuscript.

2 - 'ee[x__]' could be added to the list of protected words.

3 - The structure and content of the file MatchingProblems.dat is currently not explained. I encourage the authors to provide more information on this file so that it can be used for debugging. On their GitLab website, the authors write that the file lists amplitudes "admittedly in a very cryptic way". So I also encourage the authors to make this output more user friendly. An example output with explanations could be added to the manual.

4 - The vector-like lepton example model file runs with error messages due to the definition of gamma5 in d dimensions. It would be good to point this out explicitly in sec. 6.2.

5 - There is little information provided on the models which come with the Matchmakereft installation. It would be good to add an explanation of these files and how to run them. For instance, it would be good to explicitly mention that the file UnbrokenSM_BFM.fr has to be loaded alongside the BSM models.

6 - When running the minimal example in appendix C with the sample code provided, most but not all of the Wilson coefficients agree with those in Eqs. (79)-(93): - Eq. (80): The second term is not present when running the sample code. - Eq. (81): The first two terms are not present when running the sample code. - Eq. (84): The "1+" seems to be the tree-level contribution but only the one-loop pieces are listed here according to the text. - Eq. (92): Pieces suppressed by eight powers of the heavy scale MH^-8 are not listed here. If these are removed because kappa^4 mL^4/MH^8 ~ mL^4/MH^4 while all other pieces are only suppressed by MH^-2, it would be good to comment on this.

7 - typos and other minor issues: - The abbreviation BFM is used in the abstract without introducing it. - It would be good to use either flavor or flavour consistently. - p.2 last line: "at which experiment is performed" -> "at which experiments are performed" or "at which the experiment is performed" - p.3, first paragraph: "references there in" -> "references therein" - p.6, third line of sec 2.3, missing ED: "is perform" -> "is performed" - p.6, sec. 2.4, missing word: "Matchmaker then uses Mathematica [to] match" - p.7, sec. 3.1, first bullet point, additional words: "expects the Lagrangian of the to be defined" (remove "of the") - p.8, section 3.2, first paragraph: "wigh" -> "with" - p.15, first line of sec. 4.2, missing word: "provides an interactive experience to [the] user" - p.26, A.3: I recommend to replace "(s)he" with the gender neutral "they" - p.28, top paragraph: The W in "Wilson coefficients" is capitalised elsewhere in the paper. A lowercase W is used here three times. - p.32, missing S: "Feynrule" -> "FeynRules" (also apart from appendix C a texttt environment was used for FeynRules).

  • validity: top
  • significance: top
  • originality: top
  • clarity: top
  • formatting: excellent
  • grammar: excellent

Author:  Jose Santiago  on 2022-05-23  [id 2506]

(in reply to Report 1 on 2022-03-08)
Category:
answer to question
correction

We would like to thank the referee for the thorough report and suggestions. We think all the points raised by the referee are important and we have tried to address all of them. Before detailing how we have done so we would like to emphasize that the main purpose of this article is to introduce matchmakereft as a very useful tool to perform one-loop matching calculations (including general anomalous dimensions). There are issues that are relevant for the tool itself in terms of what it does and how it does it and other issues that are entirely dependent on how the user defines the corresponding models. We have tried to explain how the user should define the models properly in order to obtain the correct result but the main emphasis of the paper is on the former issues (the ones entirely dependent on matchmakereft). Two prime examples are heavy vector bosons and evanescent operators. It is up to the user to define the corresponding models (or reduction in the case of evanescent operators) correctly. Matchmakereft is equipped to perform the calculations correctly if the models are defined properly. Furthermore, matchmakereft is a very much alive code that keeps being maintained and expanded in capabilities by us (as of today we are currently on version 1.0.10, with some interesting improvements with respect to the original version as briefly discussed in the note added section of our revised version. Thus, some features that are not available in the current version are likely to be developed in future versions.

Let us now go to the details of how we have addressed the points raised by the referee:

  1. We have fixed the problem which indeed had to do with the specification of computer system in the conda installation.

  2. We have added ee[x___] to the list of protected words as suggested.

  3. We have added a more detailed discussion about the content of MatchingProblems.dat in section 4.1, at the end of the discussion of the command match_model_to_eft. Indeed we plan to provide a more user friendly output of the problems encountered, including gauge information, but this will be part of future developments of the tool.

  4. Indeed, we have added a sentence explaining the situation at the beginning of section 6.2 (right after the listing in page 17).

  5. We have added an explicit comment in page 13, when the command create_model is defined.

  6. We have eliminated the terms spotted by the referee in eqs. 80, 81, 84, since they were indeed referring to the result in the physical basis and not in the Green, as intended and stated in the text. Regarding eq.92, terms suppressed by (mL/MH )4 are indeed dropped, and we now indicate this explicitly immediately above eq.83 and eq.91.

  7. We have fixed all the typos spotted by the referee.

---

## Round 1 · Referee Report · Anonymous (Referee 2) · 2022-3-30

Strengths

1- The paper presents a very powerful tool, automating a class of highly technical calculations. Matchmakereft will certainly be very useful for the high energy physics community, streamlining the process of searching for heavy new physics via EFT techniques.

2- The explanations of the calculations are very clear, with an appropriate amount of examples. The code examples are also well documented, and I was able to follow them all without any problems.

3- In presenting their tool, the authors have showcased several examples and validated the results against existing calculations in the literature, some performed by the authors themselves and others by independent groups. The small differences are openly discussed and hopefully, the results of the exchanges with the corresponding authors can be published in the final version of the paper or in a short update in the future.

4- Several new results are also presented as by-products of this work, that are self-contained and merit appreciation. These include the extension of the Green's basis with the evasescent operators, the capacity to compute renormalisation group equations of generic theories, and the possibility to use matchmakereft to translate between generic bases of dimension-6 operators.

Weaknesses

1- In general, there is a significant amount of assumed knowledge required to be able to digest this paper. While this is normal for academic journals, it is mostly a question concerning the intended audience. To me, the level of detail is appropriate for experts in the field, who have already performed one-loop matching and RGE calculations by hand, as well as having had some exposure to the background field method. If the authors would like to expand their readership to EFT enthusiasts, who perhaps do not have as much experience on the one-loop side, for example. I think some sections would benefit from a small amount of additional detail and/or useful references.

2- On this topic, the assumed technical knowledge of at least Mathematica & FeynRules is also significant. Although this is, of course, unavoidable since the program makes heavy use of these two codes, it may be worthwhile to add a disclaimer of sorts, that using matchmaker requires a working knowledge of both, and preferably some experience in creating models in FeynRules.

Report

Dear Editors,

In manuscript 202201_00006, the authors present MatchMakerEFT (MM), an automated tool for matching renormalizable models of heavy new physics to low energy EFTs by integrating out the heavy states at the one-loop level. As mentioned in my comments on the strength of the paper, the tool is extremely powerful, making use of state-of-the art codes to streamline the calculations. It will undoubtedly benefit our community significantly.

The paper is well written and the matching procedure is explained quite well,. modulo my comments on the level of detail above. I will mention some specific sections/concepts that could be expanded upon in my list of points that the authors may wish to consider.

Having installed the program and run the examples in the paper, I can see that the tool works as advertised. See below for some specific question/issues.

In summary, I do not hesitate to recommend this article for publication in SciPost physics.

The comments below are mainly stylistic, or involve optional changes that I think may improve the discussion or make it more accessible to non-experts. Hopefully my questions, if not too naive, will help the authors add the extra bit of detail that would have avoided my confusion.

Requested changes

  1. There are some grammatical and typing errors in the manuscript, including missing prepositions. Further proofreading is recommended to iron these out.

  2. The process of calculating the amplitudes and performing the off-shell matching of the two theories at a particular scale is mostly described in words. The exception to this is the nice explanation of the way in which the hard region expansion to the loop amplitudes are obtained. I certainly think it would have been nice to see an explicit example that applies the MM procedure step-by-step, to help the reader to understand exactly what is being done, so that they can better use the tool for their own applications. Perhaps a subset of the exercise given in Appendix C might be a good place to include this. For example, the renormalisation procedure is not explicitly explained, even though I would have though it necessary to obtain amplitudes in both theories.

  3. In section 2.1, the procedure is summarised as "computing... the hard-region contribution to the one-light-particle-irreducible relevant amplitudes at tree and one-loop level in the UV theory and equating it to the tree level contribution in the EFT". Does this mean that, on UV theory side, the amplitudes are calculated up to one loop, while on the EFT side, they are kept at tree level? My understanding was that, in general, one should compute both sides to a given order in perturbation theory to achieve the matching.

  4. Section 2.2 in the "integration by parts" bullet. The authors mention setting scaleless integrals to zero, unless RGE mode is activated. This part could do with a little more explanation on why this is done.

  5. Right after, the identity for reducing the integrals to tadpoles is given. This kind of function (multiple powers of the same propagator) doesn't normally appear in a typical loop integral. I assume they arise from the hard region expansion, perhaps the authors could state this.

  6. Section 2.4 line 3 typo: Mathematica match -> Mathematica to match

  7. In the same section, the large redundancy of the BFM is mentioned as a source of cross checks that MM performs. This procedure could also be described in more detail, I did not really get the gist of what is being done here.

  8. The model creation section is very well presented. I had two very small comments: a) The input form for the SparseArrays to specify the generators, structure functions etc. is very opaque. Is it possible to define them in the more intuitive & readable way? e.g. SparseArray[ {index}->value, ... ] b) In 3.2.1 the BFM implementation is described, without much reference to the method itself i.e. assuming the reader is completely familiar with it. Perhaps the authors could point to som references here in order to put into context the Lagrangian for the gauge fields defined at the bottom of page 10. The last term, in particular, I am not familiar with as somebody who has never worked with the BFM.

  9. Section 3.3: a) on the need to include all operators of dimension equal to or less than those of interest, the possibility of it still working out if the matching failures occur in sectors that are not of interest is a bit vague. What is meant by "sector" Perhaps an example would help here. b) Refer to the section in which the proposed solution for the treatment of gamma^5 is discussed

  10. Section 3.4. I would refer to the protected words as "keywords". Furthermore, I find the use of dollar signs to be a nice convention to define protected keywords in mathematica modules that minimise the probability of accidental clashes. For example, one could prefix all internal MM-specific variables with "MM$", so "MM$iCPV" etc.

  11. 4.1, page 13: matching of matchmakereft -> matching of a matchmakereft

  12. When trying the CLI, that no messgae is printed when MM fails to create the model. I had to look into the log files to see that some syntax error was present in the .fr file. It would be useful to raise an error that is visible in the CLI when this occurs.

  13. Section 6.2, Page 17: a) Greeen -> Green b) M_{ij} and arbitrary -> M_{ij} an arbitrary

  14. Appendix B on gamma^5. Would it be more accurate to add "at one-loop" to "Limiting ourselves, at first, to matching renormalizable UV models to the SMEFT,..."

  15. Can the authors expand on the connection of this issue with the question of anomalies in the SMEFT, given the recent handful of papers discussing this points?

  16. Page 28: More than one scalar fields -> more than one scalar field

  17. Appendix C: I tried to run the scalar theory example using my MM installation and found it to run very smoothly. The following is alist of things I encountered that may help with debugging or better documenting. a) Running: match_model_to_eft two_scalars_MM one_scalar_MM I encountered the following mathematica warning: Part::partd: Part specification $Failed[[1]] is longer than depth of object.

b) I was not able to reconcile the results in MatchingResults.dat with the equations (79) onwards. For example, for the tree level matching result i get the following: { alpha2mass -> mL^2, alpha4 -> lambda0 - (3kappa^2)/MH^2, alpha4kin -> 1, alpha6 -> (45kappa^2lambda2)/MH^4, alpha6Rhat -> 0, alpha6Rtilde -> (4kappa^2)/MH^4 }

which doesn't match with (79)-(82).

c) I also noted the appearance in invepsilonbar in the one-loop results. This suggests the presence of poles in the matching result and the possibility is generally not discussed in the article. Could the authors explain their appearance, if they are expected or not, and if they are expected, how should they be interpreted and related to the renormalization.

d) Equations (96) & (97), there is a factor 2 w.r.t the definition of beta(aV) defined in (94)

e) Below Eq (112), the equation referenced is (93) but it seems to be that it should be (92).

  1. Appendix D, Green Basis. I was confused by the sentence that begins "We should stress that in our case, redundant and evanescent operators..." coudl the authors clarify what is meant?

  2. Page 37: can not -> cannot

  • validity: top
  • significance: top
  • originality: top
  • clarity: good
  • formatting: excellent
  • grammar: excellent

Author:  Jose Santiago  on 2022-05-23  [id 2507]

(in reply to Report 2 on 2022-03-30)
Category:
answer to question
correction

We would like to thank the referee for the thorough report and suggestions. We think all the points raised by the referee are important and we have tried to address all of them. Before detailing how we have done so we would like to emphasize that the main purpose of this article is to introduce matchmakereft as a very useful tool to perform one-loop matching calculations (including general anomalous dimensions). There are issues that are relevant for the tool itself in terms of what it does and how it does it and other issues that are entirely dependent on how the user defines the corresponding models. We have tried to explain how the user should define the models properly in order to obtain the correct result but the main emphasis of the paper is on the former issues (the ones entirely dependent on matchmakereft). Two prime examples are heavy vector bosons and evanescent operators. It is up to the user to define the corresponding models (or reduction in the case of evanescent operators) correctly. Matchmakereft is equipped to perform the calculations correctly if the models are defined properly. Furthermore, matchmakereft is a very much alive code that keeps being maintained and expanded in capabilities by us (as of today we are currently on version 1.0.10, with some interesting improvements with respect to the original version as briefly discussed in the note added section of our revised version. Thus, some features that are not available in the current version are likely to be developed in future versions.

Let us now go to the details of how we have addressed the points raised by the referee:

  1. We have read carefully the manuscript and hope to have fixed all typos now.

  2. We have added a few lines in Section 2.1 explaining more in detail the renormalization process and why the amplitudes are computed only at the tree-level in the EFT. We think going in much more detail is a bit out of the scope of the paper and we refer to the excellent lectures by Manohar and Cohen, that explain this process in full detail.

  3. We hope our response to requested change number 2 also covers requested change number 3.

  4. We have added a few sentences with a more detailed explanation, again, threading it with the ones in the previous two points.

  5. We have added the requested sentence.

  6. Fixed (thanks!)

  7. Done.

  8. (a) As long as they are defined as SparseArray they are properly read by Matchmakereft. In any case we have added a sentence where the example is given. (b) Indeed we are surprised that we forgot to include a reference. It is now added (Thanks a lot!). Also, there was a typo in the definition of the BFM Lagrangian (a typo not in Matchmakereft but in the transcription to the paper) which is now fixed.

  9. a) We have added a short explanation. b) Added the reference (thanks).

  10. We have changed words to keywords. We appreciate the suggestion by the referee but changing the notation at this point can be dangerous. We plan to implement a function to check that now protected keywords have been used in the model definition.

  11. Done.

  12. We fully agree with the referee and we are planning on implementing this in a future version of matchmakereft.

  13. Fixed (both).

  14. Added.

  15. Most of the recent discussions about anomalies in the SMEFT had to do with the bottom-up approach, i.e. whether the absence of anomalies introduced any constraints on the Wilson coefficients of higher-dimensional operators beyond the ones present at the renormalizable level. This has nothing to do with our top-down approach in which we match non-anomalous UV theories onto the SMEFT.

  16. Done.

  17. a) Indeed, it is a harmless warning that we are looking into and we will fix in the next version of matchmakereft. b) It was indeed a typo and it is now fixed. c) We have added a sentence in Section 2.1 emphasizing that we keep the 1/epsilon terms for the user to be able to cross-check UV and IR divergences but they should be removed for the renormalized Wilson coefficients. These 1/epsilon terms are both of UV and IR origin. d) We have now explicitly included a tadpole term in the UV model when computing the RGEs. This makes the calculation more transparent. We have also fixed a typo that we had before in the beta function (the factor of 2 mentioned by the referee) and the two calculations now agree. e) Fixed (thanks!).

  18. We have added a sentence clarifying what we mean. In the literature the label ”redundant” is sometimes used to denote an operator that vanishes on-shell (a condition that allows to relate some operators to others). In our case we call a redundant operator a non-vanishing operator that is equivalent on-shell to other operators in the physical basis. The same goes for evanescent operators. We hope it is clearer now.

  19. Done.

---

## Round 1 · Referee Report · Anonymous (Referee 3) · 2022-4-24

Strengths

The authors present a tool for one-loop matching that can also be used to calculate anomalous dimensions and perform changes of operator bases. The tool performs standard computational steps and relies on several other tools.

The program has a wide range of applicability and the paper is well written.

  1. Application to a wide class of models that can be matched.

  2. The program allows for an automatic calculation of one-loop anomalous dimensions (ADMs) for a wide class of Effective Theories (not only standard model degrees of freedom).

  3. The program allows for a translation of different EFT bases.

  4. Off-shell calculation allows for consistency checks that result from (low-energy) gauge invariance.

Weaknesses

Some computational steps are discussed in a potentially unnecessary level of detail (tensor reduction, partial fractions, integration by parts and Taylor expansion) while the relevant quantum field theoretical subtleties, that could occur in the calculations, are not discussed in much detail.

The weaknesses fall into two categories: Potential renormalisation issues and tests/applicability.

A one-loop matching calculation is partially scheme dependent (if tree-level is non-zero). In this context several questions arise:

  1. the authors discuss evanescent operators and expand in external momenta and masses, which leads to IR divergences that cancel with the UV divergences and render all EFT loop matrix elements zero. If an evanescent operator is generated at LO (maybe through Fierzing), are the authors sure that all finite operators renormalisations are considered in the EFT?

  2. Additional finite renormalisations might be necessary in anomalous theories, see e.g. hep-ph/9302240. Results are well known for one-loop calculations with renormalisable interactions and the authors should compare their matching results with the literature for diagrams that involve traces of gamma_5. The situation becomes more complicated when matching EFTs onto EFTs at one-loop level, but I am not sure if matchmaker is applicable in these scenarios. The authors should clarify for which models their code is applicable.

  3. Field renormalisations in matching calculations can be non-trivial. Minimal subtraction is a non-decoupling scheme, while decoupling is shown in MOM-schemes. This fixes the field renormalisation condition in order to achieve decoupling by hand in minimal subtraction. E.g. the hermitian and anti-hermitian field renormalisations are relevant for canceling gauge dependence or for the finite renormalisation of the CKM matrix. I did not see a detailed discussion of these properties.

  4. Specific models might require different regularisation schemes, such as DRED vs DREG. Is this relevant for matchmaker?

  5. When calculating ADMs of operators of different dimensions, one has to be careful that the Wilson coefficient can involve different powers of mu^epsilon so that each term in the Lagrangian has the same unit of dimension for epsilon unequal zero. This is relevant when calculating mu/d mu C. I understand that the tool is not primarily developed for ADMs, but a word of caution and reference to the literature might be useful.

The examples considered do not cover all possible applications of the program. It appears that only models with extra fermions and scalars have been considered.

  1. If one would consider extra vectors, one would need to perform the gauge fixing of the full gauge theory, before the symmetry breaking relevant for generating masses for the additional vectors occurs. Hence, it would be interesting to specify how the gauge is fixed for the light and heavy fields, and also if other types of gauge fixings are possible. Obviously, working with a non background gauge fixing would change the required EOM vanishing operators. A discussion of this issue would be useful. The authors might compare with the literature for possible examples.

  2. One should also consider additional examples, such as supersymmetric models, that have fermion flow "violation", possible finite normalisations, or other subtleties and compare with results in the literature.

Report

The program will be useful, the paper is well written and I recommend
the work for publication after the above points are considered.

Requested changes

The weaknesses 1 - 7 should be considered by the authors.

  • validity: high
  • significance: high
  • originality: good
  • clarity: high
  • formatting: perfect
  • grammar: excellent

Author:  Jose Santiago  on 2022-05-23  [id 2508]

(in reply to Report 3 on 2022-04-24)
Category:
answer to question
correction

We would like to thank the referee for the thorough report and suggestions. We think all the points raised by the referee are important and we have tried to address all of them. Before detailing how we have done so we would like to emphasize that the main purpose of this article is to introduce matchmakereft as a very useful tool to perform one-loop matching calculations (including general anomalous dimensions). There are issues that are relevant for the tool itself in terms of what it does and how it does it and other issues that are entirely dependent on how the user defines the corresponding models. We have tried to explain how the user should define the models properly in order to obtain the correct result but the main emphasis of the paper is on the former issues (the ones entirely dependent on matchmakereft). Two prime examples are heavy vector bosons and evanescent operators. It is up to the user to define the corresponding models (or reduction in the case of evanescent operators) correctly. Matchmakereft is equipped to perform the calculations correctly if the models are defined properly. Furthermore, matchmakereft is a very much alive code that keeps being maintained and expanded in capabilities by us (as of today we are currently on version 1.0.10, with some interesting improvements with respect to the original version as briefly discussed in the note added section of our revised version. Thus, some features that are not available in the current version are likely to be developed in future versions.

Let us now go to the details of how we have addressed the points raised by the referee:

  1. As mentioned above, matchmakereft is prepared to include all finite operator renormalisations correctly, assuming that they are properly included in the reduction of the evanescent structures by the user. This is one of the reasons (together with giving the user the possibility of tracking the UV and IR poles in order to check consistency of the UV-divergent contributions) to keep explicitly the 1/ ̄ε terms in the matching calculations as they can hit O(ε) evanescent structures to give finite contributions.

  2. Additional finite renormalisations are needed when using ’t Hooft-Veltman type of γ5 conventions. At the moment we have implemented an anti-commuting γ5 scheme that is sufficient for SMEFT-like EFTs (at least at one loop and mass dimension 6) that does not need finite renormalisations. We plan however to implement alternative γ5 schemes in future versions of matchmakereft, which will include finite renormalisations to eliminate spurious anomalies as needed.

  3. The matching procedure that we follow in matchmakereft is the standard one in MS-type renormalisation schemes as used in a large number of works in the literature and explained in detail in our Refs. [56,57] in our original submission (they are currently [59,60] in the revised one). Using the background field method for the gauge symmetries of the EFT ensures gauge-invariant results. (There was a missing reference for that, included now as [67] in the revised version).

  4. This is a similar case to the γ5 scheme choice. As we mention in the article, for the moment only DREG is implemented in matchmakereft. We plan to extend it to other schemes, including DRED, in future versions of the code.

  5. Indeed, matchmakereft is fully developed to compute ADMs (and in fact it has been tested against non-trivial calculations for the SMEFT at dimension 8 and for the SMEFT extended with an axion-like-particle (ALP). The dependence on μ to ensure the correct mass dimension for the renormalized Wilson coefficients is automatically computed and included in the calculation of the ADMs by matchmakereft.

  6. In principle any model (that can be implemented via FeynRules) can be used in matchmakereft. It is a matter of the user properly defining the corresponding model. Models with heavy vectors are less straightforward to define, as the referee says, as one has to fix the gauge in the full theory and include the Goldstone and ghost Lagrangians that come from the gauge-fixing procedure. The only real requirement from matchmakereft is that the background field method is used for the unbroken gauge symmetries of the EFT. Anything else is optional and the user can choose the gauge fixing procedure they consider best for the model at hand. Of course, they have to be consistent with the definition of the UV model, the EFT and the reduction of the Green to the Physical bases. This is however an issue more relevant to the model generation that to the actual matching methodology used by matchmakereft. Nevertheless, some of us are currently exploring specific models with heavy vectors and soon there will be full examples with these fields computed with matchmakereft (and cross-checked independently with other tools).

  7. We do have examples with fermion flow violation, including the see-saw model of type I and the charged scalar singlet or scalar leptoquarks, agreeing with the literature. We have performed a number of non-trivial examples, including models with heavy fermions and scalars and also with the calculation of non-trivial anomalous dimensions of a variety of EFTs. In all cases we have found that matchmakereft provides the correct result. Many of these models have finite wave-function renormalization. Thus, we feel like we have provided enough number of tests to be confident that matchmakereft produces the correct matching coefficients. We nevertheless keep adding new tests and will of course react to any sign of problems that can be eventually found. The fact that matchmakereft rests on a solid basis makes us confident that no significant changes will have to be implemented beyond possible coding bugs.

---

## Round 2 · Author Response

Dear Editor,

we would like to thank the referees for their detailed reports. We have responded to all of them in detail and we would like to resubmit a new version of our article in which all changes have been implemented.

---

## Round 2 · List of Changes

All the changes are detailed in the response to the three referee reports.

---

## Editorial Decision

published